# Networked Communication for Decentralised Agents in Mean-Field Games

## Abstract

We introduce networked communication to the mean-field game framework, in particular to oracle-free settings where $N$ decentralised agents learn along a single, non-episodic run of the empirical system. We prove that our architecture has sample guarantees bounded between those of the centralised- and independent-learning cases. We provide the order of the difference in these bounds in terms of network structure and number of communication rounds, and also contribute a policy-update stability guarantee. We discuss how the sample guarantees of the three theoretical algorithms do not actually result in practical convergence. We therefore show that in practical settings where the theoretical parameters are not observed (leading to poor estimation of the Q-function), our communication scheme significantly accelerates convergence over the independent case (and sometimes even the centralised case), without relying on the assumption of a centralised learner. We contribute further practical enhancements to all three theoretical algorithms, allowing us to present their first empirical demonstrations. Our experiments confirm that we can remove several of the theoretical assumptions of the algorithms, and display the empirical convergence benefits brought by our new networked communication. We additionally show that the networked approach has significant advantages, over both the centralised and independent alternatives, in terms of robustness to unexpected learning failures and to changes in population size.

## 1 Introduction

The mean-field game (MFG) framework (Lasry & Lions, 2007; Huang et al., 2006) has been used to address the difficulty faced by multi-agent reinforcement learning (MARL) regarding computational scalability as the number of agents increases. It models a representative agent as interacting not with the other individuals in the population on a per-agent basis, but instead with a distribution of other agents, known as the *mean field*. The MFG framework analyses the limiting case when the population consists of an infinite number of symmetric and anonymous agents, that is, they have identical reward and transition functions which depend on the mean-field distribution rather than on the actions of specific other players. In this work we focus on MFGs with stationary population distributions ('stationary MFGs', where learning is more tractable than in non-stationary ones) (Xie et al., 2021; Anahtarci et al., 2023; Zaman et al., 2023; Yardim et al., 2023), for which the solution concept is the MFG-Nash equilibrium (MFG-NE), which reflects the situation when each agent responds optimally to the population distribution that arises when all other agents follow that same optimal behaviour. The MFG-NE can be used as an approximation for the Nash equilibrium (NE) in a finite-agent game, with the error in the solution reducing as the number of agents $N$ tends to infinity (Anahtarci et al., 2023; Saldi et al., 2018; Yardim et al., 2024; Toumi et al., 2024; Hu & Zhang, 2024). MFGs have therefore been used to find approximate solutions for a wide variety of real-world problems involving a large but finite number of agents, which might otherwise have been too difficult to solve; see Appx. G for further details.

For large, complex many-agent systems in the real world (e.g. swarm robotics, autonomous vehicle traffic), it may be infeasible to find MFG-NEs analytically or via oracles/simulations of an infinite population (as they have been traditionally), such that learning must instead be conducted directly by the original finite population in its deployed environment. In such settings, in contrast to many previous methods, desirable qualities for MFG algorithms include: learning from the empirical distribution of $N$ agents (without generation/manipulation of this distribution by the algorithm itself

or by an external oracle); learning from a single continued system run that is not arbitrarily reset as in episodic learning; model-free learning; decentralisation; fast practical convergence; and robustness to unexpected failures of decentralised learners or changes in population size (Korecki et al., 2023).

Conversely, MFG frameworks have traditionally been largely theoretical, and methods for finding equilibria have often relied on assumptions that are too strong for real-world applications (Anahtarci et al., 2023; Lauriere et al., 2022; Perrin et al., 2020; Laurière et al., 2022; Guo et al., 2019a; Perrin et al., 2021; Elie et al., 2020; Carmona & Laurière, 2021; Cao et al., 2020; Germain et al., 2022; Fouque & Zhang, 2020; Algumaei et al., 2023; Angiuli et al., 2023); see Appx. G for an extended discussion of this related work. In particular, almost all prior work relies on a centralised controller to orchestrate the learning of all agents (Xie et al., 2021; Anahtarci et al., 2023; Zaman et al., 2023; Laurière et al., 2022; Guo et al., 2019b). However, outside of MFGs, the multi-agent systems community has recognised that the existence of a central controller is a very strong assumption, and one that can both restrict scalability by constituting a bottleneck for computation and communication, and reveal a single point of failure for the whole system (Zhang et al., 2021b; 2018; Wai et al., 2018; Zhang et al., 2021a; Chen et al., 2021; Jiang et al., 2024). For example, if the single server coordinating all of a smart city's autonomous vehicles were to crash, the entire road network would cease to operate. As an alternative, recent work has explored MFG algorithms for independent learning (Yardim et al., 2023; Mguni et al., 2018; Yongacoglu et al., 2022a;b; Grammatico et al., 2015a;b; Parise et al., 2015; Grammatico et al., 2016). However, prior works focus on theoretical sample guarantees instead of practical convergence speed, and have largely not considered robustness in the senses we address, despite fault-tolerance being an original motivation behind many-agent systems.

We address *all* of these desiderata by novelly introducing a communication network to the MFG setting. Communication networks have had success in other multi-agent settings, removing the reliance on inflexible, centralised structures (Zhang et al., 2021b; Wai et al., 2018; Zhang et al., 2021a; Chen et al., 2021; Doan et al., 2019; Lin et al., 2019; Heredia et al., 2020; Kar et al., 2013; Suttle et al., 2020). We focus on 'coordination games', i.e. where agents can increase their individual rewards by following the same strategy as others and therefore have an incentive to communicate policies, even if the MFG setting itself is technically non-cooperative. Thus our work can be applied to real-world problems in e.g. traffic signal control, formation control in swarm robotics, and consensus and synchronisation e.g. for sensor networks.

In this work, we show that when the agents' state-action value functions (Q-functions) can be only roughly estimated due to fewer samples/updates, possibly leading to high variance in policy updates, then propagating policies that are estimated to be better through the population via the communication network leads to faster convergence than that achieved by agents learning entirely independently. This is crucial in large complex environments that may be encountered in real applications, where the idealised hyperparameter choices (such as learning rates and numbers of iterations) required in previous works for theoretical convergence guarantees will be infeasible in practice. We compare our networked architecture with modified versions of earlier theoretical algorithms for the centralised and independent settings; we extend the original algorithms with experience replay buffers, without which we found them unable to demonstrate any learning in practical time. While the use of buffers means that the original theoretical sample guarantees no longer apply, we argue that this is preferable since these guarantees were in any case impractical. On this basis, we conduct numerical comparisons of the three architectures, demonstrating the benefits of communication for both convergence speed and system robustness. For further discussion of how networked communication can benefit robustness in large multi-agent systems, see Appx. E. In summary, our key contributions include the following:

- We prove that a *theoretical* version of our new networked algorithm (Alg. 1) has sample guarantees bounded between those of the centralised and independent settings for learning with a single, non-episodic run of the empirical system. We provide the order of the difference in these bounds in terms of network structure and number of communication rounds, and also contribute a policy-update stability guarantee (Sec. 3.3 and Appx. B.8).
- All three theoretical algorithms do not permit any learning in practical time; we show that in *practical* settings our communication scheme can significantly benefit convergence speed over the independent case, and sometimes even the centralised case (Sec. 3.4.1).
- We novelly modify all three theoretical algorithms (Alg. 2) to make their practical convergence feasible, most notably by including an experience replay buffer, allowing us to

contribute the first empirical demonstrations of all three algorithms (Sec. 3.4.2). An ablation study of the replay buffer is given in Appx. F.4 - agents do not seem to learn at all without it.

- Our experiments demonstrate the convergence benefits brought by our networked communication, and show we can remove several of the algorithms' theoretical assumptions (a goal shared by other work on the practicality of MFG algorithms (Cui et al., 2024)) (Sec. 4.1).

- We further demonstrate that our decentralised communication architecture brings significant benefits over both the centralised and independent alternatives in terms of robustness to unexpected learning failures and changes in population size (Sec. 4.1 and Appx. F.4).

The main paper is structured as follows: notation and preliminaries are given in Sec. 2; the theoretical algorithms and results are presented in Secs. 3.1-3.3; practical enhancements to the algorithms are given in Sec. 3.4; and experiments and discussion are provided in Sec. 4. Limitations and future work are found in Appx. H, and the broader social impact is considered in Sec. 5.

## 2 PRELIMINARIES

We use the following notation. $N$ is the number of agents in a population, with $\mathcal{S}$ and $\mathcal{A}$ representing the finite state and common action spaces, respectively. The sets $\mathcal{S}$ and $\mathcal{A}$ are equipped with the discrete metric $d(x, y) = \mathbb{1}_{x \neq y}$. The set of probability measures on a finite set $\mathcal{X}$ is denoted $\Delta_{\mathcal{X}}$, and $\mathbf{e}_x \in \Delta_{\mathcal{X}}$ for $x \in \mathcal{X}$ is a one-hot vector with only the entry corresponding to $x$ set to 1, and all others set to 0. For time $t \geq 0$, $\hat{\mu}_t = \frac{1}{N} \sum_{i=1}^{N} \sum_{s \in \mathcal{S}} \mathbb{1}_{s_t^i = s} \mathbf{e}_s \in \Delta_{\mathcal{S}}$ is a vector denoting the empirical state distribution of the $N$ agents at time $t$. The set of policies is $\Pi = \{\pi : \mathcal{S} \to \Delta_{\mathcal{A}}\}$, and the set of Q-functions is denoted $\mathcal{Q} = \{q : \mathcal{S} \times \mathcal{A} \to \mathbb{R}\}$. For $\pi, \pi' \in \Pi$ and $q, q' \in \mathcal{Q}$, we have the norms $||\pi - \pi'||_1 := \sup_{s \in \mathcal{S}} ||\pi(s) - \pi'(s)||_1$ and $||q - q'||_{\infty} := \sup_{s \in \mathcal{S}, a \in \mathcal{A}} |q(s, a) - q'(s, a)|$.

Function $h : \Delta_{\mathcal{A}} \to \mathbb{R}_{\geq 0}$ denotes a strongly concave function, which we implement as the scaled entropy regulariser $\lambda h_{ent}(u) = -\lambda \sum_a u(a) \log u(a)$, for $a \in \mathcal{A}$, $u \in \Delta_{\mathcal{A}}$ and $\lambda > 0$. As in some earlier works (Anahtarci et al., 2023; Yardim et al., 2023; Algumaei et al., 2023; Cui & Koeppl, 2021; Guo et al., 2022; Yu & Yuan, 2023), regularisation is theoretically required to ensure the contractivity of operators and continued exploration, and hence algorithmic convergence. However, it has been recognised that modifying the RL objective in this way can bias the NE (Yardim et al., 2023; Hu & Zhang, 2024; Lauriere et al., 2022; Su & Lu, 2022). We show in our experiments that we are able to reduce $\lambda$ to 0 with no detriment to convergence.

**Definition 1** (*N*-player symmetric anonymous games). *An N-player stochastic game with symmetric, anonymous agents is given by the tuple $\langle N, \mathcal{S}, \mathcal{A}, P, R, \gamma \rangle$, where $\mathcal{A}$ is the action space, identical for each agent; $\mathcal{S}$ is the identical state space of each agent, such that their initial states are $\{s_0^i\}_{i=1}^{N} \in \mathcal{S}^N$ and their policies are $\{\pi^i\}_{i=1}^{N} \in \Pi^N$. $P : \mathcal{S} \times \mathcal{A} \times \Delta_{\mathcal{S}} \to \Delta_{\mathcal{S}}$ is the transition function and $R : \mathcal{S} \times \mathcal{A} \times \Delta_{\mathcal{S}} \to [0,1]$ is the reward function, which map each agent's local state and action and the population's empirical distribution to transition probabilities and bounded rewards, respectively, i.e.*

$$s_{t+1}^i \sim P(\cdot | s_t^i, a_t^i, \hat{\mu}_t), \quad r_t^i = R(s_t^i, a_t^i, \hat{\mu}_t), \quad \forall i = 1, \ldots, N.$$

The policy of an agent is given by $a_t^i \sim \pi^i(s_t^i)$, that is, each agent only observes its own state, and not the joint state or empirical distribution of the population.

**Definition 2** (*N*-player discounted regularised return). *With joint policies $\boldsymbol{\pi} := (\pi^1, \ldots, \pi^N) \in \Pi^N$, initial states sampled from a distribution $\upsilon_0 \in \Delta_{\mathcal{S}}$ and $\gamma \in [0,1)$ as a discount factor, the expected discounted regularised returns of each agent $i$ in the symmetric anonymous game are given by*

$$\Psi_h^i(\boldsymbol{\pi}, \upsilon_0) = \mathbb{E}\left[\sum_{t=0}^{\infty} \gamma^t (R(s_t^i, a_t^i, \hat{\mu}_t) + h(\pi^i(s_t^i))) \Big|_{\substack{s_0^j \sim \upsilon_0 \\ a_0^j \sim \pi^j(s_t^j) \\ s_{t+1}^j \sim P(\cdot | s_t^j, a_t^j, \hat{\mu}_t)}}, \forall t \geq 0, j \in \{1, \ldots, N\}\right].$$

**Definition 3** ($\delta$-NE). *Say $\delta > 0$ and $(\pi, \boldsymbol{\pi}^{-i}) := (\pi^1, \ldots \pi^{i-1}, \pi, \pi^{i+1}, \ldots, \pi^N) \in \Pi^N$. An initial distribution $\upsilon_0 \in \Delta_{\mathcal{S}}$ and an N-tuple of policies $\boldsymbol{\pi} := (\pi^1, \ldots, \pi^N) \in \Pi^N$ form a $\delta$-NE $(\boldsymbol{\pi}, \upsilon_0)$ if*

$$\Psi_h^i(\boldsymbol{\pi}, \upsilon_0) \geq \max_{\pi \in \Pi} \Psi_h^i((\pi, \boldsymbol{\pi}^{-i}), \upsilon_0) - \delta \quad \forall i = 1, \ldots, N.$$

At the limit as $N \to \infty$, the population of infinitely many agents can be characterised as a limit distribution $\mu \in \Delta_{\mathcal{S}}$. We denote the expected discounted return of the representative agent in the infinite-agent game - termed an MFG - as $V$, rather than $\Psi$ as in the finite $N$-agent case.

**Definition 4** (Mean-field discounted regularised return). *For a policy-population pair $(\pi, \mu) \in \Pi \times \Delta_{\mathcal{S}}$,*

$$V_h(\pi, \mu) = \mathbb{E}\left[\sum_{t=0}^{\infty} \gamma^t (R(s_t, a_t, \mu) + h(\pi(s_t))) \Bigg|_{s_{t+1} \sim P(\cdot|s_t, a_t, \mu)}^{s_0 \sim \mu}\right].$$

A stationary MFG is one that has a unique population distribution that is stable with respect to a given policy, and the agents' policies are not time- or population-dependent.

**Definition 5** (NE of stationary MFG). *For a policy $\pi^* \in \Pi$ and a population distribution $\mu^* \in \Delta_{\mathcal{S}}$, the pair $(\pi^*, \mu^*)$ is a stationary MFG-NE if the following optimality and stability conditions hold:*

$$optimality: \quad V_h(\pi^*, \mu^*) = \max_\pi V_h(\pi, \mu^*),$$

$$stability: \quad \mu^*(s) = \sum_{s', a'} \mu^*(s') \pi^*(a'|s') P(s|s', a', \mu^*).$$

*If the optimality condition is only satisfied with $V_h(\pi_\delta^*, \mu_\delta^*) \geq \max_\pi V_h(\pi, \mu_\delta^*) - \delta$, then $(\pi_\delta^*, \mu_\delta^*)$ is a $\delta$-NE of the MFG, where $\mu_\delta^*$ is obtained from the stability equation and $\pi_\delta^*$.*

The MFG-NE is an approximate NE of the finite $N$-player game in which we may have originally been interested but which is difficult to solve in itself (Yardim et al., 2023; Lauriere et al., 2022):

**Proposition 1** ($N$-player NE and MFG-NE (Thm. 1, (Anahtarci et al., 2023))). *If $(\pi^*, \mu^*)$ is a MFG-NE, then, under certain Lipschitz conditions (Anahtarci et al., 2023), for any $\delta > 0$, there exists $N(\delta) \in \mathbb{N}_{>0}$ such that, for all $N \geq N(\delta)$, the joint policy $\boldsymbol{\pi} = \{\pi^*, \pi^*, \ldots, \pi^*\} \in \Pi^N$ is a $\delta$-NE of the $N$-player game.*

**Remark 1.** *It can be shown that $\delta$ can be characterised further in terms of $N$, with $(\pi^*, \mu^*)$ being an $\mathcal{O}(\frac{1}{\sqrt{N}})$-NE of the $N$-player symmetric anonymous game (Yardim et al., 2023).*

For our new, networked learning algorithm, we also introduce the concept of a time-varying communication network, where the links between agents that make up the network may change at each time step $t$. Most commonly we might think of such a network as depending on the spatial locations of decentralised agents, such as physical robots, which can communicate with neighbours that fall within a given broadcast radius. When the agents move in the environment, their neighbours and therefore communication links may change. However, the dynamic network can also depend on other factors that may or may not depend on each agent's state $s_t^i$. For example, even a network of fixed-location agents can change depending on which agents are active and broadcasting at a given time $t$, or if their broadcast radius changes, perhaps in relation to signal or battery strength.

**Definition 6** (Time-varying communication network). *The time-varying communication network $\{\mathcal{G}_t\}_{t \geq 0}$ is given by $\mathcal{G}_t = (\mathcal{N}, \mathcal{E}_t)$, where $\mathcal{N}$ is the set of vertices each representing an agent $i = 1, \ldots, N$, and the edge set $\mathcal{E}_t \subseteq \{(i,j) : i,j \in \mathcal{N}, i \neq j\}$ is the set of undirected communication links by which information can be shared at time $t$.*

A network is *connected* if there is a sequence of distinct edges forming a path between each distinct pair of vertices. The *union* of a collection of graphs $\{\mathcal{G}_t, \mathcal{G}_{t+1}, \cdots, \mathcal{G}_{t+\omega}\}$ ($\omega \in \mathbb{N}$) is the graph with vertices and edge set equalling the union of the vertices and edge sets of the graphs in the collection (Jadbabaie et al., 2003). A collection is *jointly connected* if its members' union is connected. A network's *diameter* $d_\mathcal{G}$ is the maximum of the shortest path length between any pair of nodes.

## 3 LEARNING WITH NETWORKED, DECENTRALISED AGENTS

**Summary** We first discuss theoretical versions of our operators and algorithm (Secs. 3.1, 3.2) to show that our networked framework has sample guarantees bounded between those of the centralised- and independent-learning cases (Sec. 3.3). We then show that our novel incorporation of an experience replay buffer (Sec. 3.4.2), along with networked communication, means that empirically we can remove many of the theoretical assumptions and practically infeasible hyperparameter choices that are required by the sample guarantees of the theoretical algorithms, in which cases we demonstrate that our networked algorithm can significantly outperform the independent algorithm, and sometimes even the centralised one (Sec. 4).

### 3.1 LEARNING WITH $N$ AGENTS FROM A SINGLE RUN

We begin by outlining the basic procedure for solving the MFG using the $N$-agent empirical distribution and a single, continuous system run. The two underlying operators are the same for the centralised, independent and networked architectures; in the latter two cases all agents apply the operators individually, while in the centralised setting a single 'central' agent (the agent with index $i = 1$) estimates the Q-function and computes an updated policy that is pushed to all the other agents.

We define, for $h_{\max} > 0$ and $h : \Delta_{\mathcal{A}} \to [0, h_{\max}]$, $u_{\max} \in \Delta_{\mathcal{A}}$ such that $h(u_{\max}) = h_{\max}$. We further define $q_{\max} := \frac{1+h_{\max}}{1-\gamma}$, and set $\pi_{\max} \in \Pi$ such that $\pi_{\max}(s) = u_{\max}, \forall s \in \mathcal{S}$. For any $\Delta h \in \mathbb{R}_{>0}$, we also define the convex set $\mathcal{U}_{\Delta h} := \{u \in \Delta_{\mathcal{A}} : h(u) \geq h_{\max} - \Delta h\}$.

Learning agents use the stochastic temporal difference (TD)-learning operator to repeatedly update an estimate of the Q-function of their current policy with respect to the current empirical distribution, i.e. to approximate the operator $\Gamma_q$ (Def. 12, Appx. A):

**Definition 7** (Stochastic TD-learning operator, simplified from Def. 4.1 in Yardim et al. (2023)). *We define $\mathcal{Z} := \mathcal{S} \times \mathcal{A} \times [0, 1] \times \mathcal{S} \times \mathcal{A}$, and say that $\zeta_t^i$ is the transition observed by agent $i$ at time $t$, given by $\zeta_t^i = (s_t^i, a_t^i, r_t^i, s_{t+1}^i, a_{t+1}^i)$. The TD-learning operator $\tilde{F}_\beta^\pi : \mathcal{Q} \times \mathcal{Z} \to \mathcal{Q}$ is defined, for any $Q \in \mathcal{Q}, \zeta_t \in \mathcal{Z}, \beta \in \mathbb{R}$, as*

$$\tilde{F}_\beta^\pi(Q, \zeta_t) = Q(s_t, a_t) - \beta \left( Q(s_t, a_t) - r_t - h(\pi(s_t)) - \gamma \left( Q(s_{t+1}, a_{t+1}) \right) \right).$$

Having estimated the Q-function of their current policy, agents update this policy by selecting, for each state, a probability distribution over their actions that maximises the combination of three terms (Def. 8): 1. the value of the given state with respect to the estimated Q-function; 2. a regulariser over the action probability distribution (in practice, we maximise the scaled entropy of the distribution); 3. a metric of similarity between the new action probabilities for the given state and those of the previous policy, given by the squared two-norm of the difference between the two distributions. We can alter the importance of the similarity metric relative to the other two terms by varying a parameter $\eta$, which is equivalent to changing the learning rate of the policy update. The three terms in the maximisation function can be seen in the policy mirror ascent (PMA) operator:

**Definition 8** (Policy mirror ascent operator (Def. 3.5, (Yardim et al., 2023))). *For a learning rate $\eta > 0$ and $L_h := L_a + \gamma \frac{L_s K_a}{2 - \gamma K_s}$ (where these constants are defined in Assumption 1 in Appx. A), the PMA update operator $\Gamma_\eta^{md} : \mathcal{Q} \times \Pi \to \Pi$ is defined as*

$$\Gamma_\eta^{md}(Q, \pi)(s) := \arg\max_{u \in \mathcal{U}_{L_h}} \left( \langle u, q(s, \cdot) \rangle + h(u) - \frac{1}{2\eta} ||u - \pi(s)||_2^2 \right), \forall s \in \mathcal{S}, \forall Q \in \mathcal{Q}, \forall \pi \in \Pi.$$

The theoretical learning algorithm has three nested loops (see Lines 2, 4 and 5 of Alg. 1). The policy update is applied $K$ times. Before the policy update in each of the $K$ loops, agents update their estimate of the Q-function by applying the stochastic TD-learning operator $M_{pg}$ times. Prior to the TD update in each of the $M_{pg}$ loops, agents take $M_{td}$ steps in the environment without updating. The $M_{td}$ loops exist to create a delay between each TD update to reduce bias when using the empirical distribution to approximate the mean field in a single system run (Kotsalis et al., 2022). However, we find in our experiments that we are able to essentially remove the inner $M_{td}$ loops (Sec. 4.1).

### 3.2 DECENTRALISED COMMUNICATION BETWEEN AGENTS

In our novel algorithm Alg. 1, agents compute policy updates in a decentralised way as in the independent case (Lines 3-10), before exchanging policies with neighbours by the following method, which allows policies to spread through the population. Coupled to their updated policy $\pi_{k+1}^i$, agents generate a scalar value $\sigma_{k+1}^i$ (Line 11). The value provides information that helps agents decide between policies that they may wish to adopt from neighbours. Different methods for choosing between values received from neighbours, and for generating the values in the first place, lead to different policies spreading through the population. For example, generating or choosing $\sigma_{k+1}^i$ at random leads to policies being exchanged at random (required in Thm. 1), whereas generating $\sigma_{k+1}^i$ as an approximation of the return of $\pi_{k+1}^i$ and then selecting the highest received value of $\sigma_{k+1}^j$ leads to better performing policies spreading through the population. The latter is the approach we

---

**Algorithm 1** Networked learning with single system run

---

**Require:** loop parameters $K, M_{pg}, M_{td}, C$, learning parameters $\eta, \{\beta_m\}_{m \in \{0, \ldots, M_{pg}-1\}}, \lambda, \gamma,$
$\{\tau_k\}_{k \in \{0, \ldots, K-1\}}$

**Require:** initial states $\{s_0^i\}_i, i = 1, \ldots, N$

1: Set $\pi_0^i = \pi_{\max}, \forall i$ and $t \leftarrow 0$
2: **for** $k = 0, \ldots, K - 1$ **do**
3: $\quad \forall s, a, i : \hat{Q}_0^i(s, a) = Q_{\max}$
4: $\quad$ **for** $m = 0, \ldots, M_{pg} - 1$ **do**
5: $\quad\quad$ **for** $M_{td}$ iterations **do**
6: $\quad\quad\quad$ Take step $\forall i : a_t^i \sim \pi_k^i(\cdot|s_t^i), r_t^i = R(s_t^i, a_t^i, \hat{\mu}_t), s_{t+1}^i \sim P(\cdot|s_t^i, a_t^i, \hat{\mu}_t); t \leftarrow t + 1$
7: $\quad\quad$ **end for**
8: $\quad\quad$ Compute TD update $(\forall i)$: $\hat{Q}_{m+1}^i = \tilde{F}_{\beta_m}^{\pi_k^i}(\hat{Q}_m^i, \zeta_{t-2}^i)$ (see Def. 7)
9: $\quad$ **end for**
10: $\quad$ PMA step $\forall i : \pi_{k+1}^i = \Gamma_\eta^{md}(\hat{Q}_{M_{pg}}^i, \pi_k^i)$ (see Def. 8)
11: $\quad \forall i$ : Generate $\sigma_{k+1}^i$ associated with $\pi_{k+1}^i$
12: $\quad$ **for** $C$ rounds **do**
13: $\quad\quad \forall i$ : Broadcast $\sigma_{k+1}^i, \pi_{k+1}^i$
14: $\quad\quad \forall i : J_t^i = i \cup \{j \in \mathcal{N} : (i, j) \in \mathcal{E}_t\}$
15: $\quad\quad \forall i$ : Select adopted$^i \sim \Pr\big(\text{adopted}^i = j\big) = \frac{\exp{(\sigma_{k+1}^j/\tau_k)}}{\sum_{x \in J_t^i} \exp{(\sigma_{k+1}^x/\tau_k)}} \, \forall j \in J_t^i$
16: $\quad\quad \forall i : \sigma_{k+1}^i \leftarrow \sigma_{k+1}^{\text{adopted}^i}, \pi_{k+1}^i \leftarrow \pi_{k+1}^{\text{adopted}^i}$
17: $\quad\quad$ Take step $\forall i : a_t^i \sim \pi_{k+1}^i(\cdot|s_t^i), r_t^i = R(s_t^i, a_t^i, \hat{\mu}_t), s_{t+1}^i \sim P(\cdot|s_t^i, a_t^i, \hat{\mu}_t); t \leftarrow t + 1$
18: $\quad$ **end for**
19: **end for**
20: Return policies $\{\pi_K^i\}_i, i = 1, \ldots, N$

---

use for accelerating empirical convergence (described in Sec. 3.4.1 on the practical running of our algorithm), albeit we use a softmax rather than a max function for selecting between received values. However, for our theoretical results, we do not focus on a specific method for generating $\sigma_{k+1}^i$, such that it can be arbitrary for Thms. 1 and 4 below, and with few restrictions for Thms. 2 and 3.

Agents broadcast their policy $\pi_{k+1}^i$ and the associated $\sigma_{k+1}^i$ value to their neighbours (Line 13). Agents have a certain broadcast radius, defining the structure of the possibly time-varying communication network. Of the policies and associated values received by a given agent (including its own) (Line 14), the agent selects a $\sigma_{k+1}^j$ with a probability defined by a softmax function over the received values, and *adopts* the policy associated with this $\sigma_{k+1}^i$, i.e. it sets its own current $\pi_{k+1}^i$ and $\sigma_{k+1}^i$ to the ones it has selected (Lines 15, 16). This process continues for $C$ communication rounds, before the Q-function estimation steps begin again. After each round, the agents take a step in the environment (Line 17), such that if the communication network is affected by the agents' states, then agents that are unconnected from any others in a given communication round might become connected in the next. (In our experiments we set $C$ as 1 to show the benefits to convergence speed brought by even a single communication round.) We assume the softmax function is subject to a possibly time-varying temperature parameter $\tau_k$. We discuss the effects of the values of $C$ and $\tau_k$, and the mechanism for generating $\sigma_{k+1}^i$, in subsequent sections.

**Remark 2.** *Our networked architecture is effectively a generalisation of both the centralised and independent settings (Algs. 2, 3, Yardim et al. (2023)). The independent setting is the special case where there is no communication, i.e. $C = 0$. The centralised setting is the special case when $\sigma_{k+1}^i$ is generated from a unique ID for each agent, with the central learner agent assumed to generate the highest value by default. In this case we assume $\tau_k \to 0$ (such that the softmax becomes a max function), and that the communication network becomes jointly connected repeatedly, so the central learner's policy is always adopted by the entire population, assuming $C$ is large enough that the number of jointly connected collections of graphs occurring within $C$ is equal to the largest diameter of the union of any collection (Rajagopalan & Shah, 2010; Zhang et al., 2020).*

**Remark 3.** *In practice, when referring in the following to a centralised version of the networked Alg. 1, for simplicity we assume there is no communication and instead that the updated policy $\pi^1_{k+1}$ of the central learner $i = 1$ is pushed to all other agents after Line 10, as in Alg. 2 of (Yardim et al., 2023).*

## 3.3 PROPERTIES OF POLICY ADOPTION

We first give two theoretical results comparing the sample guarantees of our networked case with those of the other settings; the results respectively depend on whether the networked agents select which communicated policies to adopt at random or not. We then provide the order of the difference in these bounds in the non-random case in terms of network structure and number of communication rounds. We finally give in Appx. B.8 a policy-update stability guarantee, which applies in all scenarios. Appx. A contains the full set of technical assumptions on which these theorems rely.

**Theorem 1** (Networked learning with **random adoption**). *Full version in Appx. B.2. Say that $\pi^*$ is the unique MFG-NE policy. Let $\varepsilon > 0$ be an arbitrary value that reduces as $k$ increases by the following relation: $k = \frac{\log 8\varepsilon^{-1}}{\log L_{\Gamma_\eta}^{-1}}$, where the constant $L_{\Gamma_\eta}$ is defined in Lem. 2 in Appx. A. Let us assume that $C > 0$ and $\tau_k \to \infty$, meaning that the softmax function approaches a uniform distribution such that the values of $\sigma^i_{k+1}$ are arbitrary and received policies are adopted at random. Then, under the technical assumptions given in Appx. B.2, the random output $\{\pi^i_K\}_i$ of Alg. 1 preserves the sample guarantees of the independent-learning case given in Lem. 3, i.e. the output satisfies, for all agents $i = 1, \ldots, N$, $\mathbb{E}\left[||\pi^i_K - \pi^*||_1\right] \leq \varepsilon + \mathcal{O}\left(\frac{1}{\sqrt{N}}\right)$.*

**Proof sketch**. Random exchange of policies learnt in a decentralised manner does not change the expectation of the random output of the purely independent-learning setting i.e. where this exchange does not occur. Full proof in Appx. B.3.

Moreover we can show that if $\sigma^i_{k+1}$ is generated arbitrarily and uniquely for each $i$, then for $\tau_k \in \mathbb{R}_{>0}$ (such that the softmax function gives a non-uniform distribution and adoption of received policies is therefore non-random), the sample complexity of the networked learning algorithm is bounded between that of the centralised and independent algorithms:

**Theorem 2** (Networked learning with **non-random adoption**). *Full version in Appx. B.5. Assume that Alg. 1 is run as in Thm. 1, except now $\tau_k \in \mathbb{R}_{>0}$. $\varepsilon$ is defined as above. Assume also that $\sigma^i_{k+1}$ is generated uniquely for each $i$, in a manner independent of any metric related to $\pi^i_{k+1}$, e.g. $\sigma^i_{k+1}$ is random or related only to the index $i$ (so as not to bias the spread of any particular policy). Let the random output of this Algorithm be denoted as $\{\pi^{i,net}_K\}_i$. Also consider an independent-learning version of the algorithm (i.e. with the same parameters except $C = 0$) and denote its random output $\{\pi^{i,ind}_K\}_i$; and a centralised version of the algorithm with the same parameters (see Rem. 3) and denote its random output as $\pi^{cent}_K$. Then, under the technical assumptions given in Appx. B.5, for all agents $i = 1, \ldots, N$, the random outputs $\{\pi^{i,net}_K\}_i$, $\{\pi^{i,ind}_K\}_i$ and $\pi^{cent}_K$ satisfy the following relations, where $ub_{net}$, $ub_{ind}$ and $ub_{cent}$ are respective upper bounds for each case:*

$$\mathbb{E}\left[||\pi^{cent}_K - \pi^*||_1\right] \leq ub_{cent}, \quad \mathbb{E}\left[||\pi^{i,net}_K - \pi^*||_1\right] \leq ub_{net}, \quad \mathbb{E}\left[||\pi^{i,ind}_K - \pi^*||_1\right] \leq ub_{ind},$$

$$where \quad ub_{cent} \leq ub_{net} \leq ub_{ind} = \varepsilon + \mathcal{O}\left(\frac{1}{\sqrt{N}}\right).$$

**Proof sketch**. Fewer distinct policies in the population means there is less bias in each learner's estimation of the Q function. So methods for reducing the number of policies in the population (such as non-random adoption in the networked case) lead to faster convergence. Full proof in Appx. B.6.

**Theorem 3** (Relation between communication network structure and order of difference between the frameworks' bounds). *In addition to the assumptions in Thm. 2, now also assume that the communication network $\mathcal{G}_t$ remains static and connected during the $C$ communication rounds, and that $\tau_k \to 0 \; \forall k$ such that the softmax essentially becomes a max function. Assume also the diameter $d_\mathcal{G}$ of the network is equal for all $k$. Then we can say that the difference in the upper bounds $ub_{net}$, $ub_{ind}$ and $ub_{cent}$ from Thm. 2 depends on $C$ and $d_\mathcal{G}$ as follows, for the tight bound big Theta ($\Theta$):*

$$ub_{cent} + \Theta\left(f(C, d_\mathcal{G})\right) \approx ub_{net} \approx ub_{ind} - \Theta\left(1 - f(C, d_\mathcal{G})\right),$$

*for the piecewise function $f(C, d_{\mathcal{G}})$ defined as*

$$f(C, d_{\mathcal{G}}) = \begin{cases} \left( \left( 1 - \frac{1}{d_{\mathcal{G}}} \right)^C \right) & \text{if } C < d_{\mathcal{G}}, \\ 0 & \text{if } C \geq d_{\mathcal{G}} \end{cases}.$$

*When $C \geq d_{\mathcal{G}}$, $ub_{net} = ub_{cent}$, so for $C > d_{\mathcal{G}}$ there is no additional improvement over the centralised bound. Equally when $C = 0$, we have exactly $ub_{net} = ub_{ind}$.*

**Proof sketch**. The difference in the architectures' convergence rates depends on the different amounts of divergence between the population's policies. In a static, connected network, max-consensus is reached when the number of communication rounds $C$ is equal to the diameter $d_{\mathcal{G}}$, giving 0 divergence as in the centralised case. The rate of divergence decrease is approximately $\frac{1}{d_{\mathcal{G}}}$ for each of the $C$ rounds, giving the exponential relation involving $C$ and $d_{\mathcal{G}}$. Full proof in Appx. B.7.

**Remark 4.** *Thm. 3 depends on the assumptions that the communication network is static and fixed, and has the same diameter $d_{\mathcal{G}}$ for all $k$. If we assume instead that the network is only repeatedly jointly connected, we can replace $d_{\mathcal{G}}$ in the results in Thm. 3 with $d_{avg} \cdot \omega$, namely the average diameter of the union of each jointly connected collection of graphs multiplied by the average number of graphs in each jointly connected collection. As noted in Rem. 2, max-consensus is reached if $C$ is large enough that the number of jointly connected collections of graphs occurring within $C$ is equal to the largest diameter of the union of any collection, giving the centralised case; there is no added benefit to higher values of $C$ than this.*

See Rem. 7 in Appx. B.7 for a similar remark loosening the assumption that $\tau_k \to 0 \; \forall k$.

### 3.4 PRACTICAL RUNNING OF ALGORITHMS

#### 3.4.1 GENERATION OF $\sigma_{k+1}^i$

The theoretical analysis in Sec. 3.3 requires algorithmic hyperparameters that render learning impractically slow in all of the centralised, independent and networked cases (see Rem. 8, Appx. B.9). For practical convergence of the algorithms, we seek to drastically increase $\{\beta_m\}$ and reduce $M_{td}$ and $M_{pg}$, though this will naturally break the theoretical guarantees and give a poorer estimation of the Q-function $\hat{Q}_{M_{pg}}^i$, and hence a greater variance in the quality of the updated policies $\pi_{k+1}^i$. However, in such cases we found empirically that an appropriate method for generating $\sigma_{k+1}^i$ dependent on $\pi_{k+1}^i$ allows our networked algorithm to significantly outperform the independent setting, and sometimes even the centralised setting, by advantageously biasing the spread of particular policies. This is instead of generating $\sigma_{k+1}^i$ arbitrarily as required in the theoretical settings in Sec. 3.3.

We do so by setting $\sigma_{k+1}^i$ to a finite approximation $\widehat{\Psi_{h,k+1}^i}(\boldsymbol{\pi}_{k+1}, \upsilon_0)$ of $\Psi_{h,k+1}^i(\boldsymbol{\pi}_{k+1}, \upsilon_0)$ where $\boldsymbol{\pi}_{k+1} := (\pi_{k+1}^1, \ldots, \pi_{k+1}^N)$, by tracking the discounted return for $E$ evaluation steps. This is given by

$$\widehat{\Psi_{h,k+1}^i}(\boldsymbol{\pi}_{k+1}, \upsilon_0) = \left[ \sum_{e=0}^{E} \gamma^e (R(s_t^i, a_t^i, \hat{\mu}_t) + h(\pi^i(s_t^i))) \Bigg|_{\substack{t=t+e \\ a_t^j \sim \pi_{k+1}^j(s_t^j) \\ s_{t+1}^j \sim P(\cdot|s_t^j, a_t^j, \hat{\mu}_t)}}, \forall j \in \{1, \ldots, N\} \right].$$

Generating $\sigma_{k+1}^i$ in this way means the policies that are more likely to be adopted and spread through the network are those that are estimated to receive a higher return in reality, despite being generated from poorly estimated Q-functions. This explains why our networked method can in practice outperform even the centralised case, where the updated policy of the arbitrary agent $i = 1$ gets pushed to all other agents regardless of its quality (see Appx. C for a more formal explanation). Naturally the quality of the finite approximation depends on the number of evaluation steps $E$, but we found empirically that $E$ can be much smaller than $M_{pg}$ and still give marked convergence benefits.

#### 3.4.2 ALGORITHM ACCELERATION BY USE OF EXPERIENCE-REPLAY BUFFER

Even with networked communication, the empirical learning of our original algorithm is too slow for practical demonstration, as also in the centralised and independent cases - see the ablation study in Appx. F.4. We therefore offer a further technical contribution allowing the first practical demonstrations of all three architectures for learning from a single continued system run.

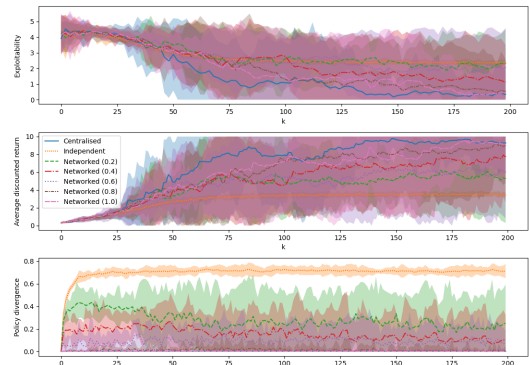

Figure 1: 'Target agreement' game. Even with only a single communication round, our networked case outperforms the independent case wrt. exploitability, and markedly outperforms wrt. return. The fact that the lowest broadcast radius (0.2) ends with similar exploitability to the independent case yet much higher return shows our networked algorithm can help agents find 'preferable' equilibria.

We modify our Alg. 1 (shown in *blue* in Alg. 2, Appx. D), as follows. Instead of using a transition $\zeta_{t-2}^i$ to compute the TD update within each $M_{pg}$ iteration and then discarding the transition, we store the transition in a buffer (Line 9) until after the $M_{pg}$ loops. Replay buffers are a common (MA)RL tool used especially with deep learning, precisely to improve data efficiency and reduce autocorrelation (Lin, 1992; Fedus et al., 2020; Xu et al., 2024). When learning does take place in our modified algorithm (Lines 11-16), it involves cycling through the buffer for $L$ iterations - randomly shuffling the buffer between each - and thus conducting the TD update on each stored transition $L$ times. This allows us to reduce the number of $M_{pg}$ loops, as well as not requiring as small a learning rate $\{\beta_m\}$, allowing much faster learning in practice. Moreover, by shuffling the buffer before each cycle we reduce bias resulting from the dependency of samples along the single path, which may justify being able to achieve adequate stable learning even when reducing the number of $M_{td}$ waiting steps within each $M_{pg}$ loop. The buffer means the theoretical guarantees given in Sec. 3.3 no longer apply, but we exchange this for practical convergence times. See Appx. D for further discussion.

## 4 EXPERIMENTS

Our technical contribution of the replay buffer to MFG algorithms for learning from continuous system runs allows us also to contribute the first empirical demonstrations of these algorithms, not just in the networked case but also in the centralised and independent cases. The latter two serve as baselines to show the advantages of the networked architecture. We follow prior works on stationary MFGs in the types of game demonstrated (Zaman et al., 2023; Lauriere et al., 2022; Algumaei et al., 2023; Lauriere, 2021; Cui et al., 2023). We focus on grid-world environments where agents can move in the four cardinal directions or remain in place. We present results from two tasks defined by the agents' reward functions; see Appx. F.1 for full technical description of our task settings.

**Cluster.** Agents are rewarded for gathering together. The agents are given no indication where they should cluster, agreeing this themselves over time.

**Target agreement.** The agents are rewarded for visiting any of a given number of targets, but their reward is proportional to the number of other agents co-located at that target. The agents must therefore coordinate on which single target they will all meet at to maximise their individual rewards.

As well as the standard scenario for these tasks, we conduct robustness tests in two settings, reflecting those elaborated in Appx. E. The first illustrates robustness to learning failures: at every iteration $k$ each learner (whether centralised or decentralised) fails to update its policy (i.e. Line 10 of Alg. 1 is not executed such that $\pi_{k+1}^i = \pi_k^i$) with a 50% probability. The second test illustrates robustness to increases in population size. Instead of having 250 agents throughout, the population begins with 50 agents learning normally, and a further 200 agents are added to the population at the marked point.

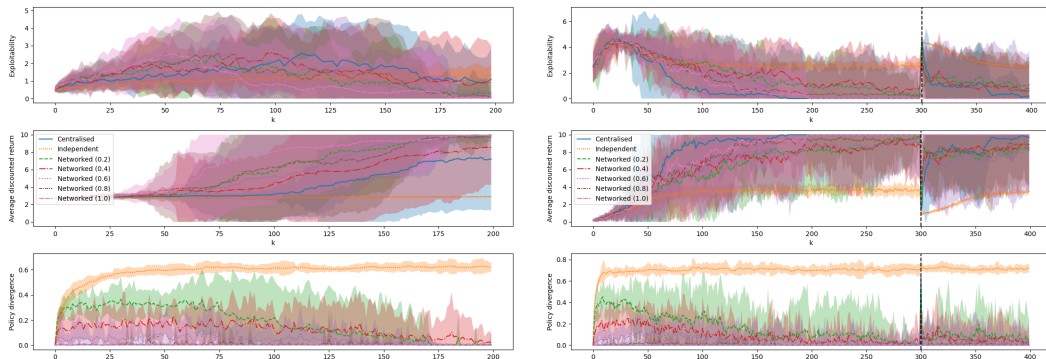

Figure 2: 'Cluster' game, testing robustness to 50% probability of policy update failure. The communication network allows agents that have successfully updated their policies to spread this information to those that have not, providing redundancy. Independent learners cannot do this and hardly appear to learn at all (no increase in return); likewise the centralised architecture is susceptible to its single point of failure. Thus our networked architecture significantly outperforms both the centralised and independent cases.

Figure 3: 'Target agreement' game, testing robustness to a five-times increase in population. The networked architectures are quickly able to spread the learnt policies to the newly arrived agents such that learning progress is minimally disturbed, whereas convergence is significantly impacted in the independent case. The largest broadcast radius (1.0), in particular, appears to suffer no disturbance at all, being much more robust than the centralised case, which takes a significant amount of time to return to equilibrium.

Experiments are evaluated via three metrics (see Appx. F.2 for a full discussion): an approximation of the **exploitability** of the joint policy $\pi_k$; the **average discounted return of the agents' policies** $\pi_k^i$; and the **population's policy divergence**. Hyperparameters are discussed in Appx. F.3.

## 4.1 DISCUSSION

We give here three example figures (**reproduced larger** in Figs. 4, 5 and 6) illustrating the benefits of the networked architecture; in each the decimals refer to each agent's broadcast radius as a fraction of the maximum possible distance in the grid (i.e. the diagonal). See figure captions for details, and Appx. F.4 for further experiments, ablation studies and discussion. For limitations and future work, see Appx. H. As well as allowing convergence within a practical number of iterations, even with only a single communication round, the combination of the buffer and the networked architecture allows us to remove in our experiments a number of the assumptions required for the theoretical algorithms:

- We significantly reduce $M_{pg}$ while still converging within a reasonable $K$. With smaller values for $M_{pg}$ (the number of samples in the buffer) and $L$ (the number of loops through the buffer for updating the Q-function), and hence with worse estimation of the Q-function, the networked architecture outperforms the independent case to an even greater extent. This underlines its advantages in allowing faster convergence in practical settings.

- We can reduce the $M_{td}$ parameter (theoretically required for the learner to wait between collecting samples when learning from a single system run) to 1, effectively removing the innermost loop of the nested learning algorithm (see Line 5 of Alg. 1).

- We can reduce the scaling parameter $\lambda$ of the entropy regulariser to 0, i.e. we converge even without regularisation, allowing us to leave the NE unbiased, and also removing Assumption 3 (Appx. A). In general an unregularised MFG-NE is not unique (Yardim et al., 2023); the ability of the agents to coordinate on one of the multiple solutions in the centralised and networked cases may explain why they outperform the independent-learning case.

- For the PMA operator (Def. 8), we conduct the optimisation over the set $u \in \Delta_{\mathcal{A}}$ instead of $u \in \mathcal{U}_{L_h}$, i.e. we can choose from all possible probability distributions over actions instead of needing to identify the Lipschitz constants given in Assumption 1 (Appx. A).

## 5 BROADER IMPACT / ETHICS STATEMENT

As with many advances in machine learning, and those relating to multi-agent systems in particular, in the long term our research on large populations of coordinating agents could have negative social outcomes if pursued by malicious actors, including surveillance and military uses. However, our work is primarily foundational and far from deployments, and it also has a large range of potential beneficial applications (such as smart grids and disaster response). Moreover, better understanding the dynamics of large multi-agent systems (as we seek to do in this paper) can contribute to ensuring safety by reducing the risks of unintended failures or outcomes.

We hope to help mitigate potential harmful consequences of this research by fostering transparency through submitting our code in the Supplementary Material, which we commit to publishing online under license upon acceptance of the paper.

## 6 REPRODUCIBILITY STATEMENT

The code files to run our experiments are uploaded in the Supplementary Material. We discuss the hyperparameters for our experiments in Table 1 in Appx. F.3. The technical assumptions for our theoretical results are given in Appx. A, and complete proofs are provided in Appx. B.

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

# TECHNICAL APPENDICES

## A   FURTHER DEFINITIONS AND ASSUMPTIONS FOR THEOREMS IN SEC. 3.3

**Assumption 1** (Lipschitz continuity of $P$ and $R$, from Assumption 1, Yardim et al. (2023)). *There exist constants $K_\mu, K_s, K_a, L_\mu, L_s, L_a \in \mathbb{R}_{\geq 0}$ such that $\forall s, s' \in \mathcal{S}, \forall a, a' \in \mathcal{A}, \forall \mu, \mu' \in \Delta_\mathcal{S}$,*

$$||P(\cdot|s, a, \mu) - P(\cdot|s', a', \mu')||_1 \leq K_\mu ||\mu - \mu'||_1 + K_s d(s, s') + K_a d(a, a'),$$

$$|R(s, a, \mu) - R(s', a', \mu')| \leq L_\mu ||\mu - \mu'||_1 + L_s d(s, s') + L_a d(a, a').$$

**Definition 9** (Population update operator, from Def. 3.1, Yardim et al. (2023)). *The single-step population update operator $\Gamma_{pop} : \Delta_\mathcal{S} \times \Pi \to \Delta_\mathcal{S}$ is defined as, $\forall s \in \mathcal{S}$:*

$$\Gamma_{pop}(\mu, \pi)(s) := \sum_{s' \in \mathcal{S}} \sum_{a' \in \mathcal{A}} \mu(s')\pi(a'|s')P(s|s', a', \mu).$$

*Let us use the short hand notation $\Gamma_{pop}^n(\mu, \pi) := \underbrace{\Gamma_{pop}(\dots \Gamma_{pop}(\Gamma_{pop}(\mu, \pi), \pi), \dots, \pi)}_{n \text{ times}}.$*

We recall that $\Gamma_{pop}$ is known to be Lipschitz:

**Lemma 1** (Lipschitz population updates, from Lem. 3.2, Yardim et al. (2023)). *$\Gamma_{pop}$ is Lipschitz with*

$$||\Gamma_{pop}(\mu, \pi) - \Gamma_{pop}(\mu', \pi')||_1 \leq L_{pop,\mu}||\mu - \mu'||_1 + \frac{K_a}{2}||\pi - \pi'||_1,$$

*where $L_{pop,\mu} := \left(\frac{K_s}{2} + \frac{K_a}{2} + K_\mu\right), \forall \pi \in \Pi, \mu \in \Delta_\mathcal{S}$.*

For stationary MFGs the population distribution must be stable with respect to a policy, requiring that $\Gamma_{pop}(\cdot, \pi)$ is contractive $\forall \pi \in \Pi$:

**Assumption 2** (Stable population, from Assumption 2, Yardim et al. (2023)). *Population updates are stable, i.e. $L_{pop,\mu} < 1$.*

**Definition 10** (Stable population operator $\Gamma_{pop}^\infty$, from Def. 3.3, Yardim et al. (2023)). *Given Assumption 2, the operator $\Gamma_{pop}^\infty : \Pi \to \Delta_\mathcal{S}$ maps a given policy to its unique stable population distribution such that $\Gamma_{pop}(\Gamma_{pop}^\infty(\pi), \pi) = \Gamma_{pop}^\infty(\pi)$, i.e. the unique fixed point of $\Gamma_{pop}(\cdot, \pi) : \Delta_\mathcal{S} \to \Delta_\mathcal{S}$.*

**Definition 11** ($Q_h$ and $q_h$ functions). *We define, for any pair $(s, a) \in \mathcal{S} \times \mathcal{A}$:*

$$Q_h(s, a|\pi, \mu) := \mathbb{E}\left[\sum_{t=0}^\infty \gamma^t(R(s_t, a_t, \mu) + h(\pi(s_t))) \middle| \begin{smallmatrix} s_0 = s, & s_{t+1} \sim P(\cdot|s_t, a_t, \mu), \\ a_0 = a, & a_{t+1} \sim \pi(\cdot|s_{t+1}) \end{smallmatrix}, \forall t \geq 0\right]$$

*and*

$$q_h(s, a|\pi, \mu) := R(s, a, \mu) + \gamma \sum_{s', a'} P(s'|s, a, \mu)\pi(a'|s')Q_h(s', a'|\pi, \mu).$$

**Definition 12** ($\Gamma_q$ operator). *The operator $\Gamma_q : \Pi \times \Delta_\mathcal{S} \to \mathcal{Q}$ mapping population-policy pairs to Q-functions is defined as $\Gamma_q(\pi, \mu) := q_h(\cdot, \cdot|\pi, \mu) \in \mathcal{Q} \ \forall \pi \in \Pi, \mu \in \Delta_\mathcal{S}$.*

We also assume that the regulariser $h$ ensures that all actions at all states are explored with non-zero probability:

**Assumption 3** (Persistence of excitation, from Assumption 3, Yardim et al. (2023)). *We assume there exists $p_{inf} > 0$ such that:*

1. *$\pi_{\max}(a|s) \geq p_{inf} \ \forall s \in \mathcal{S}, a \in \mathcal{A}$,*

2. *For any $\pi \in \Pi$ and $q \in \mathcal{Q}$ that satisfy, $\forall (s, a) \in \mathcal{S} \times \mathcal{A}$, $\pi(a|s) \geq p_{inf}$ and $0 \leq q(s, a) \leq Q_{\max}$, it holds that $\Gamma_\eta^{md}(q, \pi)(a|s) \geq p_{inf}, \forall (s, a) \in \mathcal{S} \times \mathcal{A}$.*

**Assumption 4** (Sufficient mixing, from Assumption 4, Yardim et al. (2023)). *For any $\pi \in \Pi$ satisfying $\pi(a|s) \geq p_{inf} > 0 \ \forall s \in \mathcal{S}, a \in \mathcal{A}$, and any initial states $\{s_0^i\}_i \in \mathcal{S}^N$, there exist $T_{mix} > 0, \delta_{mix} > 0$ such that $\mathbb{P}(s_{T_{mix}}^j = s'|\{s_0^i\}_i) \geq \delta_{mix}, \forall s' \in \mathcal{S}, j \in [N]$.*

**Definition 13** (Nested learning operator). *For a learning rate $\eta > 0$, $\Gamma_\eta : \Pi \to \Pi$ is defined as*

$$\Gamma_\eta(\pi) := \Gamma_\eta^{md}(\Gamma_q(\pi, \Gamma_{pop}^\infty(\pi)), \pi).$$

**Lemma 2** (Lipschitz continuity of $\Gamma_\eta$, from Lem. 3.7, (Yardim et al., 2023))). *For any $\eta > 0$, the operator $\Gamma_\eta : \Pi \to \Pi$ is Lipschitz with constant $L_{\Gamma_\eta}$ on $(\Pi, ||\cdot||_1)$.*

# B   FULL THEOREMS AND COMPLETE PROOFS

## B.1   SAMPLE GUARANTEES OF INDEPENDENT-LEARNING CASE

**Lemma 3** (Independent learning, from Thm. 4.5, Yardim et al. (2023)). *Define $t_0 := \frac{16(1+\gamma)^2}{((1-\gamma)\delta_{mix}p_{inf})^2}$. Assume that Assumptions 1, 2, 3 and 4 hold, that $\eta > 0$ satisfies $L_{\Gamma_\eta} < 1$, and that $\pi^*$ is the unique MFG-NE. The learning rates are $\beta_m = \frac{2}{(1-\gamma)(t_0+m-1)}$ $\forall m \geq 0$, and let $\varepsilon > 0$ be arbitrary. There exists a problem-dependent constant $a \in [0, \infty)$ such that if $K = \frac{\log 8\varepsilon^{-1}}{\log L_{\Gamma_\eta}^{-1}}$, $M_{pg} > \mathcal{O}(\varepsilon^{-2-a})$ and $M_{td} > \mathcal{O}(log^2\varepsilon^{-1})$, then the random output $\{\pi_K^i\}_i$ of Alg. 1 when run with $C = 0$ (such that there is no communication) satisfies for all agents $i = 1, \ldots, N$,*

$$\mathbb{E}\left[||\pi_K^i - \pi^*||_1\right] \leq \varepsilon + \mathcal{O}\left(\frac{1}{\sqrt{N}}\right).$$

## B.2   FULL VERSION OF THM. 1

**Theorem 1** (Networked learning with random adoption). *For $p_{inf}$ and $\delta_{mix}$ defined in Assumptions 3 and 4 respectively, define $t_0 := \frac{16(1+\gamma)^2}{((1-\gamma)\delta_{mix}p_{inf})^2}$. Assume that Assumptions 1, 2, 3 and 4 hold, and that $\pi^*$ is the unique MFG-NE policy. For $L_{\Gamma_\eta}$ defined in Lem. 2, we assume $\eta > 0$ satisfies $L_{\Gamma_\eta} < 1$. The learning rates are $\beta_m = \frac{2}{(1-\gamma)(t_0+m-1)}$ $\forall m \geq 0$, and let $\varepsilon > 0$ be arbitrary. Assume also that $C > 0$, with $\tau_k \to \infty$. There exists a problem-dependent constant $a \in [0, \infty)$ such that if $K = \frac{\log 8\varepsilon^{-1}}{\log L_{\Gamma_\eta}^{-1}}$, $M_{pg} > \mathcal{O}(\varepsilon^{-2-a})$ and $M_{td} > \mathcal{O}(log^2\varepsilon^{-1})$, then the random output $\{\pi_K^i\}_i$ of Alg. 1 preserves the sample guarantees of the independent-learning case given in Lem. 3, i.e. the output satisfies, for all agents $i = 1, \ldots, N$,*

$$\mathbb{E}\left[||\pi_K^i - \pi^*||_1\right] \leq \varepsilon + \mathcal{O}\left(\frac{1}{\sqrt{N}}\right).$$

*(Proof in Appx. B.3.)*

## B.3   PROOF OF THM. 1

*Proof.* If $\tau_k \to \infty$, the softmax function that defines the probability of a received policy being adopted in Line 15 of Alg. 1 gives a uniform distribution. Policies are thus exchanged at random between communicating agents an arbitrary $C > 0$ times, which does not affect the random output of the algorithm, such that the random output satisfies the same expectation as if $C = 0$. □

## B.4   CONDITIONAL TD LEARNING FROM A SINGLE CONTINUOUS RUN OF THE EMPIRICAL DISTRIBUTION OF $N$ AGENTS

**Lemma 4** (Conditional TD learning from a single continuous run of the empirical distribution of $N$ agents, from Thm. 4.2, Yardim et al. (2023)). *Define $t_0 := \frac{16(1+\gamma)^2}{((1-\gamma)\delta_{mix}p_{inf})^2}$. Assume Assumption 4 holds and let policies $\{\pi^i\}_i$ be given such that $\pi^i(a|s) \geq p_{inf}$ $\forall i$. Assume Lines 3-9 of Alg. 1 are run with policies $\{\pi^i\}_i$, arbitrary initial agents states $\{s_0^i\}_i$, learning rates $\beta_m = \frac{2}{(1-\gamma)(t_0+m-1)}$, $\forall m \geq 0$ and $M_{pg} > \mathcal{O}(\varepsilon^{-2})$, $M_{td} > \mathcal{O}(\log \varepsilon^{-1})$. If $\bar{\pi} \in \Pi$ is an arbitrary policy, $\Delta := \sum_{i=1}^N ||\pi^i - \bar{\pi}||_1$ and $Q^* := Q_h(\cdot, \cdot|\bar{\pi}, \mu_{\bar{\pi}})$, then the random output $\hat{Q}_{M_{pg}}^i$ of Lines 3-9 satisfies*

$$\mathbb{E}\left[||\hat{Q}_{M_{pg}}^i - Q^*||_\infty\right] \leq \varepsilon + \mathcal{O}\left(\frac{1}{\sqrt{N}} + \frac{1}{N}\Delta + ||\pi^i - \bar{\pi}||_1\right).$$

## B.5 FULL VERSION OF THM. 2

**Theorem 2** (Networked learning with non-random adoption). *Assume that Assumptions 1, 2, 3 and 4 hold, and that Alg. 1 is run with learning rates and constants as defined in Thm. 1, except now $\tau_k \in \mathbb{R}_{>0}$. Assume that $\sigma_{k+1}^i$ is generated uniquely for each $i$, in a manner independent of any metric related to $\pi_{k+1}^i$, e.g. $\sigma_{k+1}^i$ is random or related only to the index $i$ (so as not to bias the spread of any particular policy). Let the random output of this Algorithm be denoted as $\{\pi_K^{i,net}\}_i$. Also consider an independent-learning version of the algorithm (i.e. with the same parameters except $C = 0$) and denote its random output $\{\pi_K^{i,ind}\}_i$; and a centralised version of the algorithm with the same parameters (see Rem. 3) and denote its random output as $\pi_K^{cent}$. Then for all agents $i = 1, \ldots, N$, the random outputs $\{\pi_K^{i,net}\}_i$, $\{\pi_K^{i,ind}\}_i$ and $\pi_K^{cent}$ satisfy the following relations, where $ub_{net}$, $ub_{ind}$ and $ub_{cent}$ are respective upper bounds for each case:*

$$\mathbb{E}\left[||\pi_K^{cent} - \pi^*||_1\right] \leq ub_{cent}, \quad \mathbb{E}\left[||\pi_K^{i,net} - \pi^*||_1\right] \leq ub_{net}, \quad \mathbb{E}\left[||\pi_K^{i,ind} - \pi^*||_1\right] \leq ub_{ind},$$

$$where \quad ub_{cent} \leq ub_{net} \leq ub_{ind} = \varepsilon + \mathcal{O}\left(\frac{1}{\sqrt{N}}\right).$$

*(Proof in Appx. B.6.)*

## B.6 PROOF OF THM. 2

*Proof.* We build off the proof of our Lem. 3, given in Thm. D.9 of Yardim et al. (2023), where the sample guarantees of the independent case are worse than those of the centralised algorithm as a result of the divergence between the decentralised policies due to the stochasticity of the PMA updates. For an arbitrary policy $\bar{\pi}_k \in \Pi$, for all $k = 0, 1, \ldots, K$ define the policy divergence as the random variable $\Delta_k := \sum_{i=1}^{N} ||\pi_k^i - \bar{\pi}_k||_1$. We can say that $\Delta_{k,cent} = 0 \ \forall k$ is the divergence in the centralised case, while in the networked case the policy divergence is $\Delta_{k+1,c}$ after communication round $c \in 1, \ldots, C$. The independent case is equivalent to the scenario when $C = 0$, such that its policy divergence can be written $\Delta_{k+1,0}$.

For $\tau_k \in \mathbb{R}_{>0}$, the adoption probability $\Pr\left(\text{adopted}^i = \sigma_{k+1}^j\right) = \frac{\exp\left(\sigma_{k+1}^j/\tau_k\right)}{\sum_{x=1}^{[J_t^i]} \exp\left(\sigma_{k+1}^x/\tau_k\right)}$ (as in Line 15 of Alg. 1) is higher for some $j \in J_t^i$ than for others. This means that for $c > 0$ for which there are communication links in the population, in expectation the number of unique policies in the population will decrease, as it will likely become that $\pi_{k+1}^i = \pi_{k+1}^j$ for some $i, j \in \{1, \ldots, N\}$. As such, $\Delta_{k+1,cent} \leq \mathbb{E}[\Delta_{k+1,C}] \leq \mathbb{E}[\Delta_{k+1,0}]$, i.e. the policy divergence in the independent-learning case is expected to be greater than or equal to that of the networked case.

The proof of Lem. 3 given in Thm. D.9 of Yardim et al. (2023) ends with, for constants $\chi$ and $\xi$,

$$\mathbb{E}\left[||\pi_K^i - \pi^*||_1\right] \leq 2L_{\Gamma_\eta}^K + \frac{\chi}{1 - L_{\Gamma_\eta}} + \xi \sum_{k=1}^{K-1} L_{\Gamma_\eta}^{K-k-1} \mathbb{E}[\Delta_k],$$

where in our context the policy divergence in the independent case $\mathbb{E}[\Delta_{k+1}]$ is equivalent to $\mathbb{E}[\Delta_{k+1,C}]$ when $C = 0$, i.e. $\mathbb{E}[\Delta_{k+1,0}]$.

Thus, for all agents $i = 1, \ldots, N$, the random outputs $\{\pi_K^{i,net}\}_i$, $\{\pi_K^{i,ind}\}_i$ and $\pi_K^{cent}$ satisfy:

$$\mathbb{E}\left[||\pi_K^{i,ind} - \pi^*||_1\right] \leq ub_{ind} = 2L_{\Gamma_\eta}^K + \frac{\chi}{1 - L_{\Gamma_\eta}} + \xi \sum_{k=1}^{K-1} L_{\Gamma_\eta}^{K-k-1} \mathbb{E}[\Delta_{k,0}],$$

$$\mathbb{E}\left[||\pi_K^{i,net} - \pi^*||_1\right] \leq ub_{net} = 2L_{\Gamma_\eta}^K + \frac{\chi}{1 - L_{\Gamma_\eta}} + \xi \sum_{k=1}^{K-1} L_{\Gamma_\eta}^{K-k-1} \mathbb{E}[\Delta_{k,C}],$$

$$\mathbb{E}\left[||\pi_K^{cent} - \pi^*||_1\right] \leq ub_{cent} = 2L_{\Gamma_\eta}^K + \frac{\chi}{1 - L_{\Gamma_\eta}} + \xi \sum_{k=1}^{K-1} L_{\Gamma_\eta}^{K-k-1} \mathbb{E}[\Delta_{k,cent}].$$

Since $\Delta_{k+1,cent} \leq \mathbb{E}\left[\Delta_{k+1,C}\right] \leq \mathbb{E}\left[\Delta_{k+1,0}\right]$, we obtain our result, i.e.

$$ub_{cent} \; \leq \; ub_{net} \leq \; ub_{ind} \; = \; \varepsilon + \mathcal{O}\left(\frac{1}{\sqrt{N}}\right).$$

$\square$

**Remark 5.** *It may help to see that our result is a consequence of the following. Denote $\hat{Q}_{M_{pg}}^{i,net}$, $\hat{Q}_{M_{pg}}^{i,ind}$ and $\hat{Q}_{M_{pg}}^{cent}$ as the random outputs of Lines 3-9 of Alg. 1 in the networked, independent and centralised cases respectively. In Lem. 4, we can see that policy divergence gives bias terms in the estimation of the Q-value. Therefore, given $\Delta_{k+1,cent} \leq \mathbb{E}\left[\Delta_{k+1,C}\right] \leq \mathbb{E}\left[\Delta_{k+1,0}\right]$, we can also say*

$$\mathbb{E}\left[||\hat{Q}_{M_{pg}}^{cent} - Q^*||_\infty\right] \; \leq \; \mathbb{E}\left[||\hat{Q}_{M_{pg}}^{i,net} - Q^*||_\infty\right] \; \leq \; \mathbb{E}\left[||\hat{Q}_{M_{pg}}^{i,ind} - Q^*||_\infty\right].$$

*In other words, the networked case will require the same or fewer outer iterations $K$ to reduce the variance caused by this bias than the independent case requires (where the bias is non-vanishing), and the same or more iterations than the centralised case requires.*

## B.7 PROOF OF THM. 3

*Proof.* From the proof of Thm. 2 in Appx. B.6 we have:

$$\mathbb{E}\left[||\pi_K^{i,ind} - \pi^*||_1\right] \leq ub_{ind} = 2L_{\Gamma_\eta}^K + \frac{\chi}{1 - L_{\Gamma_\eta}} + \xi \sum_{k=1}^{K-1} L_{\Gamma_\eta}^{K-k-1}\mathbb{E}\left[\Delta_{k,0}\right],$$

$$\mathbb{E}\left[||\pi_K^{i,net} - \pi^*||_1\right] \leq ub_{net} = 2L_{\Gamma_\eta}^K + \frac{\chi}{1 - L_{\Gamma_\eta}} + \xi \sum_{k=1}^{K-1} L_{\Gamma_\eta}^{K-k-1}\mathbb{E}\left[\Delta_{k,C}\right],$$

$$\mathbb{E}\left[||\pi_K^{cent} - \pi^*||_1\right] \leq ub_{cent} = 2L_{\Gamma_\eta}^K + \frac{\chi}{1 - L_{\Gamma_\eta}} + \xi \sum_{k=1}^{K-1} L_{\Gamma_\eta}^{K-k-1}\mathbb{E}\left[\Delta_{k,cent}\right].$$

With a static, connected network and $\tau_k \to 0 \; \forall k$, max-consensus is always reached after $C = d_{\mathcal{G}}$ communication rounds, such that $\Delta_{k,cent} = \Delta_{k,d_{\mathcal{G}}} = 0$ (Nejad et al., 2009). The convergence rate of the max-consensus algorithm is $\frac{1}{d_{\mathcal{G}}}$ (Nejad et al., 2009), i.e. there is a decrease in the number of policies in the population by a factor of approximately $\frac{1}{d_{\mathcal{G}}}$ with each communication round up to $C = d_{\mathcal{G}}$, and therefore there is also a decrease in the policy divergence $\mathbb{E}\left[\Delta_{k,c}\right]$ by a factor of approximately $\frac{1}{d_{\mathcal{G}}}$ with each communication round. Thus

$$\mathbb{E}\left[\Delta_{k,c+1}\right] \approx \mathbb{E}\left[\Delta_{k,c}\right] - \left(\mathbb{E}\left[\Delta_{k,c}\right] \times \frac{1}{d_{\mathcal{G}}}\right), \text{ simplifying to}$$

$$\mathbb{E}\left[\Delta_{k,c+1}\right] \approx \mathbb{E}\left[\Delta_{k,c}\right] \times \left(1 - \frac{1}{d_{\mathcal{G}}}\right).$$

By induction

$$\mathbb{E}\left[\Delta_{k,C}\right] \approx \mathbb{E}\left[\Delta_{k,0}\right] \times \left(\left(1 - \frac{1}{d_{\mathcal{G}}}\right)^C\right),$$

however, we know that $\Delta_{k,d_{\mathcal{G}}} = 0$, so we can more accurately use the piecewise function $f(C, d_{\mathcal{G}})$, defined as:

$$f(C, d_{\mathcal{G}}) = \begin{cases} \left(\left(1 - \frac{1}{d_{\mathcal{G}}}\right)^C\right) & \text{if } C < d_{\mathcal{G}}, \\ 0 & \text{if } C \geq d_{\mathcal{G}} \end{cases},$$

giving

$$\mathbb{E}\left[\Delta_{k,C}\right] \approx \mathbb{E}\left[\Delta_{k,0}\right] \times f(C, d_{\mathcal{G}}).$$

We can therefore also say:

$$ub_{ind} = 2L_{\Gamma_\eta}^K + \frac{\chi}{1 - L_{\Gamma_\eta}} + \xi \sum_{k=1}^{K-1} L_{\Gamma_\eta}^{K-k-1}\mathbb{E}\left[\Delta_{k,0}\right],$$

$$ub_{net} \approx 2L_{\Gamma_\eta}^K + \frac{\chi}{1 - L_{\Gamma_\eta}} + \xi \sum_{k=1}^{K-1} L_{\Gamma_\eta}^{K-k-1} \mathbb{E}\left[\Delta_{k,0}\right] \times f(C, d_{\mathcal{G}}),$$

$$ub_{cent} = 2L_{\Gamma_\eta}^K + \frac{\chi}{1 - L_{\Gamma_\eta}}.$$

We therefore firstly have

$$ub_{ind} - ub_{net} \approx \xi \sum_{k=1}^{K-1} L_{\Gamma_\eta}^{K-k-1} \mathbb{E}\left[\Delta_{k,0}\right] - \xi \sum_{k=1}^{K-1} L_{\Gamma_\eta}^{K-k-1} \mathbb{E}\left[\Delta_{k,0}\right] \times f(C, d_{\mathcal{G}}),$$

which simplifies to

$$ub_{ind} - ub_{net} \approx \xi \sum_{k=1}^{K-1} L_{\Gamma_\eta}^{K-k-1} \mathbb{E}\left[\Delta_{k,0}\right] \times \left(1 - f(C, d_{\mathcal{G}})\right).$$

This gives us one of the results, where we focus on the functional dependence on $C$ and $d_{\mathcal{G}}$ by using the tight bound big Theta ($\Theta$):

$$ub_{net} \approx ub_{ind} - \Theta\left(1 - f(C, d_{\mathcal{G}})\right).$$

Secondly, we have

$$ub_{net} \approx ub_{cent} + \xi \sum_{k=1}^{K-1} L_{\Gamma_\eta}^{K-k-1} \mathbb{E}\left[\Delta_{k,0}\right] \times f(C, d_{\mathcal{G}}),$$

giving us the second result

$$ub_{net} \approx ub_{cent} + \Theta\left(f(C, d_{\mathcal{G}})\right).$$

$\square$

**Remark 6.** *If it is always $\sigma_{k+1}^1$ and $\pi_{k+1}^1$ that is adopted by the whole population (i.e. $i = 1$), then this is exactly the same as the centralised case. If the $\sigma_{k+1}^j$ and $\pi_{k+1}^j$ that gets adopted has different $j$ for each $k$ then this is akin to a version of the centralised setting where the index of the central learning agent may differ for each $k$.*

**Remark 7.** *Thm. 3 assumes $\tau_k \to 0 \; \forall k$. If we assume instead $\tau_k \in \mathbb{R}_{>0}$, then we have $ub_{net} \to ub_{ind}$ as $C \to 0$, and $ub_{net} \to ub_{cent}$ as $C \to \infty$. This is because the spread of policies is now probabilistic rather than deterministic, and depends on the interplay of $\tau_k$ with how large are the differences in the received values of $\sigma_{k+1}^j$. Therefore consensus (and hence reduction in divergence between policies) is reached only asymptotically. This applies to both static, connected networks and to repeatedly jointly connected ones, assuming the latter becomes jointly connected infinitely often.*

## B.8 POLICY-UPDATE STABILITY GUARANTEE

**Theorem 4** (Policy-update stability guarantee). *Let Alg. 1 run as per Thm. 1 or Thm. 2, and say that $\varepsilon_k$ is the error term at iteration $k = \frac{\log 8\varepsilon_k^{-1}}{\log L_{\Gamma_\eta}^{-1}}$. For all agents $i$, the maximum possible distance between $\pi_k^{i,net}$ and $\pi_{k+1}^{i,net}$ is given by $\mathbb{E}\left[||\pi_k^{i,net} - \pi_{k+1}^{i,net}||_1\right] \leq \varepsilon_k + \varepsilon_{k+1} + \mathcal{O}\left(\frac{1}{\sqrt{N}}\right)$. This bound provides policy-update stability guarantees during the learning process; moreover the bound shrinks with each successive $k$ since $\varepsilon_k$ decreases with $k$. Equivalent analysis can also be conducted for both the centralised and independent cases.*

*Proof.* Thms. 1 and 2 bound the difference between each agent's current policy $\pi_k^i$ and the unique equilibrium policy $\pi^*$, with the difference depending on the bias term $\varepsilon_k$ that relates to the iteration $k$ as indicated. Policies $\pi_k^i$ and $\pi_{k+1}^i$ fall within balls centred on $\pi^*$ with radii of $\varepsilon_k + \mathcal{O}\left(\frac{1}{\sqrt{N}}\right)$ and $\varepsilon_{k+1} + \mathcal{O}\left(\frac{1}{\sqrt{N}}\right)$ respectively. This means that the maximum possible distance between $\pi_k^i$ and $\pi_{k+1}^i$ is the sum of these radii, i.e. $\mathbb{E}\left[||\pi_k^i - \pi_{k+1}^i||_1\right] \leq \varepsilon_k + \varepsilon_{k+1} + \mathcal{O}\left(\frac{1}{\sqrt{N}}\right)$, giving the result. $\square$

### B.9    REMARK ON THEORETICAL HYPERPARAMETERS WHEN USED IN PRACTICAL SETTINGS

**Remark 8.** *The theoretical analysis in Sec. 3.3 and Appx. B requires algorithmic hyperparameters (Thms. 1 and 2) that render convergence impractically slow in all of the centralised, independent and networked cases. (Indeed Yardim et al. (2023) do not provide empirical demonstrations of their algorithms for the centralised and independent cases.) In particular, the values of $\delta_{mix}$ and $p_{inf}$ give rise to very large $t_0$, causing very small learning rates $\{\beta_m\}_{m \in \{0,...,M_{pg}-1\}}$, and necessitating very large values for $M_{td}$ and $M_{pg}$.*

## C    EXPLANATION OF NETWORKED AGENTS OUTPERFORMING CENTRALISED AGENTS IN PRACTICAL SETTINGS

After the PMA step in Line 17 of Alg. 2, we have randomly updated policies $\{\pi_{k+1}^i\}_i$, $i = 1, ..., N$, where the randomness stems both from each agent's independent collection of samples and from the random sampling of each one's buffer when updating the Q-functions $\hat{Q}_{M_{pg}}^i$. The policies' associated finitely approximated returns are $\{\sigma_{k+1}^i\}_i$.

Let us assume for simplicity that in the networked case the population shares a single policy after the $C$ communication rounds, i.e. that this has been enabled by the connectivity and diameter of the communication network $\mathcal{G}_t$ and the values of $C$ and $\tau_k \in \mathbb{R}_{>0}$, as per Rems. 4 and 7. Call this network consensus policy $\pi_{k+1}^{\text{net}}$, and its associated finitely approximated return $\sigma_{k+1}^{\text{net}}$. Recall that the centralised case, where the updated policy of arbitrary agent $i = 1$ is automatically pushed to all the others, is equivalent to a networked case where policy consensus is reached on a random one of the policies $\{\pi_k^i\}_i$, $i = 1, ..., N$; call this policy arbitrarily given to the whole population $\pi_{k+1}^{\text{cent}}$, and its associated finitely approximated return $\sigma_{k+1}^{\text{cent}}$.

Since $\pi_{k+1}^{\text{cent}}$ is chosen at random regardless of its quality, in expectation $\sigma_{k+1}^{\text{cent}}$ will be the mean value of $\{\sigma_{k+1}^i\}_i$ for each $k$, though there will be high variance. Conversely, the softmax adoption probability (Line 27 of Alg. 2) for the networked case is such that in expectation the $\pi_{k+1}^{\text{net}}$ that gets adopted by the whole networked population will have higher than average $\sigma_{k+1}^{\text{net}}$ (indeed if $\tau_k \to 0$ it will have the highest $\sigma_{k+1}^{\text{net}}$ for each $k$). That is, the probability distribution is weighted by their relative estimated performance. As such, $\mathbb{E}[\sigma_{k+1}^{\text{net}}] > \mathbb{E}[\sigma_{k+1}^{\text{cent}}]$. There is also less variance in $\sigma_{k+1}^{\text{net}}$ than $\sigma_{k+1}^{\text{cent}}$, as the former is biased towards higher values.

If the networked case results in higher average $\sigma_{k+1}$ being adopted than the centralised case, then the policies of which $\sigma_{k+1}$ gives an approximated return are also biased towards being better performing, and with less variance in quality. Thus the networked agents can improve their return and converge faster than the centralised agents, by choosing updates in a more principled manner. This intuition applies even if we loosen the assumption that the networked population converges on a single consensus policy within the $C$ communication rounds. (The same logic can of course also be applied to understand why networked agents outperform entirely independent ones.) It is significant that the communication scheme not only allows us to avoid the undesirable assumption of a centralised learner, but even to outperform it.

## D    ALGORITHM ACCELERATION BY USE OF EXPERIENCE-REPLAY BUFFER (FURTHER DETAILS)

The intuition behind the better learning efficiency resulting from our experience replay buffer in Alg. 2 is as follows. The value of a state-action pair $p$ is dependent on the values of subsequent states reached, but the value of $p$ is only updated when the TD update is conducted on $p$, rather than every time a subsequent pair is updated. By learning from each stored transition multiple times, we not only make repeated use of the reward and transition information in each costly experience, but also repeatedly update each state-action pair in light of its likewise updated subsequent states.

We leave $\beta_m$ fixed across all iterations, as we found empirically that this yields sufficient learning. We have not experimented with decreasing $\beta$ as $l$ increases, though this may benefit learning.

The transitions in the buffer are discarded after the replay cycles and a new buffer is initialised for the next iteration $k$, as in Line 4. As such the space complexity of the buffer only grows linearly with the number of $M_{pg}$ iterations within each outer loop $k$, rather than with the number of $K$ loops.

---

**Algorithm 2** Networked learning with experience replay

---

**Require:** loop parameters $K, M_{pg}, M_{td}, C, L, E$, learning parameters $\eta, \beta, \lambda, \gamma, \{\tau_k\}_{k \in \{0,\ldots,K-1\}}$
**Require:** initial states $\{s_0^i\}_i, i = 1, \ldots, N$
1: Set $\pi_0^i = \pi_{\max}, \forall i$ and $t \leftarrow 0$
2: **for** $k = 0, \ldots, K-1$ **do**
3:      $\forall s, a, i : \hat{Q}_0^i(s, a) = Q_{\max}$
4:      $\forall i$: Empty $i$'s buffer
5:      **for** $m = 0, \ldots, M_{pg} - 1$ **do**
6:          **for** $M_{td}$ iterations **do**
7:              Take step $\forall i : a_t^i \sim \pi_k^i(\cdot|s_t^i), r_t^i = R(s_t^i, a_t^i, \hat{\mu}_t), s_{t+1}^i \sim P(\cdot|s_t^i, a_t^i, \hat{\mu}_t); t \leftarrow t + 1$
8:          **end for**
9:          $\forall i$: Add $\zeta_{t-2}^i$ to $i$'s buffer
10:      **end for**
11:      **for** $l = 0, \ldots, L-1$ **do**
12:          $\forall i$ : Shuffle buffer
13:          **for** transition $\zeta_b^i$ in $i$'s buffer **do** $(\forall i)$
14:              Compute TD update $(\forall i)$: $\hat{Q}_{m+1}^i = \tilde{F}_\beta^{\pi_k^i}(\hat{Q}_m^i, \zeta_{t-2}^i)$ (see Def. 7)
15:          **end for**
16:      **end for**
17:      PMA step $\forall i : \pi_{k+1}^i = \Gamma_\eta^{md}(\hat{Q}_{M_{pg}}^i, \pi_k^i)$ (see Def. 8)
18:      $\forall i : \sigma_{k+1}^i = 0$
19:      **for** $e = 0, \ldots, E-1$ evaluation steps **do**
20:          Take step $\forall i : a_t^i \sim \pi_{k+1}^i(\cdot|s_t^i), r_t^i = R(s_t^i, a_t^i, \hat{\mu}_t), s_{t+1}^i \sim P(\cdot|s_t^i, a_t^i, \hat{\mu}_t)$
21:          $\forall i : \sigma_{k+1}^i = \sigma_{k+1}^i + \gamma^e(r_t^i + h(\pi_{k+1}^i(s_t^i)))$
22:          $t \leftarrow t + 1$
23:      **end for**
24:      **for** $C$ rounds **do**
25:          $\forall i$ : Broadcast $\sigma_{k+1}^i, \pi_{k+1}^i$
26:          $\forall i : J_t^i = i \cup \{j \in \mathcal{N} : (i, j) \in \mathcal{E}_t\}$
27:          $\forall i$ : Select $\text{adopted}^i \sim \Pr\big(\text{adopted}^i = j\big) = \frac{\exp(\sigma_{k+1}^j/\tau_k)}{\sum_{x \in J_t^i} \exp(\sigma_{k+1}^x/\tau_k)} \; \forall j \in J_t^i$
28:          $\forall i : \sigma_{k+1}^i \leftarrow \sigma_{k+1}^{\text{adopted}^i}, \pi_{k+1}^i \leftarrow \pi_{k+1}^{\text{adopted}^i}$
29:          Take step $\forall i : a_t^i \sim \pi_{k+1}^i(\cdot|s_t^i), r_t^i = R(s_t^i, a_t^i, \hat{\mu}_t), s_{t+1}^i \sim P(\cdot|s_t^i, a_t^i, \hat{\mu}_t); t \leftarrow t + 1$
30:      **end for**
31: **end for**
32: Return policies $\{\pi_K^i\}_i, i = 1, \ldots, N$

---

# E EXTENDED DISCUSSION ON ROBUSTNESS OF COMMUNICATION NETWORKS IN MFGS AND RELATED EXPERIMENTAL SETTINGS

We consider two scenarios to which we desire real-world many-agent systems (e.g. robotic swarms or autonomous vehicle traffic) to be robust; these scenarios form the basis of our experiments on robustness (see Sec. 4 and Figs. 2, 3, 8 and 9). The networked setup affords population **fault-tolerance** and **online scalability**, which are motivating qualities of many-agent systems.

Firstly, we consider a scenario in which the learning/updating procedure of agents fails with a certain probability within each iteration, in which cases $\pi_{k+1}^i = \pi_k^i$ (see Figs. 2 and 8 for our experimental results and discussion of this scenario). In real-life decentralised settings, this might be particularly liable to occur since the updating process might only be synchronised between agents by internal clock ticks, such that some agents may not complete their update in the allotted time but will nevertheless be required to take the next step in the environment. Such failures slow the

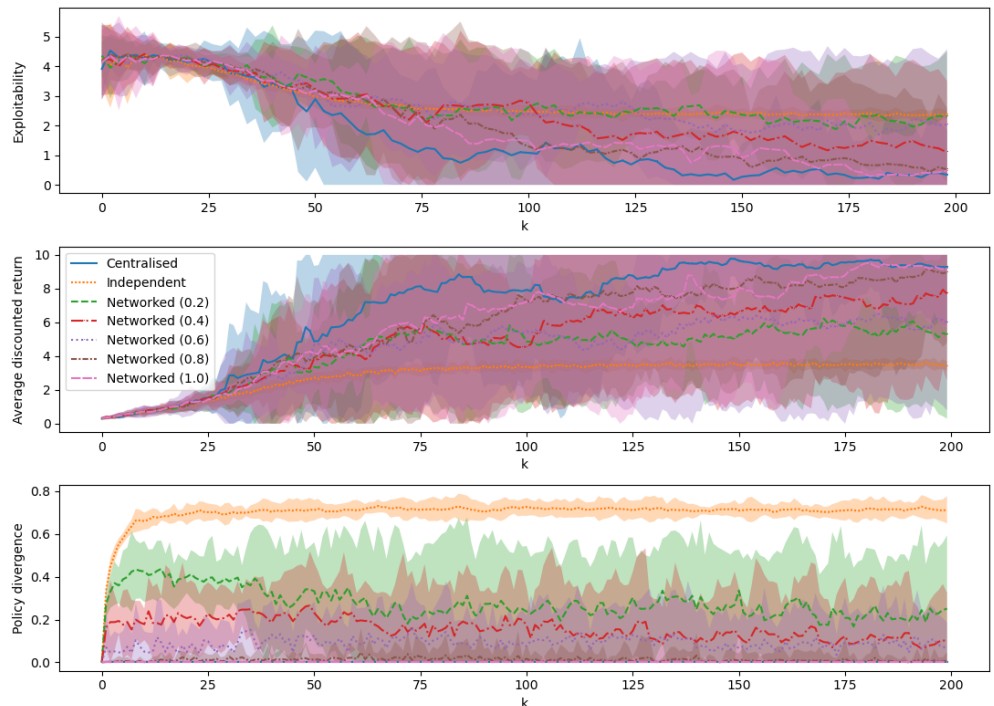

Figure 4: Larger version of Fig. 1. 'Target agreement' game. Even with only a single communication round, our networked case outperforms the independent case wrt. exploitability, and markedly outperforms wrt. return. The fact that the lowest broadcast radius (0.2) ends with similar exploitability to the independent case yet much higher return shows our networked algorithm can help agents find 'preferable' equilibria.

improvement of the population in the independent case, and in the centralised setting it means no improvement occurs at all in any iteration in which failure occurs, as there is a single point of failure. Networked communication instead provides redundancy in case of failures, with the updated policies of any agents that have managed to learn spreading through the population to those that have not. This feature thus ensures that improvement can continue for potentially the whole population even if a high number of agents do not manage to learn at a given iteration.

Secondly, we may want to arbitrarily increase the size of a population of agents that are already learning or operating in the environment (we can imagine extra fleets of autonomous cars or drones being deployed) - see Appx. G for comparison with other works considering this type of robustness (Eck et al., 2023; Gao et al., 2024; Dawood et al., 2023; Wu et al., 2024b). A purely independent setting would require all the new agents to learn a policy individually given the existing distribution, and the process of their following and improving policies from scratch may itself disturb the NE that has already been achieved by the original population. With a communication network, however, the policies that have been learnt so far can quickly be shared with the new agents in a decentralised way, hopefully before their unoptimised policies can destabilise the current NE. This would provide, for example, a way to bootstrap a large population from a smaller pre-trained group, if training were considered expensive in a given setting. See Figs. 3 and 9 for our experimental results and discussion of this scenario.

## F  EXPERIMENTS

Experiments were conducted on a MacBook Pro, Apple M1 Max chip, 32 GB, 10 cores. We use `scipy.optimize.minimize` (employing Sequential Least Squares Programming) to conduct the optimisation step in Def. 8, and the JAX framework to accelerate and vectorise some elements of our code.

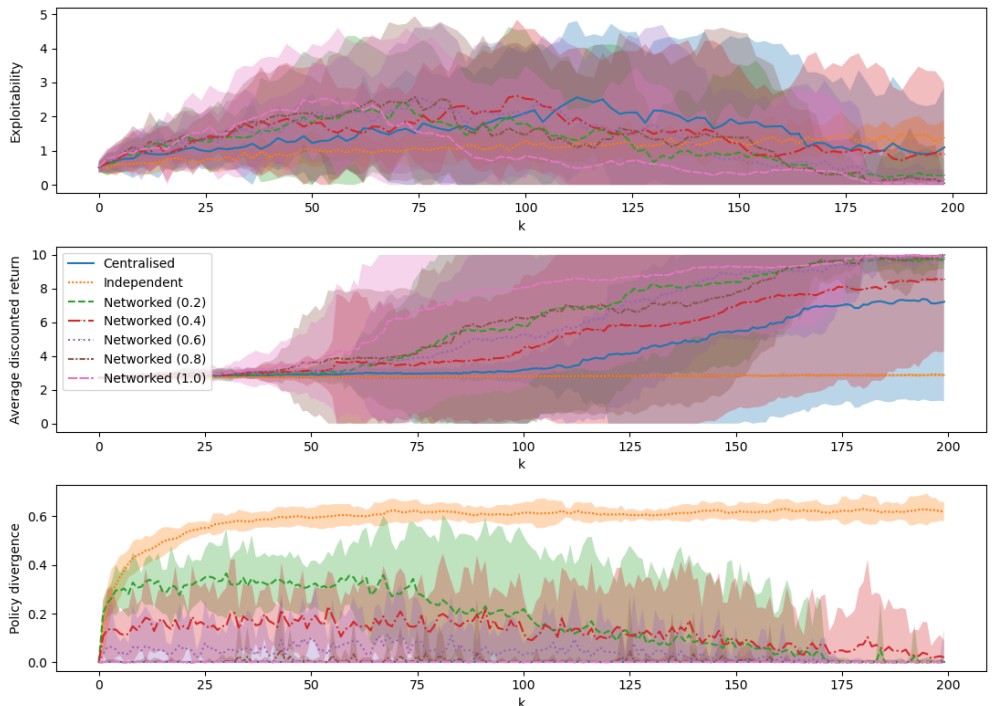

Figure 5: Larger version of Fig. 2. 'Cluster' game, testing robustness to 50% probability of policy update failure. The communication network allows agents that have successfully updated their policies to spread this information to those that have not, providing redundancy. Independent learners cannot do this and hardly appear to learn at all (no increase in return); likewise the centralised architecture is susceptible to its single point of failure. Thus our networked architecture significantly outperforms both the centralised and independent cases.

For reproducibility, the code to run our experiments is provided with our Supplementary Material, and will be made publicly available upon publication.

### F.1 GAMES

We conduct numerical tests with two games (defined by the agents' objectives), chosen for being particularly amenable to intuitive and visualisable understanding of whether the agents are learning behaviours that are appropriate and explainable for the respective objective functions. In all cases, rewards are normalised in [0,1] after they are computed.

**Cluster.** This is the inverse of the 'exploration' game in (Lauriere et al., 2022), where in our case agents are encouraged to gather together by the reward function $R(s_t^i, a_t^i, \hat{\mu}_t) = \log(\hat{\mu}_t(s_t^i))$. That is, agent $i$ receives a reward that is logarithmically proportional to the fraction of the population that is co-located with it at time $t$. We give the population no indication where they should cluster, agreeing this themselves over time.

**Agree on a single target.** Unlike in the above 'cluster' game, the agents are given options of locations at which to gather, and they must reach consensus among themselves. If the agents are co-located with one of a number of specified targets $\phi \in \Phi$ (in our experiments we place one target in each of the four corners of the grid), and other agents are also at that target, they get a reward proportional to the fraction of the population found there; otherwise they receive a penalty of -1. In other words, the agents must coordinate on which of a number of mutually beneficial points will be their single gathering place. The reward function is given by $R(s_t^i, a_t^i, \hat{\mu}_t) = r_{targ}(r_{collab}(\hat{\mu}_t(s_t^i)))$,

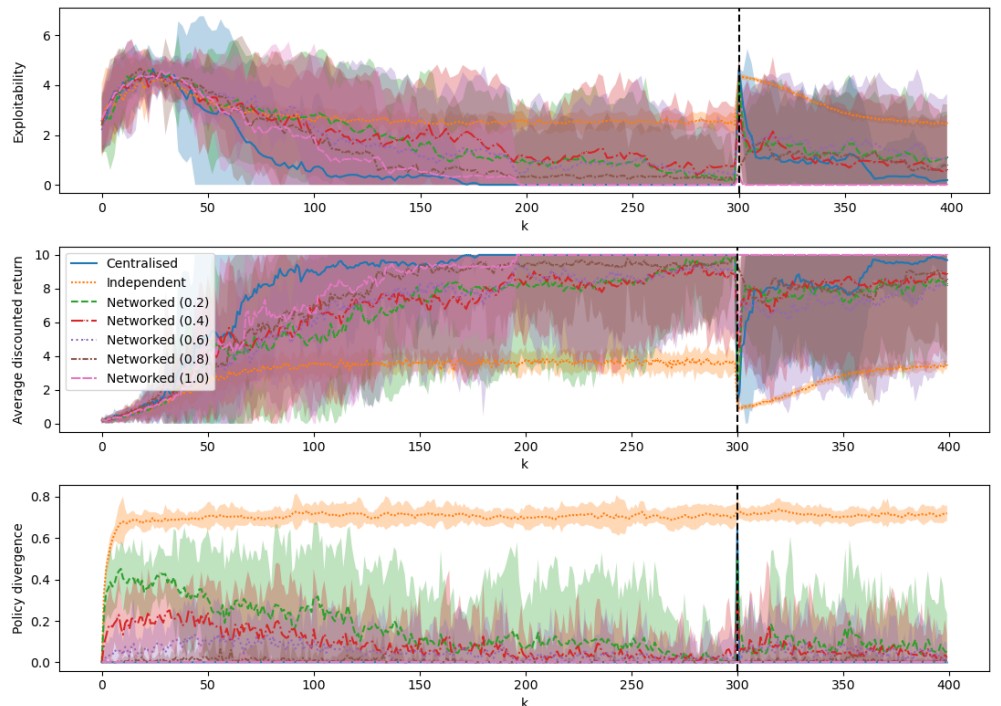

Figure 6: Larger version of Fig. 3. 'Target agreement' game, testing robustness to a five-times increase in population. The networked architectures are quickly able to spread the learnt policies to the newly arrived agents such that learning progress is minimally disturbed, whereas convergence is significantly impacted in the independent case. The largest broadcast radius (1.0), in particular, appears to suffer no disturbance at all, being much more robust than the centralised case, which takes a significant amount of time to return to equilibrium.

where

$$r_{targ}(x) = \begin{cases} x & \text{if } \exists \phi \in \Phi \text{ s.t. } \text{dist}(s_t^i, \phi) = 0 \\ -1 & \text{otherwise,} \end{cases}$$

$$r_{collab}(x) = \begin{cases} x & \text{if } \hat{\mu}_t(s_t^i) > 1/N \\ -1 & \text{otherwise.} \end{cases}$$

## F.2 EXPERIMENTAL METRICS

To give as informative results as possible about both performance and proximity to the NE, we provide several metrics for each experiment. All metrics are plotted with 2-sigma error bars ($2 \times$ standard deviation), computed over the 10 trials (each with a random seed) of the system evolution in each setting. This is computed based on a call to `numpy.std` for each metric over each run.

### F.2.1 EXPLOITABILITY

Works on MFGs frequently use the *exploitability* metric to evaluate how close a given policy $\pi$ is to a NE policy $\pi^*$ (Laurière et al., 2022; Perrin et al., 2020; Laurière et al., 2022; Algumaei et al., 2023; Pérolat et al., 2022). The metric quantifies how much an agent can benefit by deviating from the policy pursued by the rest of the population, by measuring the difference between the return given by a policy that maximises the expected discounted regularised (via $h$) reward $V_h$ for a given population distribution, and the return given by the policy that gives rise to this distribution. If $\pi$ has a large exploitability then an agent can significantly improve its return by deviating from $\pi$, meaning that $\pi$ is far from $\pi^*$, whereas an exploitability of 0 implies that $\pi = \pi^*$. Denote by $\mu^\pi$ the distribution generated when $\pi$ is the policy followed by all of the population aside from the deviating agent; then

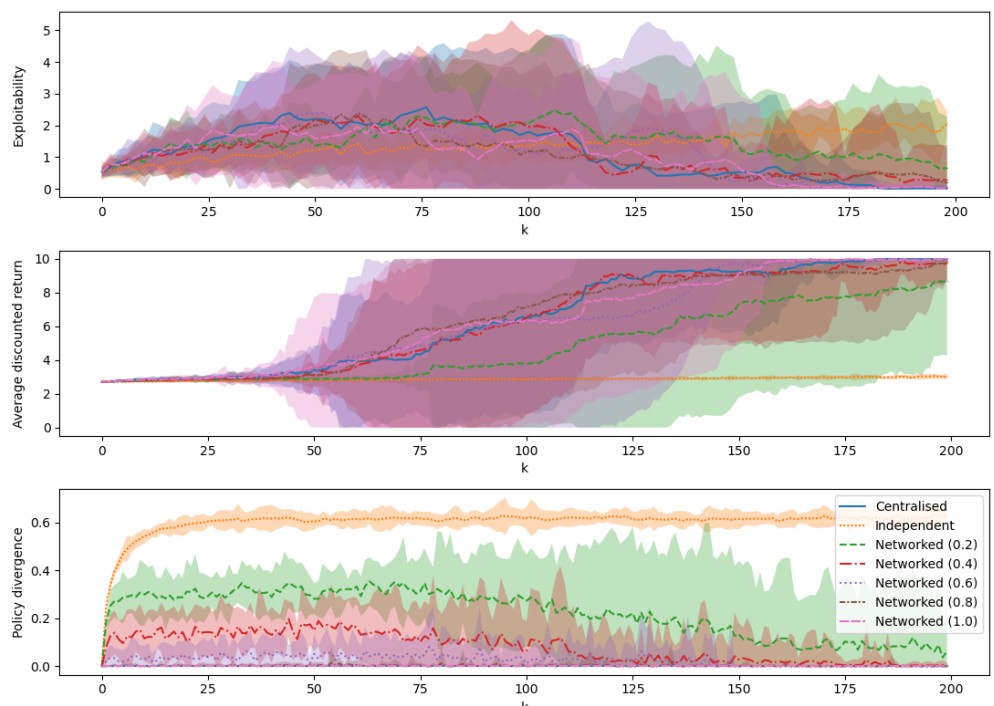

Figure 7: 'Cluster' game. Even with only a single communication round, our networked architecture significantly outperforms the independent case, which hardly appears to be learning at all. All broadcast radii except the smallest (0.2) appear to match and at times even outperform the centralised case.

the exploitability of policy $\pi$ is defined as:

$$\mathcal{E}(\pi) = \max_{\pi'} V_h(\pi', \mu^\pi) - V_h(\pi, \mu^\pi).$$

Since we do not have access to the exact best response policy $\arg\max_{\pi'} V_h(\pi', \mu^\pi)$, we instead approximate the exploitability metric, similarly to (Perrin et al., 2021), as follows. We freeze the policy of all agents apart from a deviating agent, for which we store its current policy and then conduct 40 $k$ loops of policy improvement (we found that 40 iterations was enough to converge to a policy that maximised $V_h$ for the given population distribution). To approximate the expectations, we take the best return of the deviating agent across the 40 $k$ loops, as well as the mean of all the other agents' returns across these same loops. We then revert the agent back to its stored policy, before learning continues for all agents. As such, the quality of our approximation is limited by the number of policy improvement rounds, which must be restricted for the sake of running speed of the experiments. Due to the expensive computations required for this metric, we evaluate it on alternate $k$ iterations.

Since prior works conducting empirical testing have generally focused on the centralised setting, evaluations have not had to consider the exploitability metric when not all agents are following a single policy $\pi_k$, as may occur in the independent or networked settings. The method described above for approximating exploitability involves calculating the mean return of all non-deviating agents' policies. While this is $\pi_k$ in the centralised case, if the non-deviating agents do not share a single policy, then this method is in fact approximating the exploitability of their joint policy $\boldsymbol{\pi}_k^{-d}$, where $d$ is the deviating agent.

### F.2.2 AVERAGE DISCOUNTED RETURN

We record the average discounted return of the agents' policies $\pi_k^i$ during the $M_{pg}$ steps - this allows us to observe that settings that converge to similar exploitability values may not have similar average

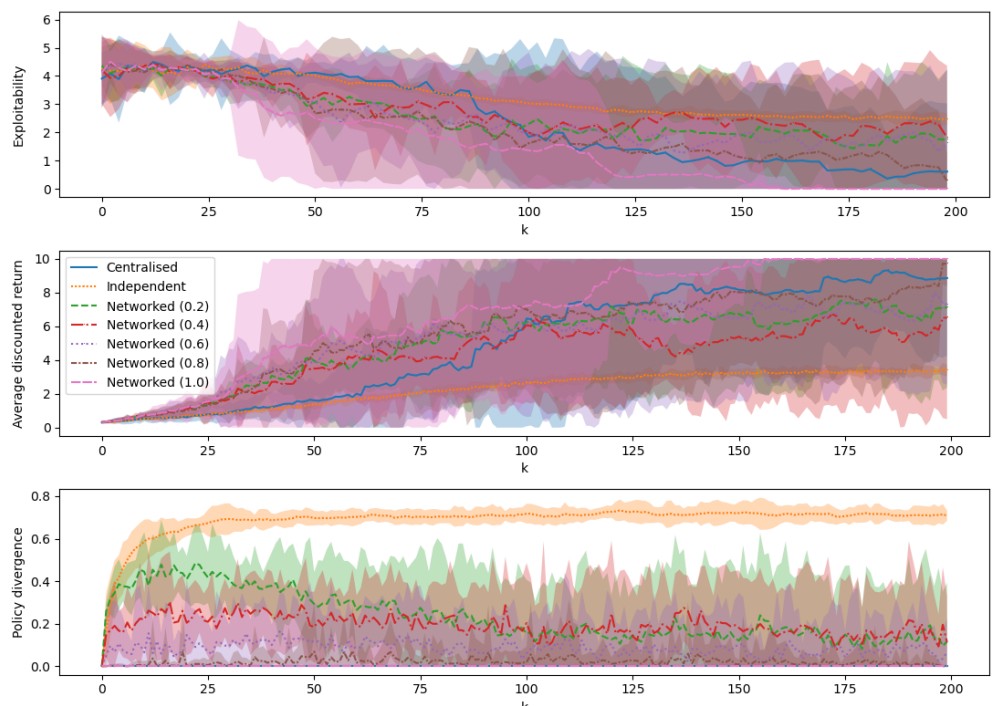

Figure 8: 'Target agreement' game, testing robustness to 50% probability of policy update failure. All the networked cases significantly outperform the independent case and also learn much faster than the centralised case for long periods. The communication network allows agents that have successfully updated their policies to spread this information to those that have not, providing redundancy. Independent learners cannot do this so have even slower convergence than normal; likewise the centralised architecture is susceptible to its single point of failure, hence learning can be slower than in the networked case.

agent returns, suggesting that some algorithms are better than others at finding not just NE, but preferable NE. See for example Fig. 13, where the networked agents converge to similar exploitability as the independent agents, but receive higher average reward.

### F.2.3 POLICY DIVERGENCE

We record the population's average policy divergence $\frac{1}{N}\Delta_k := \frac{1}{N}\sum_{i=1}^{N}||\pi_k^i - \pi_k^1||_1$ for the arbitrary policy $\bar{\pi} = \pi^1$. This allows us to demonstrate that populations approaching the NE (i.e. with joint exploitability approaching zero) do not necessarily actually share a single policy $\pi^*$ as suggested by the theoretical sample guarantees in Sec. 3.3. Our experimental plots show that this is particularly often the case in the independent setting. The greater divergence in the independent case also indicates why convergence is slower here (see Rem. 5).

### F.3 HYPERPARAMETERS

See Table 1 for our hyperparameter choices. In general, we seek to show that our networked algorithm is robust to 'poor' choices of hyperparameters e.g. low numbers of iterations, as may be required when aiming for practical convergence times in complex real-world problems. By contrast, the convergence speed of the independent-learning algorithm (and sometimes also the centralised algorithm) suffers much more significantly without idealised hyperparameter choices. As such, our experimental demonstrations in the plots generally involve hyperparameter choices at the low end of the values we tested during our research.

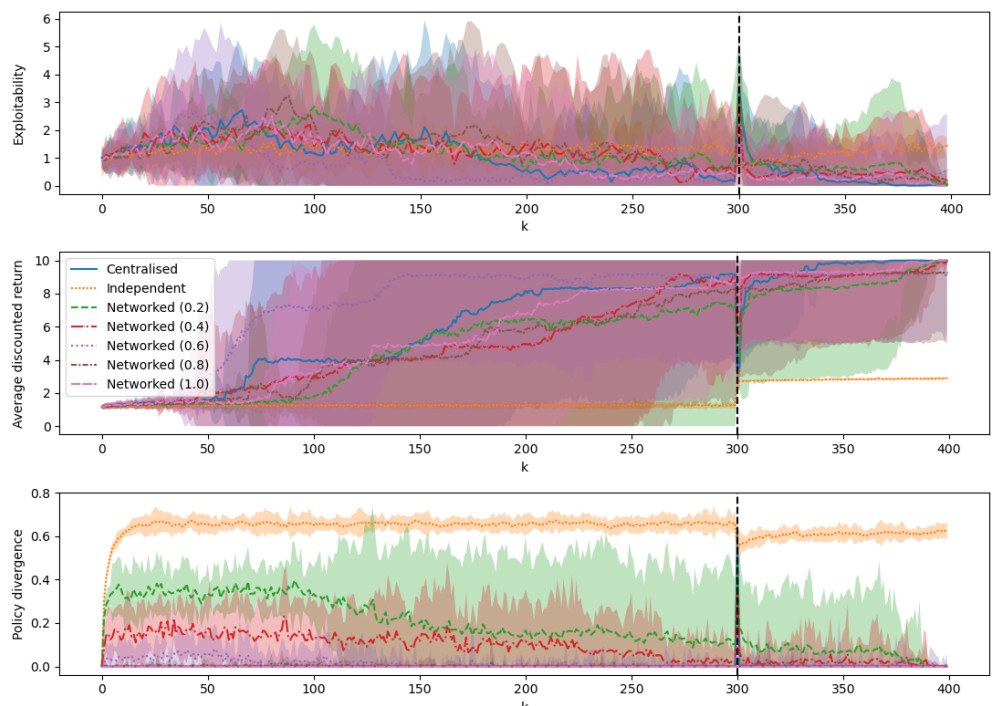

Figure 9: 'Cluster' game, testing robustness to a five-times increase in population. While the independent algorithm appears to enjoy similar exploitability to the other cases (see Rem. 9), we can see from its average return that it is not in fact learning at all; while the return rises after the increase in population size this is only because there are now more agents with which to be co-located, rather than because learning has progressed. Since here, unlike in the 'target agreement' game in Fig. 3, independent agents have hardly improved their return in the first place, we do not see the adverse effect that the addition of agents to the population has on the progress of learning. All networked cases perform similarly to or significantly outperform the centralised case, and all significantly outperform the independent case in terms of return. The communication network allows the learnt policies to quickly spread to the newly arrived agents, such that the progression of learning is minimally disturbed, without needing to rely on the assumption of a centralised learner. The fact that, in all cases, the return prior to the population increase is lower than in Fig. 7, is reflective of the fact that the error in the solution reduces as $N$ tends to infinity.

We can group our hyperparameters into those controlling the size of the experiment, those controlling the number of iterations of each loop in the algorithm and those affecting the learning/policy updates or policy adoption ($\beta, \eta, \lambda, \tau, \gamma$).

### F.4 ADDITIONAL EXPERIMENTS AND DISCUSSION

In this section we showcase results with our standard hyperparameter choices continuing from those shown in Sec. 4.1 (Figs. 7, 8 and 9), and we also vary several hyperparameters to show their effects on convergence (Figs. 10 - 13). We also give an ablation study of our replay buffer in Figs. 14 and 15.

**Remark 9.** *Note that the reward structure of our coordination games is such that exploitability sometimes increases from its initial value before it decreases down to 0. This is because agents are rewarded proportionally to how many other agents are co-located with them: when agents are evenly dispersed at the beginning of the run, it is difficult for even a deviating, best-responding agent to significantly increase its reward. However, once some agents start to aggregate, a best-responding agent can take advantage of this to substantially increase its reward (giving higher exploitability), before all the other agents catch up and aggregate at a single point, reducing the exploitability down to 0. Due to this arc, in some of our plots the independent case may have lower exploitability at certain points than the other architectures, but this is not necessarily a sign of good performance.*

Table 1: Hyperparameters

| Hyper-param. | Value | Comment |
|---|---|---|
| Gridsize | 8x8 / 16x16 | Most experiments are run on the smaller grid, while Figs. 10 and 11 showcase learning in a larger state space. |
| Trials | 10 | We run 10 trials with different random seeds for each experiment. We plot the mean and 2-sigma error bars for each metric across the trials. |
| Pop. | 250 | We tested $N$ in {25,50,100,200,250,500}, with the networked architecture generally performing equally well with all population sizes $\geq 50$. We chose 250 for our demonstrations, to show that our algorithm can handle large populations, indeed often larger than those demonstrated in other mean-field works, especially for grid-world environments (Yongacoglu et al., 2022a; Cui & Koeppl, 2021; Cui et al., 2023; Guo et al., 2023; Subramanian & Mahajan, 2019; Yang et al., 2018b; Ganapathi Subramanian et al., 2021; 2020; Subramanian et al., 2022), while also being feasible to simulate wrt. time and computation constraints.
In experiments testing robustness to population increase, the population instead begins at 50 agents and has 200 added at the marked point. |
| $K$ | 200 / 400 | $K$ is chosen to be large enough to see exploitability reducing, and converging where possible. |
| $M_{pg}$ | 500 / 1000 | We wish to illustrate the benefits of our networked architecture and replay buffer in reducing the number of loops required for convergence, i.e. we wish to select a low value that still permits learning. We tested $M_{pg}$ in {300,500, 600,800,1000,1200,1300,1400,1500,1800,2000,2500,3000}, and chose 500 for demonstrations on the 8x8 grids, and 1000 for the 16x16 grids. It may be possible to optimise these values further in combination with other hyperparameters. |
| $M_{td}$ | 1 | We tested $M_{td}$ in {1,2,10,100}, and found that we could still achieve convergence with $M_{td} = 1$. This is much lower than the requirements of the theoretical algorithms, essentially allowing us to remove the innermost nested learning loop. |
| $C$ | 1 | We tested $C$ in {1,20,50,300}. We choose 1 to show the convergence benefits brought by even a single communication round, even in networks that may have limited connectivity; higher $C$ has even better performance. |
| $L$ | 100 | As with $M_{pg}$, we wish to select a low value that still permits learning. We tested $L$ in {50,100,200,300,400,500}. In combination with our other hyperparameters, we found $L \leq 50$ led to less good results, but it may be possible to optimise this hyperparameter further. |
| $E$ | 100 | We tested $E$ in {100,300,1000}, and choose the lowest value to show the benefit to convergence even from very few evaluation steps. It may be possible to reduce this value further and still achieve similar results. |
| $\gamma$ | 0.9 | Standard choice across RL literature. |
| $\beta$ | 0.1 | We tested $\beta$ in {0.01,0.1} and found 0.1 to be small enough for sufficient learning at an acceptable speed. Further optimising this hyperparameter (including by having it decay with increasing $l \in 0,\ldots,L-1$, rather than leaving it fixed) may lead to better results. |
| $\eta$ | 0.01 | We tested $\eta$ in {0.001,0.01,0.1,1,10} and found that 0.01 gave stable learning that progressed sufficiently quickly. |
| $\lambda$ | 0 | We tested $\lambda$ in {0,0.0001,0.001,0.01,0.1,1}. Since we can reduce $\lambda$ to 0 with no detriment to empirical convergence, we do so in order not to bias the NE. |
| $\tau_k$ | cf. comment | For fixed $\tau_k \, \forall k$, we tested {1,10,100,1000}. In our experiments for fixed $\tau_k$ the value is 100 (see Figs. 12 and 13); this yields learning, but does not perform as well as if we anneal $\tau_k$ as follows. We begin with $\tau_0 = 10000/(10 ** \lceil (K - 1)/10 \rceil)$, and multiply $\tau_k$ by 10 whenever $k \bmod 10 = 1$ i.e. every 10 iterations. Further optimising the annealing process may lead to better results. |

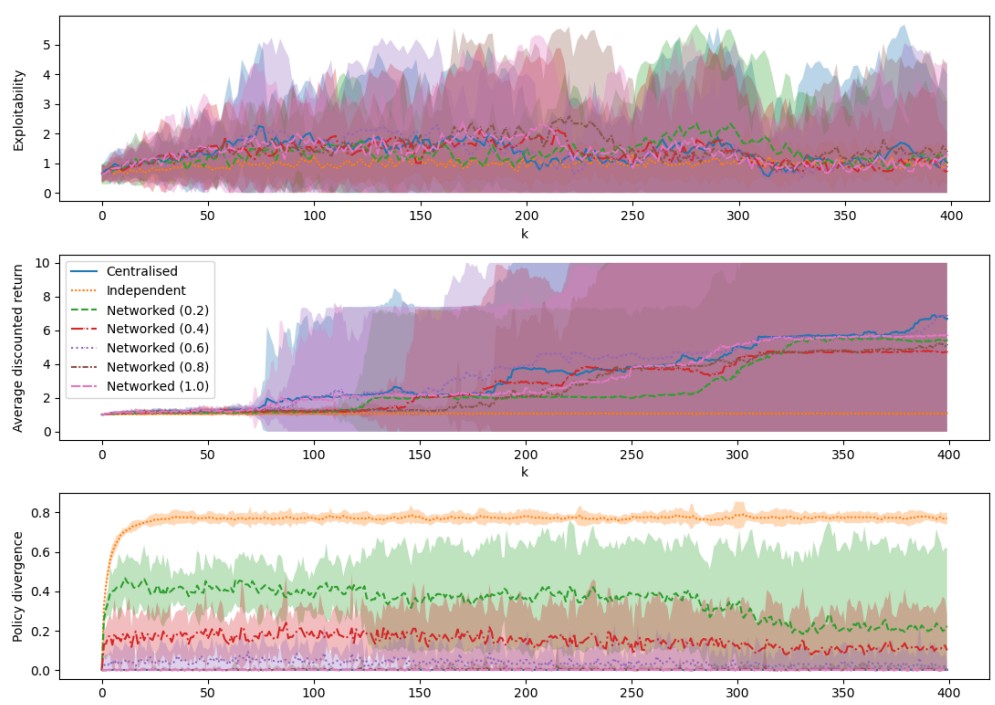

Figure 10: 'Cluster' game on the larger 16x16 grid. While the independent-learning case has similar exploitability to the other settings, we can see that it is not actually learning to increase its return at all, making this an undesirable equilibrium. (I.e. agents are moving about randomly so there is little a deviating agent can do to increase its reward, hence exploitability is low even though the agents are not in fact clustered - see Rem. 9.) All the networked settings perform similarly to the centralised case and significantly outperform the return of the independent agents.

*In fact, we can see in some such cases that the independent case is not learning at all, with the independent agents' average return not increasing and the exploitability staying level rather than ultimately decreasing (see, for example, Figs. 2, 7, 9 and 10).*

In our additional experiments, where the results are discussed fully in each figure's caption, the factors we vary to show the effects on convergence are as follows:

- **Grid size**. Figs. 10 and 11 show the result of learning on a grid of size 16x16 instead of 8x8 as in all other experiments. There is at times greater differentiation in this setting than in the 8x8 grid between the performances of the different broadcast radii of the networked architecture (as is to be expected in a less densely populated environment). The networked architecture continues to significantly outperform the independent case for most broadcast radii, and sometimes even the centralised case.

- **Ablation study of softmax temperature annealing scheme**. Figs. 12 and 13 illustrate the effect of fixed $\{\tau_k\}_{k \in \{0,\ldots,K-1\}} = 100$, where the networked architecture does not perform as well as if we use the stepped annealing scheme employed in all the other experiments and detailed in Table 1. The intuition behind the better performance achieved with the annealing scheme is as follows. If we begin with $\tau_k \to 0$ (such that the softmax approaches being a max function), we heavily favour the adoption of the highest rewarded policies to speed up progress in the early stages of learning. Subsequently we increase $\tau_k$ in steps, promoting greater randomness in adoption, so that as the agents come closer to equilibrium, poorer policy updates that nevertheless receive a high return (due to randomness) do not introduce too much instability to learning and prevent convergence.

- **Ablation study of experience replay buffer**. Figs. 14 and 15 illustrate how crucial is our incorporation of the experience replay buffer. Without it, as in the original theoretical

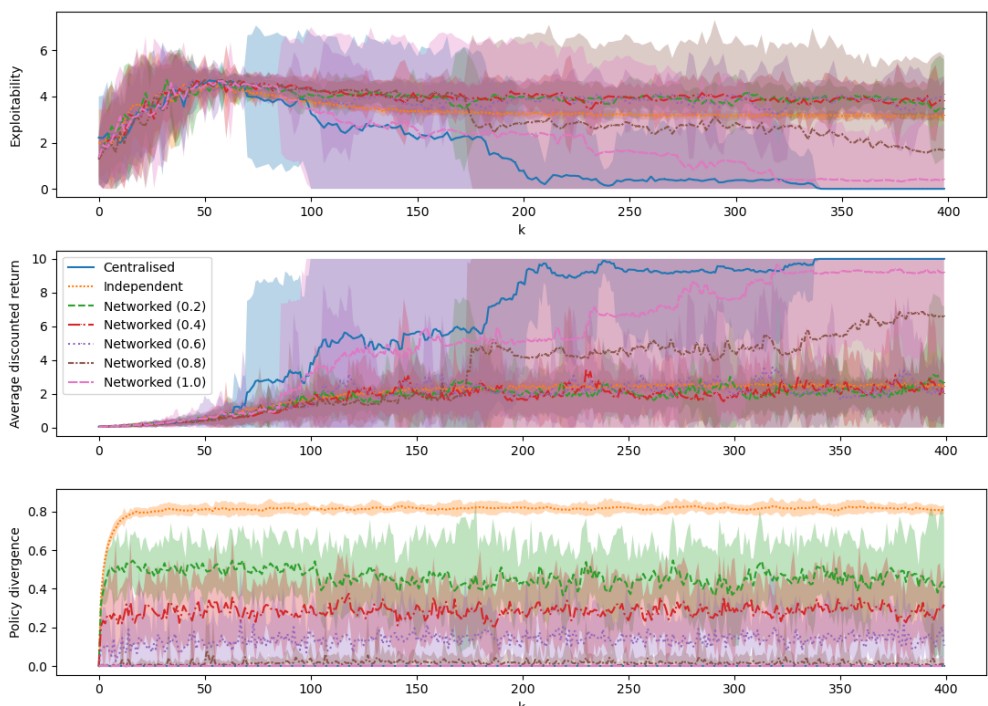

Figure 11: 'Target agreement' game on the larger 16x16 grid. There is greater differentiation in this setting than in the 8x8 grid (Fig. 1) between the different broadcast radii in the networked cases, as is to be expected in a less densely populated environment. The two largest broadcast radii (1.0 and 0.8), which have the most connected networks, significantly outperform the independent case in terms of both exploitability and return. However, the other broadcast radii perform similarly to the independent case.

version of the algorithms, there is no noticeable improvement in any of the agents' returns, i.e. no noticeable learning, even after $K = 400$ iterations. When removing the buffer for these experiments we run the core learning section of the algorithm as in Lines 3-10 of Alg. 1, keeping the hyperparameters the same as in our main experiments, i.e. $M_{pg} = 500$, $M_{td} = 1$, etc. (see Tab. 1).

## G    EXTENDED RELATED WORK

Multi-agent reinforcement learning (MARL) (Zhang et al., 2021b; Busoniu et al., 2008) is a generalisation of reinforcement learning (Sutton & Barto, 2018) that has recently seen empirical success in a variety of domains, underpinned by breakthroughs in deep learning, including robotics (Leottau et al., 2018; Lv et al., 2023; Orr & Dutta, 2023; Guan et al., 2024; Ali et al., 2023), smart autonomy and infrastructures (Shalev-Shwartz et al., 2016; Mannion et al., 2016), complex games (Samvelyan et al., 2019; Vinyals et al., 2019a; Berner et al., 2019), economics (Rashedi et al., 2016; Shavandi & Khedmati, 2022), social science and cooperative AI (Leibo et al., 2017; Cao et al., 2018; Jaques et al., 2019; McKee et al., 2020). However, it has been computationally difficult to scale MARL algorithms beyond configurations with agents numbering in the low tens, as the joint state and action spaces grow exponentially with the number of agents (Xie et al., 2021; Lauriere et al., 2022; Perrin et al., 2020; Shavandi & Khedmati, 2022; Daskalakis et al., 2006; Vinyals et al., 2019b; Mcaleer et al., 2020). Nevertheless, the value of reasoning about interactions among very large populations of agents has been recognised, and an informal distinction is sometimes drawn between multi- and *many*-agent systems (Wang et al., 2020a; Zheng et al., 2018; Cui et al., 2022). The latter situation can be more useful (as in cases where better solutions arise from the presence of more agents (Orr & Dutta, 2023; Ornia et al., 2022; Shiri et al., 2019; Eck et al., 2023)), more parallelisable (Andréen et al., 2016),

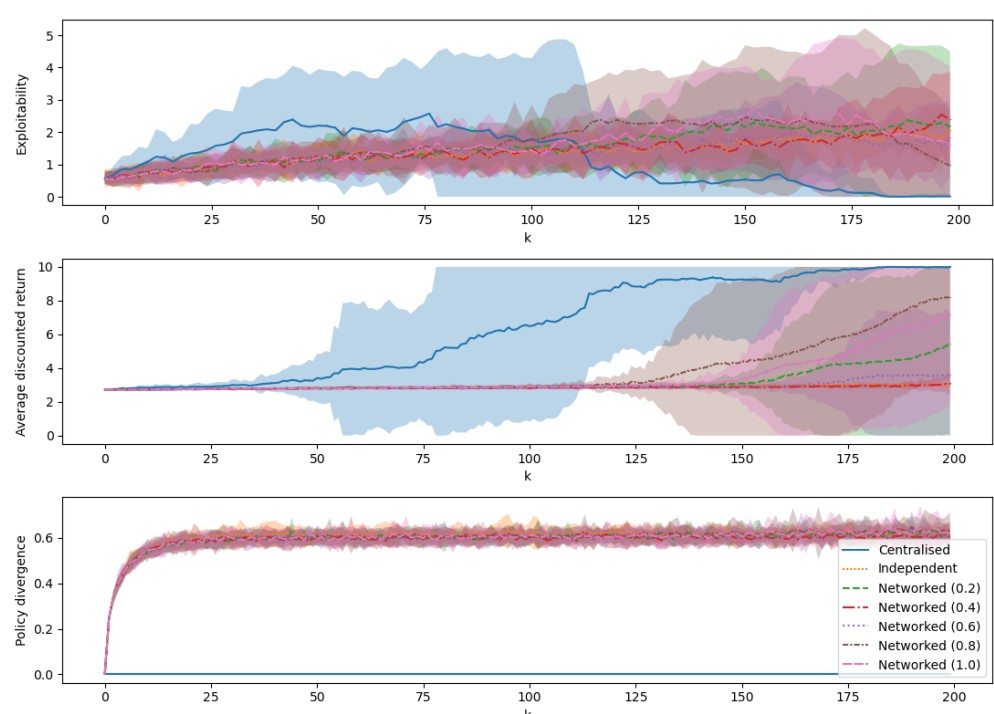

Figure 12: 'Cluster' game with $\tau_k$ fixed as 100 for all $k$; compare this to Fig. 7 where $\tau_k$ is annealed. Without the annealing scheme, the networked architecture appears to perform similarly to the independent case in terms of exploitability, but several broadcast radii outperform the independent case in terms of return, demonstrating that our networked algorithm can still help agents find 'preferable' equilibria. However, whereas with annealing the networked architecture converges similarly to the centralised case, here it performs less well.

more fault tolerant (Chang et al., 2023), or otherwise more reflective of certain real-world systems involving large numbers of decision makers (Eck et al., 2023; Rashedi et al., 2016; Shavandi & Khedmati, 2022; Meigs et al., 2020). Indeed, MFGs have been applied to a wide variety of real world problems, including financial markets (Trimborn et al., 2018); cryptocurrency mining (Li et al., 2024); autonomous vehicles (Huang et al., 2020); traffic signal control (Hu et al., 2023); resource management in fisheries (Yoshioka et al., 2024); crowdsensing (Yang et al., 2023); electric vehicle charging (Dey & Xu, 2023); communication networks (Wang et al., 2024; 2020b); swarms (Le Ménec, 0); data collection by UAVs (Emami et al., 2024); edge computing (Aggarwal et al., 2024; Shen et al., 2024; Miao et al., 2024); cloud resource management (Mao et al., 2022); smart grids, and other large-scale cyber-physical systems (Bauso & Tembine, 2016; Mishra et al., 2023; Benamor et al., 2022).

Our networked communication framework possesses all of the following desirable qualities for mean-field algorithms when applied to large, complex real-world many-agent systems: learning from the empirical distribution of $N$ agents without generation or manipulation of this distribution by the algorithm itself or by an external oracle; learning from a single continued system run that is not arbitrarily reset as in episodic learning (also referred to in other works as a single sample path/trajectory (Zaman et al., 2023; Yardim et al., 2023)); model-free learning; decentralisation; fast practical convergence; and robustness to unexpected failures of decentralised learners or changes in the size of the population.

Conversely, as we emphasise in Sec. 1, the MFG framework was originally mainly theoretical (Lasry & Lions, 2007; Huang et al., 2006). The MFG-NE is traditionally found by solving a coupled system of dynamical equations: a forward evolution equation for the mean-field distribution, and a backwards equation for the representative agent's optimal response to the mean-field distribution, as in Def. 5 (Yoshioka et al., 2024); crucially, these methods relied on the assumption of an infinite population

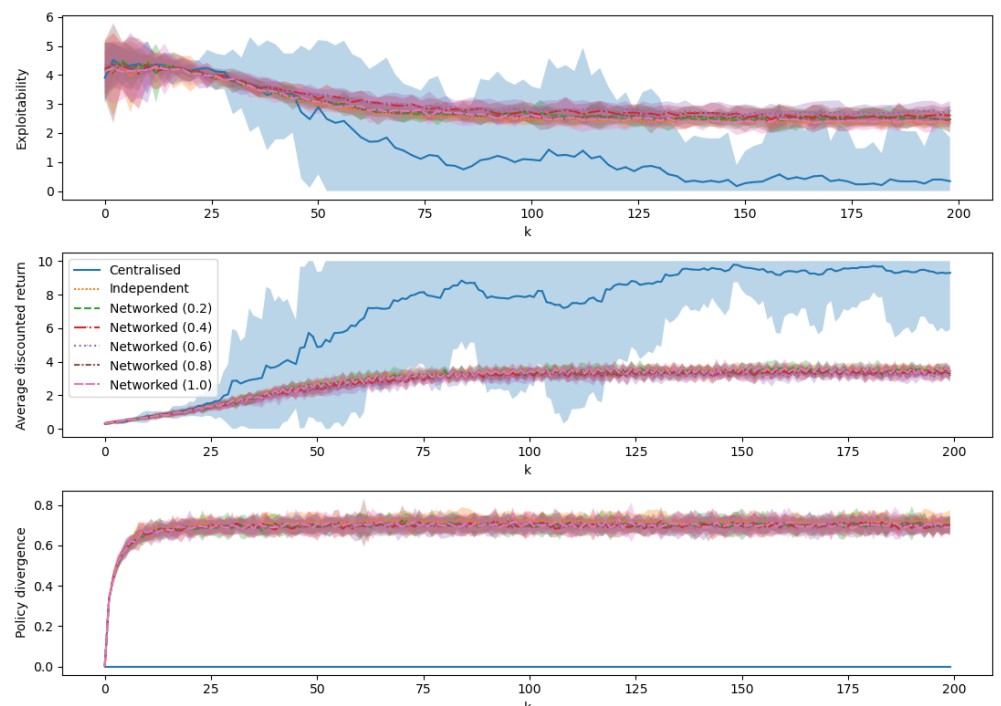

Figure 13: 'Target agreement' game with $\tau_k$ fixed as 100 for all $k$. Without our annealing scheme for the softmax temperature, the networked architecture does not outperform the independent case. Compare this to Fig. 1 which shows the benefit of annealing $\tau_k$.

(Laurière et al., 2022). Early work solved the coupled equations using numerical methods that did not scale well for more complex state and action spaces (Achdou & Capuzzo-Dolcetta, 2010; Carlini & Silva, 2014; Briceño-Arias et al., 2018; Achdou et al., 2020); or, even if they could handle higher-dimensional problems, the methods were based on known models of the environment's dynamics (i.e. they were model-based) (Anahtarci et al., 2023; Guo et al., 2019a; Carmona & Laurière, 2021; Cao et al., 2020; Germain et al., 2022; Fouque & Zhang, 2020; Huang et al., 2024b;a), and/or computed a best-response to the mean-field distribution (Huang et al., 2006; Lauriere et al., 2022; Perrin et al., 2020; Laurière et al., 2022; Guo et al., 2019a; Perrin et al., 2021; Elie et al., 2020; Algumaei et al., 2023). The latter approach is both computationally inefficient in non-trivial settings (Yardim et al., 2023; Laurière et al., 2022), and in many cases is not convergent (as in general it does not induce a contractive operator) (Lauriere et al., 2022; Cui & Koeppl, 2021). Subsequent work, including our own, has therefore moved towards model-free and/or policy-improvement scenarios (Mishra et al., 2023; Laurière et al., 2022; Guo et al., 2023; Subramanian & Mahajan, 2019; Angiuli et al., 2022; Mishra et al., 2020; Cacace, Simone et al., 2021; Perolat et al., 2021; Lee et al., 2021), possibly with learning taking place by observing *N*-agent *empirical* population distributions (Yardim et al., 2023; Hu & Zhang, 2024; Yongacoglu et al., 2022a).

Most prior works, including algorithms designed to solve the MFG using an *N*-agent empirical distribution, have also assumed an oracle that can generate samples of the game dynamics (for any distribution) to be provided to the learning agent (Anahtarci et al., 2023; Guo et al., 2019a; 2023; Anahtarci et al., 2019; Fu et al., 2019), or otherwise that the algorithm has direct control over the population distribution at each time step, such as in cases when the agents' policies and distribution are updated on different timescales (Angiuli et al., 2023), with the *fictitious play* method being particularly popular (Xie et al., 2021; Zaman et al., 2023; Mao et al., 2022; Lauriere et al., 2022; Perrin et al., 2020; 2021; Mguni et al., 2018; Cui et al., 2024; Lauriere, 2021; Subramanian & Mahajan, 2019; Angiuli et al., 2022; Tembine et al., 2012; Cardaliaguet, Pierre & Hadikhanloo, Saeed, 2017; Geist et al., 2021; Frédéric Bonnans et al., 2021). In practice, many-agent problems may not admit such arbitrary generation or manipulation (for example, in the context of robotics or

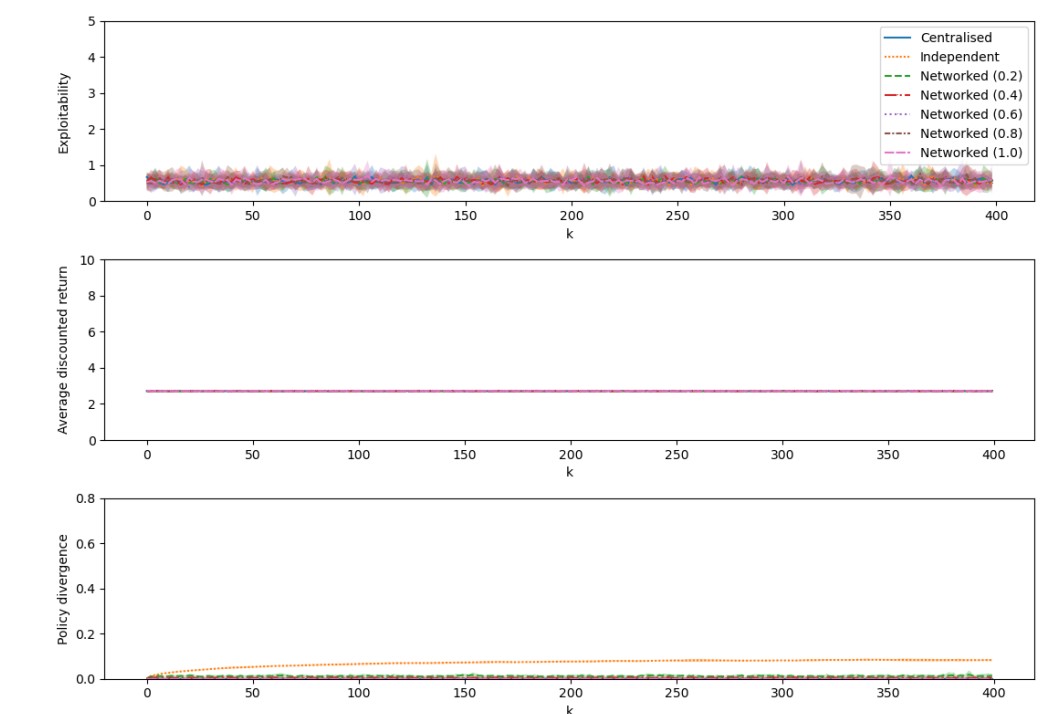

Figure 14: 'Cluster' game with our experience replay buffer removed. There is no noticeable improvement in any of the agents' returns, i.e. no noticeable learning, even after $K = 400$ iterations.

controlling vehicle traffic), and so a desirable quality of learning algorithms is that they update only the agents' policies, rather than being able to arbitrarily reset their states. Learning may thus also need to leverage continuing, rather than episodic, tasks (Sutton & Barto, 2018). Yardim et al. (2023), Yongacoglu et al. (2022a) and our own work therefore present algorithms that seek the MFG-NE using only a single run of the empirical population. Almost all prior work relies on a centralised controller to conduct learning on behalf of all the agents (Xie et al., 2021; Anahtarci et al., 2023; Zaman et al., 2023; Laurière et al., 2022; Guo et al., 2019b). More recent work, including our own, has explored MFG algorithms for decentralised learning with $N$ agents (Yardim et al., 2023; Mguni et al., 2018; Yongacoglu et al., 2022a;b; Grammatico et al., 2015a;b; Parise et al., 2015; Grammatico et al., 2016).

Naturally, inter-agent communication is most applicable in settings where learning takes place along a continuing system run, rather than the distribution being manipulated by an oracle or arbitrarily reset for new episodes, since these imply a level of external control over the population that results in centralised learning. Equally, it is in situations of learning from finite numbers of real, deployed agents (rather than simulated settings) that we are most likely to be concerned with fault tolerance. As such, our work is most closely related to Yardim et al. (2023) and Yongacoglu et al. (2022a), which provide algorithms for centralised and independent learning with empirical distributions along continued system runs: we contribute a networked learning algorithm in this setting. Yongacoglu et al. (2022a) empirically demonstrates an independent learning algorithm when agents observe compressed information about the mean-field distribution as well as their local state, but they do not compare this to any other algorithms or baselines. Yardim et al. (2023) compares algorithms for centralised and independent learning theoretically, but does not provide empirical demonstrations. In contrast, in addition to providing theoretical guarantees, we empirically demonstrate our networked learning algorithm, where agents observe only their local state, in comparison to both centralised and independent baselines, as well as concerning ourselves with the speed of practical convergence and robustness, unlike these works.

Improving the training speed and sample efficiency of (deep) (multi-agent) RL is gaining increasing attention (Yu et al., 2024; Wiggins et al., 2023; Wu et al., 2024a; Patel et al., 2024), though our

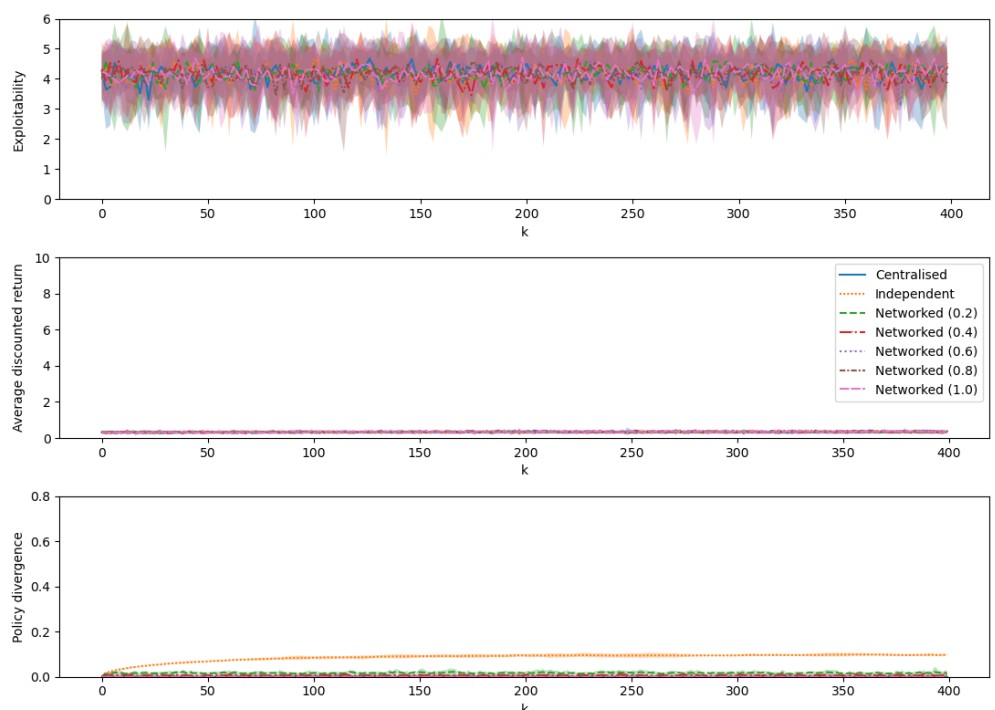

Figure 15: 'Target agreement' game with our experience replay buffer removed. There is no noticeable improvement in any of the agents' returns, i.e. no noticeable learning, even after $K = 400$ iterations.

own work is one of the only on MFGs to be concerned with this. Huang & Lai (2024) trains on a distribution of MFG configurations to speed up inference on unseen problems, but does not learn online in a decentralised manner as in our own work. Similarly, while some attention has been given to the robustness of multi-agent systems to varying numbers of agents, where it is sometimes referred to as 'ad-hoc teaming', 'open-agent systems', 'scalability' or 'generalisation' (Eck et al., 2023), it has more commonly been addressed in MARL (Gao et al., 2024; Dawood et al., 2023) than in MFGs (Wu et al., 2024b). Wu et al. (2024b) presents an MFG approach that allows new agents to join the population during *execution*, but training itself takes place offline in a centralised, episodic manner. Our networked communication framework presented in the current work, on the other hand, allows decentralised agents to join the population during online learning and to have minimal impact on the learning process by adopting policies from existing members of the population through communication.

An existing area of work called 'robust mean-field games' studies the robustness of these games to uncertainty in the transition and reward functions (Bauso & Tembine, 2016; Bauso et al., 2012; 2016; Moon & Başar, 2017; Huang & Huang, 2017; Yang et al., 2018a; Tirumalai & Baras, 2022; Aydın & Saldi, 2023), but does not consider fault-tolerance, despite this being one of the original motivations behind many-agent systems. On the other hand, we focus on robustness to failures and changes in the agent population itself.

We note a similarity between 1. our method for deciding which policies to propagate through the population (described in Sec. 3.4.1) and 2. the computation of evaluation/fitness functions within evolutionary algorithms to indicate which solutions are desirable to keep in the population for the next generation (Eiben & Smith, 2015). Moreover, the research avenue broadly referred to as 'distributed embodied evolution' involves swarms of agents independently running evolutionary algorithms while operating within a physical/simulated environment and communicating behaviour parameters to neighbours (Haasdijk et al., 2014; Trueba et al., 2015), and is therefore even more similar to our setting, where decentralised RL updates are computed locally and then shared with neighbours. In distributed embodied evolution, the computed fitness of solutions helps determine both which are preserved by agents during local updates, and also which are chosen for broadcast or

adoption between neighbours (Hart et al., 2015; Fernández Pérez et al., 2018; Fernández Pérez & Sanchez, 2019). Indeed, some works on distributed embodied evolution specifically consider features or rewards relating to the joint behaviour of the whole population (Gomes & Christensen, 2013; Prieto et al., 2016), similar to MFGs. The adjacent research area of cultural/language evolution for swarm robotics (Cambier et al., 2020; 2018; 2021) has similarly demonstrated the combination of evolutionary approaches and multi-agent communication networks for self-organised behaviours in swarms. However, unlike our own work, none of these areas employ reinforcement learning in the update of policies or the computation of the fitness functions.

We preempt objections that communication with neighbours might violate the anonymity that is characteristic of the mean-field paradigm, by emphasising that the communication in our algorithm takes place outside of the ongoing learning-and-updating parts of each iteration. Thus the core learning assumptions of the mean-field paradigm are unaffected, as they essentially apply at a different level of abstraction (a convenient approximation) to the reality we face of $N$ agents that interact within the same environment. Indeed, prior works have combined networks with mean-field theory, such as using a mean field to describe adaptive dynamical networks (Berner et al., 2023).

## H    Limitations and future work

Our algorithm for the networked case (Alg. 1), as well as prior work on the centralised and independent cases (Yardim et al., 2023), all have multiple nested loops. This is a potential limitation for real-world implementation, since the decentralised agents might be sensitive to failures in synchronising these loops. However, in practice, we show that our networked architecture provides redundancy and robustness (which the independent-learning algorithm lacks) in case of learning failures that may result from the necessities of synchronisation (see Appx. E). We have also shown that networked communication in combination with the replay buffer allows us to reduce the hyperparameter $M_{td}$ to 1, essentially removing the inner 'waiting' loop. Nevertheless, our algorithm still features multiple loops, and future work lies in simplifying the algorithms further to aid practical implementation, possibly by techniques such as asynchronous communication (Ma et al., 2024).

Since the MFG setting is technically non-cooperative, we have preempted objections that agents would not have incentive to communicate their policies by focusing on coordination games, i.e. where agents seek to maximise only their individual returns, but receive higher rewards when they follow the same strategy as other agents. In this case they stand to benefit by exchanging their policies with others. Nevertheless, in real-world settings, the communication network could be vulnerable to malfunctioning agents or adversarial actors poisoning the equilibrium by broadcasting untrue policy information. It is outside the scope of this paper to analyse how much false information would have to be broadcast by how many agents to affect the equilibrium, but real-world applications may need to compute this and prevent it. Future research to mitigate this risk might build on work such as Piazza et al. (2024), where 'power regularisation' of information flow is proposed to limit the adverse effects of communication by misaligned agents.

While our MFG *algorithms* are designed to handle arbitrarily large numbers of agents (and theoretically perform better as $N \to \infty$), the *code* for our experiments naturally still suffers from a bottleneck of computational speed when simulating agents that in the real world would be acting and learning in parallel, since the GPU can only process JAX-vectorised elements in batches of a certain size.

Our experiments are based on relatively small toy examples that clearly demonstrate the advantages of our new approach, but which lack the complexity of the real-world applications to which we wish to address the approach. Moreover in our current experiments only the reward function depends on the mean-field distribution, and not the transition function, even though this is possible in theory; we will explore this element in future experiments. It is feasible that in more complex problems, it may not be possible to reduce hyperparameter values to the same extent we have demonstrated in our experimental examples.

Moreover, real-world examples would likely require handling larger and continuous state/action spaces (the latter perhaps building on related work such as Tang et al. (2024)), which in turn may require (non-linear) function approximation. Future work therefore involves incorporating neural networks into our networked communication architecture for oracle-free, non-episodic MFG settings. Extending our algorithms in this way, which would depend on modifying the PMA step (Vieillard

et al., 2020; Wu et al., 2024b), would allow us to introduce communication networks to MFGs with *non-stationary* equilibria, in addition to those with larger state/action spaces. Our method for non-stationary games will likely have agents' policies depending both on their local state and also on the population distribution (Mishra et al., 2020; Laurière et al., 2022; Perrin et al., 2022; Carmona et al., 2023), but such a high-dimensional observation object would require moving beyond tabular settings to those of function approximation. The present work demonstrates the benefits of the networked communication architecture when the Q-function is poorly estimated and introduces experience relay buffers to the setting of learning from a continuous run of the empirical system. Both elements are an important bridge to employing (non-linear) function approximation in this setting, where the problems of data efficiency and imprecise value estimation can be even more acute, and where we may want to employ experience replay buffers to provide uncorrelated data to train the neural networks (Zhang & Sutton, 2017). When the policy functions are approximated rather than tabular, our agents would communicate the functions' parameters instead of the whole policy as now.

In our future work with non-stationary equilibria, where agents' policies will also depend on the population distribution, it may be a strong assumption to suppose that decentralised agents with local state observations and limited communication radius would be able to observe the entire population distribution. We will therefore explore a framework of networked agents estimating the empirical distribution from only their local neighbourhood as in (Ganapathi Subramanian et al., 2021), and possibly also improving this estimation by communicating with neighbours (Yongacoglu et al., 2022a), such that this useful information spreads through the network along with policy parameters.

