# OpenReview forum: "Networked Communication for Decentralised Agents in Mean-Field Games"
_ICLR.cc/2025/Conference — Submitted to ICLR 2025_

### Official Review · Reviewer_yxK2 · 2024-10-17

**Soundness:** 2
**Presentation:** 1
**Contribution:** 3
**Rating:** 3
**Confidence:** 3

**Summary:**

This paper studies mean field games in which there is assumed to be a stationary distribution of agents’ state at equilibrium. The authors propose an algorithm for networked policy learning in these games, prove that its performance interpolates that of purely independent and centralized learning schemes, and evaluate the performance of all three approaches in two toy problems. This evaluation shows that the proposed approach works well in parameter regimes and with algorithmic modifications which are not explained by the theory.

**Strengths:**

The main theoretical result is particularly interesting: it is quite intuitive that learning in a networked setting would interpolate the performance of independent and centralized schemes, but proving that this is so is a very nice contribution.

**Weaknesses:**

* It would be better if the limitations and future work were included in the main text, not the appendix.
* Operator \Gamma_q is not defined properly in the main text. First use is on line 233.
* It would be useful to provide some explanation of what is going on in Definition 7 for readers unfamiliar with Yardim 2023.
* The construction of Definition 8 is not self-contained. All variables ought to be clearly defined in the main text; otherwise the paper is very difficult to understand.
* Nitpick: I suggest moving Alg. 1 up 1 page, so that one does not need to scroll so much between its introduction on line 256 and the algorithm itself.
* The prose description of the algorithm in Sec. 3.2 is difficult to understand as it refers to terms and relations which are not on the same page. I suggest defining all terms formally and precisely inline with proper equation numbers, and structuring the description at a high level with paragraph headers whose titles describe what is going on.
* I also suggest adding comments in the algorithm in order to make it more clear what is going on in different parts. Currently, it is hard to parse because there are so many symbols that were not clearly defined in the main text, and are also not explained in the algorithm block clearly enough to make sense of them without referring back to the main text and trying to connect the dots. For example, I have no idea what \sigma^i_k is precisely, and can only deduce that it is a scalar by examining its use in line 15 of Alg. 1. Line 351 says it can be chosen at random (on line 11 of Alg. 1), but that only adds to my confusion.
* \epsilon is undefined in Thm. 2.
* Graph diameter is never defined as far as I can see.
* The transition buffer usage appears to be a major contribution of the paper, yet it is not described precisely in the final subsection of Sec. 3.
* Fig. 1 is way too small to be readable. All text should be readable by the naked eye. (Same for Fig. 2 and Fig. 3.)
    * It is unclear how many independent trials these results are based upon. Confidence intervals are overlapping for many of the methods, which makes it difficult to visually distinguish between them, and also makes any statistical claims unclear. For example, stating that “our networked architecture significantly outperforms…” (line 508) does not really make sense because all the confidence intervals are so often overlapping, and without a direct pointer to a graph where this is not happening it is hard to justify that conclusion.
    * “Exploitability” is not defined as far as I see in the main text. Neither is “policy divergence.”
* More precise descriptions of the tasks and experiment setup should be provided in the main text of Sec. 4. Without additional information here, the results are difficult to interpret.
* It would be nice if there were figures (in the main text) illustrating the relative performance of the theoretically-sound version of the algorithm with the “hacked” version which breaks the theoretical guarantees but is claimed to perform better. This seems to be a major oversight in the results, since it is such an important part of the authors’ claimed contributions in the introduction and in the discussion on page 10.
* The point on line 538 does not make sense because it relies on something defined only in the appendices.

**Questions:**

* The restriction to stationary MFGs seems to be quite substantial, in the sense that — I can imagine — a player could have an incentive to act in a way which differs from how others act in some games, and that could plausibly lead to a change in the empirical distribution of agents’ state. By restricting ourselves to cases where this does not happen, are we removing potentially-important NE from consideration?
    * My sense is that the only reason for considering this restriction is that it allows us to write down policies as maps from agents’ individual state to action distributions. Without this restriction of stationarity, I believe we would need players’ policies to depend also upon the current (overall) state distribution. If this is the case, it would be helpful to explain things in this way to justify the restriction as necessary for tractability reasons.
* Thm. 1 asserts that there is a unique MFG-NE point. Under what conditions is this assured?
* In Thm. 3, why are there three different upper bounds being discussed? There only appears to be one ultimate upper bound for all three methods in Thm.  2, because the intermediate bounds are not quantified explicitly. Reading the equation on line 373, it seems like we are just talking about somehow roughly quantifying the differences between the three terms on the left of the equation on line 359.
    * I don’t understand how (1/d_G)^C = 0 on line 376. That does not seem arithmetically possible.
    * I am not clear on how to interpret the \approx symbols in line 373. Is there a precise interpretation?
* It is unclear to me why the changes discussed on line 404 break the theoretical properties derived in the previous section. Especially because this is so central to the structure of the paper, I urge the authors to emphasize the relevant assumptions in the main text rather than punting so much to the appendices.
* The intuition in the paragraph starting on line 418 is very clear! Can the authors provide any theoretical analysis which supports this intuition? Or failing this, can the authors provide a clear remark explaining why such analysis is difficult so as to provide a template for future investigation on this topic?
* Why does the green networked curve appear to be doing better in Fig. 2 than it was in Fig. 1? Isn’t the scenario harder? What is going on here?
    * The statement “independent learners cannot do this and hardly appear to learn at all” is confusing because the reader must deduce on their own that this is referring only to the center subfigure and not to the others. Labelling subfigures would help.
* It is not quite clear to me how the proposed method is handling the change in population size. I suggest the authors clarify exactly how this is done.
* I would appreciate further discussion of the non-uniqueness issue raised in line 535, and how it relates to convergence to a single equilibrium as \lambda \to 0. It is not clear to me how coordination mathematically arises here, or why things do not break (e.g., due to ill-conditioning) as \lambda \to 0.

---

> ### Author Response · Authors · 2024-11-20
> **Part 1: Rebuttal by Authors**
>
> Thank you very much for reviewing our paper. We are pleased that you find it 'a very nice contribution' with the theoretical result 'particularly interesting'. Given this praise, we are surprised that you have initially rejected our paper.
>
> Weakness 1/2/4/12/14: The ICLR page limit for the main text is 10 pages, so as you can appreciate, it is not possible to keep all the details in the main text even if we would like to (without moving something else to the appendix instead). $\textbf{None of the other reviewers}$ share your concerns regarding readability being affected by the placement of details in the appendix or on different pages of the main paper, and in fact reviewer HFSn says '$\textbf{the paper is well-written and easy to follow. The authors provide all the necessary theories and experimental settings in the main text,}$ with more detailed proofs and specifics included in the appendix' and 'the theoretical analysis in the paper is comprehensive'. Meanwhile reviewer y6Sh says 'The paper is $\textbf{very well written and very clear}$ … The theoretical results from analyzing the number of independent learned policies are $\textbf{elegant, interesting and significant}$ … The discussion of related work and placement of contribution is very extensive and good'. We always refer to the relevant section of the appendix / lines of the algorithms / definitions where items are fully defined, accompanied with hyperlinks to aid navigation. It is very common to leave full details of experimental settings to the appendix.
>
> Weakness 3: As clearly stated, this is a temporal difference update, a very common method in reinforcement learning. Here it is augmented with the regulariser $h$ that we have already defined ourselves in Lines 134-135.
>
> Weakness 5: We will update this in the manuscript.
>
> Weakness 6: Having moved Alg.1 as per your suggestion, the prose description is now on the same page as the terms to which it refers. We give a line-by-line reference from the prose description to the relevant line of the algorithm. Our definitions are numbered and our section titles describe which element of the algorithm we are discussing. As mentioned above, other reviewers praise our paper's writing, clarity and comprehensiveness.
>
> Weakness 7: Our prose description of the algorithm in Sec. 3.2 has line-by-line references to the part of the algorithm being described. Having moved Alg. 1 as you suggested, these descriptions are now on the same page as the algorithm as well as being hyperlinked. We think that comments in the algorithm itself will make things harder to understand due to cluttering, as well as pushing us over the page limit. You say that the algorithm has '$\textit{so many}$ symbols that were not clearly defined in the main text'. What are the symbols that have not been introduced in the text? $\sigma^i_{k}$ is indeed a scalar which we have introduced in Lines 268-271; we then also devote all of Sec 3.4.1 to discussing it. These lines specifically discuss the arbitrary choice of this scalar for the theoretical results, so we find this criticism unfounded. As mentioned above, other reviewers praise our paper's writing, clarity and comprehensiveness.
>
> Weakness 8: As indicated in Thm. 2, $\varepsilon$ has the same meaning as in Thm. 1, but we will clarify this in the manuscript.
>
> Weakness 9: The diameter of a network is commonly defined as being the maximum length of the shortest path between any pair of nodes in the graph. We will clarify this in the manuscript.
>
> Weakness 10: In what way does Sec. 3.4.2 not precisely describe the buffer? We provide a colour-coded version of the algorithm highlighting the modifications for the buffer, and give a line-by-line prose description of how it works with hyperlinks to the relevant lines. We do not believe this criticism to be well-founded, and other reviewers praise our paper's writing and clarity.
>
> Continued in next comment...

---

> > ### Author Response · Authors · 2024-11-20
> > **Part 2: Rebuttal by Authors**
> >
> > ... Continued from above.
> >
> > Weakness 11: The small size was necessary to fit the page count, but we will reproduce these plots larger in the appendices to aid visibility. As stated clearly in Line 1722 and our hyperparameter table (Line 1897, Table 1) we run 10 independent trials for each experiment. We give 2-sigma error bars, i.e. a 95.4\% confidence interval, which is fairly conservative, and this is a not a concern raised by any other reviewer. It is clear that the average performance in the networked and centralised cases is substantially better than the independent case in most scenarios; to help justify statistical significance, we point you to: Fig. 2, where all the networked cases' error ends with no overlap with that of the independent case (and for much of the time only the 0.4 radius has any overlap); and Figs. 4 and 6, which have a similar lack of overlap to Fig. 2 but this time the exception is the 0.2 radius. We can make these pointers clear in our updated manuscript.
> > As we explicitly state and hyperlink in Line 512 of the main text, the metrics are defined fully in Appx. E.2 for reasons of space.
> >
> > Weakness 13: When we ran our original networked algorithm (as well as the centralised and independent alternatives) without the replay buffer, we did not find any discernable learning or improvement in return/exploitability $\textit{at all}$, despite leaving the algorithms running for several days and many thousands of $k$ iterations. We are currently rerunning the original versions of the algorithms without the buffers to provide empirical evidence of this lack of learning in practical time. We will update the manuscript with these ablation results when they are ready, hopefully by the end of this discussion period, or otherwise in time for the final version (since they are inherently slow to run).
> >
> > Question 1: This first point is slightly distinct from your broader question - your intuition regarding the relationship here is not quite correct. Games where agents have an incentive to act differently to others do not necessarily lead to non-stationary population distributions - e.g. a coverage game where agents must spread to different parts of the grid - and equally agents can have identical policies and still have non-stationary distributions - e.g. a shoal of fish evading a shark. So it is not necessarily the case that we are removing potentially important equilibria, but more we are restricting the types of task we consider.
> >
> > When the distribution is non-stationary, policies must either be time-dependent (most common in the finite-horizon setting) or population-dependent. This is much harder to solve, as it may require solving not just one MDP but an MDP for every possible population distribution. Therefore works on MFGs (such as many of those we cite) have very commonly focused on methods for solving stationary games, and have only more recently begun considering population-dependent policies. We will clarify in the manuscript that this restriction is indeed for tractability reasons in our setting; as we note in Lines 2167-2172, extension to non-stationary equilibria is left to future work.
> >
> >
> > Question 2: The uniqueness of the equilibrium relates to the regulariser we introduce in Lines 134-138, in particular to a sufficiently large value for the scaling parameter $\lambda$. As we state in Line 535, in general the unregularised MFG-NE will not be unique. Yardim 2023 contains further details.
> >
> > Question 3i: Thm. 2 should also give the relationship between each case in terms of their respective upper bounds, i.e. $ub_{\text{cent}} \leq ub_{\text{net}} \leq ub_{\text{ind}}$. We will update this in the manuscript.
> >
> >
> > Question 3ii: We wrote it this way for simplicity, but you are technically correct. We will instead write the term $\left(\frac{1}{d_\mathcal{G}}\right)^C$ in the bound in Line 373  as the piecewise function $f(C, d_\mathcal{G})$, defined as the following:
> >
> > $f(C, d_\mathcal{G}) =
> > \begin{cases}
> > \left(\frac{1}{d_\mathcal{G}}\right)^C &\text{if } C < d_\mathcal{G},\\\\
> > 0 &\text{if } C \geq d_\mathcal{G}
> > \end{cases}
> > $ .
> >
> >
> >
> > Continued in next comment...

---

> > > ### Author Response · Authors · 2024-11-20
> > > **Part 3: Rebuttal by Authors**
> > >
> > > ... Continued from above.
> > >
> > > Question 3iii: The convergence rate of a networked max-consensus algorithm such as this is $\frac{1}{d_\mathcal{G}}$, meaning that the number of policies in the population decreases monotonically by a factor of $\textit{approximately}$ (but not exactly) $\frac{1}{d_\mathcal{G}}$ with each communication round, up until the point when $C \geq d_\mathcal{G}$, after which there is only one policy remaining. Given that this is the case, when the number of policies in the population decreases at a rate of $\frac{1}{d_\mathcal{G}}$, we can also say that the policy divergence
> > >
> > > (defined as $\Delta_{k}$ := $\sum_{i=1}^N ||\pi^i_k -\pi^{\mathrm{max}}_k||_1$, for the consensus policy $\pi^{\mathrm{max}}_k\in\Pi$)
> > >
> > > decreases monotonically at a rate of $\textit{approximately}$ $\frac{1}{d_\mathcal{G}}$. Since this convergence rate reflects the $\textit{average}$ decrease in the number of policies with each communication round, we say that the order of the difference is approximately as given rather than exactly. We can clarify this in the manuscript if deemed helpful.
> > >
> > >
> > >
> > > Question 4: Again, it is not possible to include all the technical details in the 10-page limit and it is very common to leave full details to the appendix; other reviewers praise the clarity and elegance of our theoretical results. The full version of Thm. 1, given in Appx B.2, includes that we have $\varepsilon > 0$, $\beta_m = \frac{2}{(1 - \gamma)(t_0 + m -1)}$, and a problem-dependent constant $a \in [0,\infty)$ such that $M_{pg} > \mathcal{O}(\varepsilon^{-2-a})$ and $M_{td} > \mathcal{O}(log^{2}\varepsilon^{-1})$. This makes $\beta_m$ very small, and $M_{pg}$ and $M_{td}$ very large. As noted in Lines 260-261, $M_{td}$ theoretically needs to be large to create a delay between each TD update to reduce bias when using the empirical distribution to approximate the mean field in a continuous system evolution, while $M_{pg}$ theoretically needs to be large (and $\beta_m$ small) to adequately approximate the Q-function. While theoretically necessary, these values prohibit learning in any practical time, so we seek to change them while still permitting learning in practice.
> > >
> > > Question 5: After the PMA (policy update) step of the algorithm, we have randomly updated policies \{$\pi^i_k$\}$_i$, $i = 1, ..., N$.
> > >
> > > Let us assume that in the networked case, as per Remarks 4 and 5, the structure of the dynamic network and the values of $C$ and $\tau_k \in \mathbb{R}_{>0}$ are such that the population shares a single policy after the $C$ communication rounds. Call this policy $\pi^{\text{net}}_k$, and its associated finitely approximated return $\sigma^{\text{net}}_k$. Recall that the centralised case, where the updated policy of an arbitrary agent is pushed to all the others, is equivalent to a networked case where policy consensus is reached on a random one of the policies \{$\pi^i_k$\}$_i$, $i = 1, ..., N$; call this policy arbitrarily given to the whole population $\pi^{\text{cent}}_k$, and its associated finitely approximated return $\sigma^{\text{cent}}_k$. Since $\pi^{\text{cent}}_k$ is chosen at random, in expectation $\sigma^{\text{cent}}_k$ will be the mean value of {$\sigma^i_k$\}$_i$, $i = 1, \ldots, N$ for each $k$, though there will be high variance. Conversely, the softmax adoption probability (Line 27 of Alg. 2) for the networked case is such that in expectation the $\pi^{\text{net}}_k$ that gets adopted by the whole networked population will have higher than average $\sigma^{\text{net}}_k$ (indeed if $\tau_k \rightarrow 0$ it will have the highest $\sigma^{\text{net}}_k$ for each $k$). As such, $\mathbb{E}[\sigma^{\text{net}}_k] > \mathbb{E}[\sigma^{\text{cent}}_k]$. There is also less variance in $\sigma^{\text{net}}_k$ than $\sigma^{\text{cent}}_k$, as the former is biased towards higher values. If the networked case results in higher average $\sigma_k$ being adopted than the centralised case, then the policies of which $\sigma_k$ gives an approximated return are also biased towards being better performing (and with less variance in quality). Thus the networked case can learn and converge faster than the centralised case, by choosing updates in a more principled manner.
> > >
> > >
> > >
> > > Continued in next comment...

---

> > > > ### Author Response · Authors · 2024-11-20
> > > > **Part 4: Rebuttal by Authors**
> > > >
> > > > ... Continued from above.
> > > >
> > > > Question 6i: In our 'target agreement' experiments there are four targets, each located at one corner of the grid. In general we might consider the second ('cluster') scenario harder since there are more possible equilibrium points over which to find consensus (agents can cluster at any gridpoint rather than only one of four corners) and agents are suffering from a 50\% learning failure rate. However, from the perspective of agents with the smallest communication radius (the green curve), the first scenario may be harder. Here agents are deciding which $\textit{corner}$ to meet at (i.e. the possible equilibrium points are as far as possible from each other, and further than the communication radius), but are probably unable to communicate far enough to break the tie between them to coordinate on a consensus point. They therefore remain divided across the corners and gain limited reward. While it may be less intuitive, having more possible equilibrium points, each of which is separated from the next one by a distance that is smaller than the communication radius, may make it less likely that different parts of the population become isolated from each other in 'local optima'. Moreover, it $\textit{may}$ be that the 50\% chance of failure is also counterintuitively advantageous for this small broadcast radius, as it also may mean fewer ties need breaking (since in general policies might receive similar returns while pulling in different directions towards different possible equilibria).
> > > >
> > > > Question 6ii: The subfigures are inherently labelled by their y-axis, but we will indicate in the manuscript that we are mainly referring to the middle subfigure in this instance.
> > > >
> > > >
> > > > Question 7: We discuss this already in Lines 1604-1609, and the caption to Fig. 3. Please let us know if there is something specific that is still not clear.
> > > >
> > > > Question 8: Many games have multiple possible equilibria (with some possibly more desirable than others, as in the 'stag hunt' game in game theory). The fact that there is not a single (unique) equilibrium in a certain game does not in general prevent players from converging on one of the multiple (non-unique) equilibria, but it may make it harder to converge, for example if players oscillate between the equilibrium points they are aiming for, which may in turn incentivise oscillation in other players. One way to help players settle on one of the multiple possible equilibria is to provide an external method for aligning relevant parts of their strategies, e.g. a centralised controller that forces all agents to have identical policies, or inter-agent communication that allows agents to adopt the same policies as each other. Alternatively, one can add regularisation to the game, which means that the rewards are modified to limit the number of equilibria to a single one to which the players are pushed to converge, but at the expense of biasing the equilibrium away from what may have been originally intended by the reward structure. For example, in a cluster game with heavy entropy regularisation the players may converge to an equilibrium where they are all always moving randomly around the grid.  While this is technically the unique equilibrium of this biased game, it does not reflect the original problem of having the agents cluster into one place, such that this regularisation is undesirable. As such, while regularisation is theoretically required to prove convergence, in practice we want to reduce $\lambda$ to 0 if possible, meaning that techniques that encourage convergence (such as our networked communication scheme, which converges faster than the independent case) are desirable.
> > > >
> > > >
> > > > We hope these clarifications have addressed your concerns. Given your praise for our ‘particularly interesting’ and ‘very nice contribution’ and the praise from other reviewers about the structure, writing and clarity of our paper, we hope you will raise your score to an accept.

---

> > > > > ### Author Response · Authors · 2024-11-25
> > > > > **Discussion period closing**
> > > > >
> > > > > Since the discussion period has nearly finished, we kindly ask you to read and respond to our rebuttal. We have provided an extensive response to your review, and we wanted to check that we have been able to address all of your concerns. If there are no further barriers to acceptance, we would appreciate if you could update your score accordingly. Thank you very much for your time.

---

> > > > > > ### Comment · Reviewer_yxK2 · 2024-11-26
> > > > > > **Thank you for the detailed response**
> > > > > >
> > > > > > Thanks for the detailed responses. As is clear from my description of the paper's strength, I very much want to like this paper. The authors have done a nice job of clarifying several of the questions I had, though many of the responses above amount to "we just disagree with you" or "it's in the appendix, you should have found it there." I will respond with advice I once got from a friend when I was a PhD student, in a similar context: "when someone says there's a problem with some part of your paper, that probably means there was a better way to explain it so that a well-intentioned, intelligent reader is less likely to get confused."
> > > > > >
> > > > > > I will raise my score slightly to reflect my improved understanding of the paper. However, until this paper is substantially more self-contained in the main text so that all key ideas and important insights can be well understood without digging through the appendices (this seems to be the central theme of the remaining issues, particularly those regarding clarity of presentation), I must maintain my opinion that the paper is not appropriate as a conference manuscript. Quite honestly, I think this paper should (ultimately) be accepted and published in a top venue, but if the authors do not believe it is possible to fit into the conference format then perhaps a journal is more appropriate.

---

### Official Review · Reviewer_y6Sh · 2024-11-02

**Soundness:** 3
**Presentation:** 4
**Contribution:** 2
**Rating:** 6
**Confidence:** 4

**Summary:**

The paper considers a networked learning setting for mean field games that is in-between centralized learning and independent learning. Agents independently learn as in prior work, but share policies through a graph to reduce sample complexity by spreading better policies. The main contribution is theoretical, implying that communication should result in better sample complexity than independent learning. The resulting algorithms were also empirically verified on two problems.

**Strengths:**

- The paper is very well written and very clear on the main ideas.
- The work novelly fuses analysis of networked learning with the more realistic recent independent learning concept in MFGs.
- The theoretical results from analyzing the number of independent learned policies are elegant, interesting and significant, as they imply the usefulness of communication in the field of learning in MFGs.
- The discussion of related work and placement of contribution is very extensive and good.

**Weaknesses:**

- Although the primary contribution is theoretical, the empirical results are limited in terms of considered problems and their complexity. The exact definition of the problems remains unclear to me, but as far as I understand there is no example with mean field interaction in the state dynamics.
- The performed empirical changes discussed on page 10 were not empirically shown to be important / no ablation was performed.
- The robustness results are limited to failure in the communication networks and naturally the numbers of agents, so I am not sure what is the benefit over independent learning for robustness.
- See also questions below.

**Questions:**

- What is the benefit over independent learning and centralized learning in terms of robustness? In Fig. 3 it seems to me that all the curves observe the same relative shock.
- I am not sure why it would make sense to "cooperate" in a competitive game (sharing policies with other agents).
- Could one consider other types of robustness, e.g., against self-interested agents manipulating the policies of others, or model errors / heterogeneity?
- How is the approximation of decrease in divergence obtained in Appx. B.7?
- Figures 1, 2, 3 are hard to read due to their small size. What is depicted in the shaded region?

---

> ### Author Response · Authors · 2024-11-20
> **Part 1: Rebuttal by Authors**
>
> Thank you very much for reviewing our paper. We are pleased that you find it 'novel','very well written' and 'very clear', with theoretical results that are 'elegant, interesting and significant', and 'very extensive and good' placement of contribution and discussion of related work.
>
> Weakness 1: The grid-world games we choose are similar to those used in related works on MFGs, as we note in Lines 470-472 (for those that include experiments at all, which many theoretical works do not!). In fact, our tasks, where agents must coordinate on one of numerous possible equilibria, are more complex to solve than those in other works, which often focus on 'exploration' games, where agents must simply spread throughout the environment. The definitions of the tasks are given in Appx. E.1, though please do let us know if we can clarify these further. You are right that the transition function does not depend on the mean-field distribution in our examples here; we will add this to Appx. G as an element to explore in future work.
>
> Weakness 2: When we ran our original networked algorithm (as well as the centralised and independent alternatives) without the replay buffer, we did not find any discernable learning or improvement in return/exploitability at all, despite leaving the algorithms running for several days and many thousands of $k$ iterations. We are currently rerunning the original versions of the algorithms without the buffers to provide empirical evidence of this lack of learning in practical time. We will update the manuscript with these ablation results when they are ready, hopefully by the end of this discussion period, or otherwise in time for the final version (since they are inherently slow to run).
>
> Weakness 3: Please note that our robustness results do not consider failures in the communication network, but rather failures of the learners to update their policies in a given iteration (see Lines 482-484, 1561-1600, and also 028, 055, 116). As elaborated in Lines 1561-1600 and the captions to Figs. 2 and 5, when independent learners fail to update, the improvement of the whole population is slowed. In contrast, networked communication provides redundancy in case of these failures, with the updated policies of any agents that have managed to learn spreading through the population to those that have not. This feature thus ensures that improvement can continue for potentially the whole population even if a high number of agents do not manage to learn at a given iteration. We can see this benefit in Figs. 2 and 5, where networked learners outperform the independent learners, and generally also the centralised learners (where if the central learner fails to update, none of the population can improve, unlike in our networked case).
>
> Question 1: See Weakness 3 for discussion of the benefit of our networked architecture over the others in terms of robustness to failures to learn an updated policy in a given iteration. Regarding robustness to population increase, in Fig. 3, the networked case with the largest broadcast radius (pink, 1.0) $\textit{does not experience any shock at all}$. The centralised case also appears to experience a far greater initial shock than the networked cases, getting as bad as the independent curve, albeit quickly recovering. Similarly in Fig. 6, the centralised case has a shock doubly as large as most networked cases (apart from that with the smallest broadcast radius (green, 0.2)), while the largest broadcast radius (pink, 1.0) again has no shock at all. Recall also that in our experiments the networked agents use only $\textit{a single communication round}$ per $k$ iteration to demonstrate the benefit of even minimal communication; having more $C$ rounds would further reduce the shock observed by the networked cases by ensuring that learnt policies reached all the newly joined agents before the next iteration. Meanwhile the centralised and independent cases would still observe their large shocks, (if independent learners have managed to improve at all in the first place). Moreover, a centralised learner may in any case be an unrealistic assumption in real-world scenarios.
>
> Question 2: As we note in Lines 079-083: "We focus on 'coordination games', i.e. where agents can increase their individual rewards by following the same strategy as others and therefore have an incentive to communicate policies, even if the MFG setting itself is technically non-cooperative."
>
> Continued in next comment...

---

> ### Author Response · Authors · 2024-11-20
> **Part 2: Rebuttal by Authors**
>
> ... Continued from above.
>
> Question 3: Our work inherently also considers robustness to poor local estimation of the Q-function (i.e. model errors), where we find that with fewer samples / TD updates, the margin by which our networked architecture outperforms the alternatives increases. This is due to our method for spreading policies that are estimated to receive a higher return in reality despite being generated from poorly estimated Q-functions - we note this in Lines 019, 417-420, 1870 and 2173. One could of course consider other types of robustness, and we do discuss scenarios such as you suggest in Lines 2144-2153 of our future work section. We hope that the fact that our work opens ideas for further potentially fruitful research avenues is not a reason for a low score.
>
>
> Question 4: Nejad et al. (2009) [1] investigates the convergence rate of max-consensus algorithms such as that occurring under these conditions. In their Corollary 4.2, they prove that to achieve max-consensus, "the required number of communication instants is the maximum of the shortest path length between any pair of nodes", which is the definition of the diameter of the network. We denote this diameter as $d_\mathcal{G}$; they use $l$. They then give $\frac{1}{l}$ ($\frac{1}{d_\mathcal{G}}$ in our notation) as "the convergence rate of the max-consensus algorithm". In other words, if we start with $N$ policies in the population and finish with 1, the reduction in the number of policies with each round is equal on average, i.e. the reduction is linear. If the number of policies in the population decreases at a rate of $\frac{1}{d_\mathcal{G}}$, we can also say that the policy divergence
>
> (defined as $\Delta_{k} := \sum_{i=1}^N \|\|\pi^i_k -\pi^{\mathrm{max}}_k\|\|_1$,
> for the consensus policy $\pi^{\mathrm{max}}_k$)
>
> decreases at a rate of approximately $\frac{1}{d_{\mathcal{G}}}$. We can clarify this in the manuscript if deemed helpful.
>
>
> Question 5: Apologies for this: the small size was necessary to fit the page count, but we will reproduce these plots larger in the appendices to aid visibility. As noted in Appx. E.2, the shaded regions are 2-sigma error bars, i.e. a 95.4\% confidence interval.
>
> Thank you again for your time and input; we hope that our answers to your questions will encourage you to raise your score.
>
> [1] B. M. Nejad, S. A. Attia and J. Raisch, "Max-consensus in a max-plus algebraic setting: The case of fixed communication topologies," 2009 XXII International Symposium on Information, Communication and Automation Technologies, Sarajevo, Bosnia and Herzegovina, 2009, pp. 1-7, doi: 10.1109/ICAT.2009.5348437.

---

> > ### Author Response · Authors · 2024-11-25
> > **Discussion period closing**
> >
> > Since the discussion period has nearly finished, we wanted to check that we have been able to address all of your concerns. If there are no further clarifications required, we hope that you will feel able to raise your score. Thank you very much for your time.

---

> > > ### Comment · Reviewer_y6Sh · 2024-11-25
> > >
> > > I thank the authors for their very detailed response and nice work. My questions have been answered. I would like to keep my score, as there is no score 7 and I still feel the contribution / significance / impact for ICLR may be somewhat limited (coordination games and MFGs is a bit niche).

---

### Official Review · Reviewer_HFSn · 2024-11-04

**Soundness:** 3
**Presentation:** 3
**Contribution:** 3
**Rating:** 6
**Confidence:** 3

**Summary:**

This paper proposes a networked communication multi-agent reinforcement learning framework to solve mean-field games. The paper first provides theoretical insights into this networked communication paradigm, including the conceptual, sample efficiency, and convergence speed relationships between networked communication learning and centralized versus decentralized learning approaches. Based on this, the paper presents the practical implementation of the paradigm, including a specific method for generating communication content and the introduction of an experience replay buffer to accelerate learning. Through experiments on two grid-world scenarios, the authors demonstrate that the networked communication-based algorithm achieves performance comparable to centralized algorithms, along with higher robustness.

**Strengths:**

1. The theoretical analysis in the paper is comprehensive and aligns with intuition. The authors point out that the networked communication paradigm conceptually lies between centralized and decentralized approaches and demonstrate that this is also true in terms of sample efficiency and convergence speed, which is very interesting.
2. The experimental results demonstrate that the networked communication method is robust and high-performing, combining the advantages of centralized and decentralized framework. This is consistent with the authors' claims.
3. The paper is well-written and easy to follow. The authors provide all the necessary theories and experimental settings in the main text, with more detailed proofs and specifics included in the appendix.

**Weaknesses:**

1. In the theoretical section, the authors present a highly general framework (Algorithm 1), which can evolve into various specific algorithms by adjusting parameters. However, I note that the authors do not compare their practical algorithm with existing MFG algorithms in their experiments. Can any existing algorithms be compared in the experiments?
2. The communication concept in the paper differs from the typical concept of communication in multi-agent reinforcement learning, which may cause confusion for readers. In those works, communication content between agents is often a learnable vector. In contrast, the networked communication paradigm proposed here is more akin to an evolutionary paradigm, where the policies with higher estimated can propagate within the population.

**Questions:**

1. How much improvement in learning speed does the introduction of the experience replay buffer bring?
2. How are the RL states specifically configured in the experiments?

---

> ### Author Response · Authors · 2024-11-20
> **Rebuttal by Authors**
>
> Thank you very much for reviewing our paper. We are really pleased that you find it 'well-written and easy to follow' and 'very interesting', with theoretical analysis that is 'comprehensive' and intuitive, and experimental results that are 'consistent' with this.
>
> Weakness 1: Within our specific setting of learning with the finite empirical population through a single continuous evolution of the system, the only existing algorithms are those of Yardim et al. (2023) for the centralised and decentralised cases. However, as we discuss in our paper, while these algorithms are amenable to theoretical analysis (sample guarantees), they do not learn in computationally practical time. Specifically, when we tried to run these algorithms with their theoretically required hyperparameters, $\textit{we did not find any discernable learning or improvement in return/exploitability at all}$, let alone convergence, despite leaving the algorithms running for several days and many thousands of $k$ iterations. Yardim et al. (2023) also do not provide any empirical demonstrations of their algorithms. Therefore, as we mention in Lines 467-470 and elsewhere, we also update the centralised and independent theoretical algorithms with replay buffers (as well as doing so for our own networked algorithm), so that we can compare with at least existing 'architectures' if not the algorithms themselves (which would simply show no noticeable learning at all). Nevertheless, we are currently rerunning the original versions of the algorithms without the buffers to provide empirical evidence of this lack of learning in practical time. This will serve both as a comparison with existing algorithms, and should also address your Question 1. We will update the manuscript with these results when they are ready, hopefully by the end of this discussion period, or otherwise in time for the final version (since they are inherently slow to run).
>
> Weakness 2: Please note that our algorithms are not themselves MARL algorithms; we are using MFG algorithms to approximate the solution of finite-player games with large populations, since MARL struggles to scale computationally in such scenarios. As you suggest, agents in MARL algorithms might, for example, communicate the 'learnable vector' representing their local estimate of the global shared Q-function, so as to average their local estimates. Since in our MFG setting we are ultimately interested in finding high-performing equilibrium policies, we directly privilege the spread of Q-functions that lead to higher-performing policies (where each agent's policy is computed directly from its learnt Q-table as per Def. 8), rather than simply averaging them regardless of quality. In order to estimate the quality of an estimated Q-function, we compute the policy maximising it and observe the return of the policy in the environment (Lines 17-23 of Algorithm 2). Having already computed the policy from the Q-function for evaluation purposes, it is more computationally efficient to communicate the policy rather than the learnable vector from which it is derived, only to have to derive the policy from it again in the next iteration. You are right that our approach has parallels with evolutionary algorithms, and we ourselves discuss this similarity at length in Lines 2106-2122 of Appx. F.
>
> Given that we explain our approach explicitly and you praise our paper as being 'well-written and easy to follow', we are surprised that you find this 'may cause confusion for readers'. This criticism seems speculative, especially as no other reviewers have raised this as a concern. We therefore hope this is not a cause for a lower rating, but we are nevertheless happy to clarify this in the paper if deemed useful.
>
> Question 1: (See also Weakness 1 above.) When we ran our original networked algorithm (as well as the centralised and independent alternatives) without the replay buffers, we did not find any discernable learning or improvement in return/exploitability at all, despite leaving the algorithms running for several days and many thousands of $k$ iterations. Therefore by enabling discernable learning and indeed convergence, the replay buffer is a very significant improvement. We are currently rerunning the original versions of the algorithms without the buffers to provide empirical evidence of this lack of learning in practical time. We will update the manuscript with these results when they are ready.
>
> Question 2: In our grid-world environment, the current location of an agent in the environment is its local state. For these tabular algorithms, we convert the location coordinates into a scalar value (such that the top-left gridpoint is 0, and the bottom-right is $\left(\mathrm{dimension} \times (\mathrm{dimension} - 1)\right)$. These values are then indices for the rows of the Q-table, with a row for each state.
>
> Thank you again for your time and input. We hope that our clarifications will encourage you to raise your score.

---

> > ### Author Response · Authors · 2024-11-25
> > **Discussion period closing**
> >
> > Since the discussion period has nearly finished, we wanted to check that we have been able to address all of your concerns. If there are no further clarifications required, we hope that you will feel able to raise your score. Thank you very much for your time.

---

> > > ### Author Response · Authors · 2024-12-02
> > > **Discussion period closing again**
> > >
> > > Apologies for following up again, but we wanted to check before the end of the discussion period whether we have resolved all of your concerns, and if so whether you will be able to raise your score? Thank you very much for getting back to us.

---

### Official Review · Reviewer_xtau · 2024-11-04

**Soundness:** 2
**Presentation:** 2
**Contribution:** 2
**Rating:** 3
**Confidence:** 3

**Summary:**

This paper investigates learning in mean-field games with a communication network. Specifically, previous work investigated centralized (communicate arbitrarily) and independent learning (no communication at all) in mean-field games. This paper extends the result to the cases when players can only communicate with some neighbors, which is an interpolation between the centralized and the independent cases.

The contribution of the paper is two-fold. Firstly, it shows that when players can communicate partially, the sample complexity is bounded between that of the centralized and the independent cases. Secondly, it shows that the algorithm will converge faster empirically with a replay buffer, including centralized, independent, and partially communicated settings.

[1] Batuhan Yardim, Semih Cayci, Matthieu Geist, and Niao He. Policy Mirror Ascent for Efficient and Independent Learning in Mean Field Games. In International Conference on Machine Learning, pp. 39722–39754. PMLR, 2023.

**Strengths:**

- The authors provide experiments to justify the results.
- Using a communication network to interpolate between centralized and independent learning is interesting.

**Weaknesses:**

## Main Concerns
- The theoretical results in this paper do not show the superiority over independent learning. The assumptions in Theorem 3 are too restrictive ($\tau_k\to 0$) and the gap $(\frac{1}{d_{\mathcal G}})^C$ is small.

## Minor Weaknesses
- Writings: The writing of this paper can be further improved.
  - For example, in Section 3.2, where the authors first introduce $\sigma_{k+1}^i$, the authors may add a brief explanation about what it is.
  - On line 376, when $C\geq d_{\mathcal G}$, it is not $(\frac{1}{d_{\mathcal G}})^{C}=0$ but instead that term in the bound above disappears.

I would be glad to raise the score once the author addresses the issues above.

**Questions:**

- In Theorem 3, when $C$ grows larger, why does the gap between learning with partial communication and independent learning seem to be smaller instead of larger? Since more communication may further stabilize the learning process.
- Why when $d_{\mathcal G}$ is 1, the gap between learning with partial communication and centralized learning is 1? Since they are now identical to each other. Also, I'm wondering why the gap between learning with partial communication and independent learning is 0 when $d_{\mathcal G}$ is 1.
- Why is there no ablation study of replay buffer?

---

> ### Author Response · Authors · 2024-11-20
> **Rebuttal by Authors**
>
> Thank you for taking the time to review our paper, and we are very pleased that you find our approach ‘interesting'.
>
> Main concerns: We are surprised and confused by this criticism, since we do indeed prove theoretically that the sample complexity of the networked case is better than the independent case when there is communication, as you state yourself in your summary. This is not criticism expressed by any of the other reviewers, many of whom in fact specifically highlight this theoretical result: they say that we 'demonstrate that this is … true in terms of sample efficiency and convergence speed, which is $\textbf{very interesting}$' (HFSn); 'the main contribution is theoretical, implying that communication should result in better sample complexity … The theoretical results from analyzing the number of independent learned policies are $\textbf{elegant, interesting and significant}$, as they imply the usefulness of communication in the field of learning in MFGs' (y6Sh); 'the main theoretical result is $\textbf{particularly interesting}$ … proving that this is so is $\textbf{a very nice contribution}$' (yxK2).
>
> Why is the assumption that $\tau_k \rightarrow 0$ too restrictive? $\tau_k$ is a hyperparameter for which the value can be chosen by the person running the algorithm.  We assume to begin with in Thm. 3 that $\tau_k \rightarrow 0$ to ease the mathematical analysis, before specifically discussing the loosening of this assumption in Remark 5 (other assumptions are loosened in Remark 4).
>
> What do you mean for the gap to be 'small' - small with respect to what? Even if this were true, it would directly contradict your statement that we 'do not show the superiority over independent learning', since there nevertheless is a gap even if 'small'. In fact, please recall that the term $\mathcal{O}\left(\left(\frac{1}{d_\mathcal{G}}\right)^C\right)$ does not represent a specific or fixed numerical difference, but rather describes the asymptotic scaling behaviour of the difference with respect to $d_\mathcal{G}$ and $C$. So this term indicates the dependence of the size of the gap on size of the network and the amount of communication, rather than indicating its exact size, so the gap may not be small at all.
>
> Minor weakness 1: In Lines 268-272, we do indeed already have a brief explanation of $\sigma^{i}_{k+1}$ when we first introduce it, as you suggest. Please let us know if there is something specific that can be clarified here.
>
> Minor weakness 2: We wrote it this way for simplicity, but you are technically correct. We can instead write the term $\left(\frac{1}{d_\mathcal{G}}\right)^C$ in the bound in Line 373  as the piecewise function $f(C, d_\mathcal{G})$, defined as the following:
>
> $f(C, d_\mathcal{G}) =
> \begin{cases}
> \left(\frac{1}{d_\mathcal{G}}\right)^C &\text{if } C < d_\mathcal{G},\\\\
> 0 &\text{if } C \geq d_\mathcal{G}
> \end{cases}
> $ .
>
> Question 1: This is not the case - the gap does indeed grow larger as $C$ grows larger. Larger $C$ means $\left(\frac{1}{d_\mathcal{G}}\right)^C$ gets smaller, which in turn means that $\left(1 - \left(\frac{1}{d_\mathcal{G}}\right)^C\right)$ gets larger, and hence also the gap. I.e. when $C=0$, $\left(\frac{1}{d_\mathcal{G}}\right)^C = 1$, and hence $\left(1 - f(C, d_\mathcal{G})\right) = 0$, while when $C \geq d_\mathcal{G}$, $f(C, d_\mathcal{G})=0$ and hence $\left(1 - f(C, d_\mathcal{G})\right) = 1$, with the term increasing monotonically between 0 and 1 as $C$ increases from 0 to $d_\mathcal{G}$.
>
> Question 2: Recall that the piecewise function $f(C, d_\mathcal{G})$ is such that when $d_\mathcal{G}$ is 1, $f(C, d_\mathcal{G}) = 0$ for any case other than $C=0$. Thus if there is any communication when $d_\mathcal{G} = 1$, the difference term between the centralised and networked cases reduces to 0, indicating no difference, as is expected. Likewise, the difference term between the independent and networked cases reduces to $\mathcal{O}(1-0) = \mathcal{O}(1)$, i.e. the difference is non-decaying.
>
> Question 3: When we ran our original networked algorithm (as well as the centralised and independent alternatives) without the replay buffer, we did not find any discernable learning or improvement in return/exploitability at all, despite leaving the algorithms running for several days and many thousands of $k$ iterations. We are currently rerunning the original versions of the algorithms without the buffers to provide empirical evidence of this lack of learning in practical time. We will update the manuscript with these ablation results when they are ready, hopefully by the end of this discussion period, or otherwise in time for the final version (since they are inherently slow to run).
>
> We hope that our clarifications show that our work is ready for acceptance, and that this will encourage you to significantly raise your score.

---

> > ### Comment · Reviewer_xtau · 2024-11-22
> >
> > Thank you for your response! Now I have a better understanding of the paper.
> >
> > For $\sigma_{k+1}^i$, I think an intuitive explanation about why the algorithm needs this and what this is will be helpful.
> >
> > Additionally, I still have a few questions.
> >
> > - In Theorem 3, should it be $\Theta$ or $\Omega$ for the gap between $ub_{net}$ and $ub_{ind}$ instead of $O$?
> > - May you write explicitly what is in the $O$ of Theorem 3?

---

> > > ### Author Response · Authors · 2024-11-22
> > >
> > > Thank you very much for reading our rebuttal, and we are glad that this has helped your understanding.
> > >
> > > Point 1: $\sigma_{k+1}^{i}$ is a scalar value generated in association with and coupled to a policy $\pi_{k+1}^{i}$. The value provides information that helps agents decide between policies that they may wish to adopt from neighbours. Different methods for choosing between values received from neighbours $\left(\sigma_{k+1}^{j} \forall j \in J^i_t \right)$ and generating the values in the first place, leads to different ways for policies to spread through the population. For example, generating or choosing $\sigma_{k+1}^{i}$ at random leads to policies being exchanged through the population at random (as is required in Thm. 1), whereas generating $\sigma_{k+1}^{i}$ as an approximation of the return of $\pi_{k+1}^{i}$ and then selecting the highest received value of $\sigma_{k+1}^{j}$ leads to better performing policies spreading through the population. The latter is the approach we use for our empirical results, albeit that we use a softmax rather than a max function for selecting between received values.
> > >
> > > We will add this explanation to the text if deemed helpful.
> > >
> > > Question 1: Thank you for this question - you are right that the tight bound $\Theta$ might be more mathematically appropriate here - we can make this change in the manuscript.
> > >
> > > Question 2: As you can see from Lines 1447-1467 of Appx. B.6, we are using this notation to contain the coefficient by which the value in the brackets -- $\left(1 - \left(\frac{1}{d_\mathcal{G}}\right)^C\right)$ or $\left(\left(\frac{1}{d_\mathcal{G}}\right)^C\right)$ -- is multiplied, namely $\xi\sum_{k=1}^{K-1}L_{\Gamma_{\eta}}^{K-k-1}\mathbb{E}\left[\Delta_{k,0}\right]$. I.e. we abstract away this term to focus on the dependence on $d_\mathcal{G}$ and $C$.
> > >
> > > Please let us know if there is anything else we can clarify, otherwise we hope these explanations will encourage you to raise your score. Thank you again for your time in reviewing our work.

---

> > > > ### Comment · Reviewer_xtau · 2024-11-23
> > > >
> > > > Thank you for your response.
> > > >
> > > > I'm wondering on line 1423: *there is a decrease in the divergence by a factor of approximately ...*. Why it is approximately and what is omitted?
> > > >
> > > > Moreover, in Theorem 3, when $d_G$ is larger, the gap between $ub_{cent}$ and $ub_{net}$ gets smaller. Why is this the case?

---

> ### Author Response · Authors · 2024-11-24
>
> Thank you for reading our response.
>
> Question 1: Nejad et al. (2009) [1] gives $\frac{1}{d_\mathcal{G}}$ ($\frac{1}{l}$ in their notation) as "the convergence rate of the max-consensus algorithm" affecting the $\textit{number of policies}$ in the population. I.e., if we start with $N$ policies in the population and finish with 1, it will take $d_\mathcal{G}$ rounds, with the reduction in the number of policies with each round being $\textit{approximately}$ $N/d_\mathcal{G}$. The exact amount will depend on the exact structure of the network and the exact initial distribution of policies, so we can only give an approximate decrease. If the number of policies in the population decreases by approximately a factor of $\frac{1}{d_\mathcal{G}}$ with each round, we can also say that the policy divergence
>
> (defined as $\Delta_{k} := \sum_{i=1}^N \|\|\pi^i_k -\pi^{\mathrm{max}}_k\|\|_1$,
> for the consensus policy $\pi^{\mathrm{max}}_k$)
>
> decreases by $\textit{approximately}$ a factor of $\frac{1}{d_\mathcal{G}}$ with each round. We can clarify this in the manuscript if deemed helpful.
>
>
> Question 2: Thank you for pointing this out. Thm. 3 should instead have
>
> $$ub_{cent} + \Theta\left(1-\left(f(C, d_\mathcal{G})\right)\right) \\;\\; \approx \\;\\;  ub_{net} \\;\\;  \approx \\;\\;  ub_{ind} - \Theta\left(f(C, d_\mathcal{G})\right),$$
>
> where $f(C, d_\mathcal{G})$ is the piecewise function defined as the following:
>
> $f(C, d_\mathcal{G}) =
> \begin{cases}
> \left(\frac{C}{d_\mathcal{G}}\right) &\text{if } C \leq d_\mathcal{G},\\\\
> 1 &\text{if } C > d_\mathcal{G}
> \end{cases}
> $.
>
> The $\Theta$ still contains the term $\xi\sum_{k=1}^{K-1}L_{\Gamma_{\eta}}^{K-k-1}\mathbb{E}\left[\Delta_{k,0}\right]$, which multiplies the term in the brackets.
>
> We will update the proof appropriately, beginning from Lines 1424-1428, which should instead read:
>
> $$ \mathbb{E}\left[\Delta_{k,c+1}\right] \approx \mathbb{E}\left[\Delta_{k,c}\right] \times \left(1 - \frac{1}{d_\mathcal{G}}\right) \text{, giving}$$
>
>  $$
>  \mathbb{E}\left[\Delta_{k,C}\right] \approx \mathbb{E}\left[\Delta_{k,0}\right] \\times \left(1 - \frac{C}{d_\mathcal{G}}\right).
>  $$
>
> Thank you again for engaging with our rebuttal; we hope that these changes will convince you that our paper is ready to accept.
>
> [1] B. M. Nejad, S. A. Attia and J. Raisch, "Max-consensus in a max-plus algebraic setting: The case of fixed communication topologies," 2009 XXII International Symposium on Information, Communication and Automation Technologies, Sarajevo, Bosnia and Herzegovina, 2009, pp. 1-7, doi: 10.1109/ICAT.2009.5348437.

---

> > ### Comment · Reviewer_xtau · 2024-11-24
> >
> > Thank you for your response.
> >
> > Could you explain how to derive the relation between $\Delta_{k,c+1}$ and $\Delta_{k,c}$?

---

> ### Author Response · Authors · 2024-11-24
>
> As per the previous comment, the number of policies in the population decreases from $N$ to 1 over the $d_\mathcal{G}$ communication rounds, so the policy divergence decreases by approximately a factor of $\frac{1}{d_\mathcal{G}}$ with each round, on average. Therefore we can say that
>
> $$ \mathbb{E}\left[\Delta_{k,c+1}\right] \approx \mathbb{E}\left[\Delta_{k,c}\right] - \left(\mathbb{E}\left[\Delta_{k,c}\right]  \times \frac{1}{d_\mathcal{G}}\right) \text{, simplifying to}$$
>
> $$ \mathbb{E}\left[\Delta_{k,c+1}\right] \approx \mathbb{E}\left[\Delta_{k,c}\right] \times \left(1 - \frac{1}{d_\mathcal{G}}\right). \\;\\; \text{(1)}$$
>
> As above, this is approximate, with the exact decrease depending on the structure of the network and distribution of the policies.
> The induction from (1) would give
>
> $$\mathbb{E}\left[\Delta_{k,C}\right] \approx \mathbb{E}\left[\Delta_{k,0}\right] \times \left(\left(1 - \frac{1}{d_\mathcal{G}}\right)^C\right),$$
>
> which does not exactly give rise to (2) below, which is the (more important) approximately linear relation regarding the decrease in the number of policies averaged over the total number of communication rounds. Hence we can remove (1) if this is confusing.
>
> This more important (and cumulatively accurate) relation for our purposes is the second one given in the comment above, namely:
>
> $$
> \mathbb{E}\left[\Delta_{k,C}\right] \approx \mathbb{E}\left[\Delta_{k,0}\right] - \left(\mathbb{E}\left[\Delta_{k,0}\right] \\times \frac{C}{d_\mathcal{G}}\right) \text{, which simplifies to}
>  $$
>
> $$
>   \mathbb{E}\left[\Delta_{k,C}\right] \approx \mathbb{E}\left[\Delta_{k,0}\right] \\times \left(1 - \frac{C}{d_\mathcal{G}}\right). \\;\\; \text{(2)}
>  $$
>
> This holds for $C\leq d_\mathcal{G}$; when $C = d_\mathcal{G}$ we have $\mathbb{E}\left[\Delta_{k,C}\right] = 0$ as expected. The divergence cannot be negative so there is no further change for $C> d_\mathcal{G}$.
>
> We hope that this answers your question.

---

> > ### Comment · Reviewer_xtau · 2024-11-24
> >
> > Thank you for your prompt response. However, I think the number of policies does not decrease uniformly in most graph structures. For instance, in a random graph with each player $w$ degrees, $E[\Delta_{k, C}]=O(N-w^C)$ if my understanding of the paper is correct. Therefore, an accurate estimation of $\Delta_{k,C}$ might be better.

---

> > > ### Author Response · Authors · 2024-11-24
> > >
> > > Thank you for your reply - do you have a reference for this?
> > >
> > > In this case we can remove (2) and use the inductive relation from above:
> > >
> > > $$\mathbb{E}\left[\Delta_{k,C}\right] \approx \mathbb{E}\left[\Delta_{k,0}\right] \times \left(\left(1 - \frac{1}{d_\mathcal{G}}\right)^C\right).$$
> > >
> > > This would ultimately give
> > >
> > > $$ub_{cent} + \Theta\left(f(C, d_\mathcal{G})\right) \\;\\; \approx \\;\\;  ub_{net} \\;\\;  \approx \\;\\;  ub_{ind} - \Theta\left(1- f(C, d_\mathcal{G})\right),$$
> > >
> > > where $f(C, d_\mathcal{G})$ is the piecewise function defined as the following:
> > >
> > > $f(C, d_\mathcal{G}) =
> > > \begin{cases}
> > >  \left(\left(1 - \frac{1}{d_\mathcal{G}}\right)^C\right) &\text{if } C < d_\mathcal{G},\\\\
> > > 0 &\text{if } C \geq d_\mathcal{G}
> > > \end{cases}
> > > $.

---

> > > > ### Comment · Reviewer_xtau · 2024-12-02
> > > >
> > > > I do not have a reference for it, but I think it can be proved by directly using the [definition](https://en.wikipedia.org/wiki/Erd%C5%91s%E2%80%93R%C3%A9nyi_model).
> > > >
> > > > I agree with the bound above. My major concern is that the current statement is not rigorous, especially the *"approximate"* in the proof. I think writing explicitly in the main paper about how $f(C, d_{G})$ corresponds to the graph might be better. Moreover, it is not a function of $C, d_{G}$ but rather a function of the communication graph.
> > > >
> > > > Therefore, I think the current version is not ready for published and I will maintain my score.

---

> > > > > ### Author Response · Authors · 2024-12-02
> > > > >
> > > > > Thank you very much for your response.
> > > > >
> > > > > Firstly, it is not necessarily true to say that the network is a random graph, as the edges might be influenced by the behaviour/movement of the agents (which in turn might depend on their policies, which in turn might depend on the network), such that the edges might not be independent. This is a reason for us seeking to keep our theorem general, as using the Erdős–Rényi model would again only be approximate.
> > > > >
> > > > > Secondly, it is of course true that the bound ‘is not a function of $C, d_{G}$ but rather a function of the communication graph’, but since the decrease in the number of policies within each communication round depends on the exact structure/connectivity of the specific time-varying network, surely we can only speak in more general terms about more general convergence rates, which will only give an approximate result dependent on $C, d_{G}$?
> > > > >
> > > > > What could we do to make the statement more ‘rigorous’, as presumably even if we clarify in the main paper that $f(C, d_\mathcal{G})$ relates to the reduction in the number of policies in the graph, the result will still be ‘approximate’? We could perhaps write the bound ‘in expectation to’ an infinite set of random graphs in the Erdős–Rényi model, but this would still be an approximation since the graphs are not necessarily random as noted above? Moreover the reduction in the number of policies only approximately relates to the reduction in policy divergence, defined as $\Delta_{k,c} := \sum_{i=1}^N \|\|\pi^i_{k,c} -\pi^{\mathrm{consensus}}_k\|\|_1$, where the precise reduction in this quantity relates to the size of the difference between the policies that are lost from or kept in the population.
> > > > >
> > > > > Or perhaps we could use a proportionality relation, though the same issues may still apply?
> > > > >
> > > > > Thank you very much for your help in improving our paper.

---

### Official Review · Reviewer_jMnr · 2024-11-05

**Soundness:** 2
**Presentation:** 2
**Contribution:** 1
**Rating:** 1
**Confidence:** 4

**Summary:**

This paper aims to introduce networked communication to the mean field game framework and show how communication accelerates convergence of the learning. Two games are provided, namely, cluster and agree on a single target, to validate the proposed framework.

**Strengths:**

It introduces a practical communication network to accelerate the convergence of independent learning, without relying on a central learner.

**Weaknesses:**

1. The experiments have shown extensive results but only on two games. The insights are more on the practical side that introducing communication could benefit convergence.
2. The theoretical contribution of this paper is unclear. It showed that practically introducing communication could benefit convergence, but where the theoretical contribution lies is not well motivated.

**Questions:**

The statements are not clear. MARL is a learning framework assuming a finite number of N agents, while MFG is a game-theoretical framework with N-player limit. My understanding is that this paper aims to address the challenge in MARL when the population size grows, but still finite. Thus, it remains unclear how the proposed algorithm is related to MFG, and how MFG can help with the learning of N agents even under the communication scheme. Could the authors clarify the relationship?

---

> ### Author Response · Authors · 2024-11-20
> **Rebuttal by Authors**
>
> Thank you very much for reviewing our paper and for noting its 'extensive' experimental results.
>
> Weakness 1: We are surprised that you both praise our 'extensive' experimental results while taking issue with us 'only' experimenting on two games. Experiments on two games is very typical for works on MFGs (for those that include experiments at all, which many - especially theoretical works - do not!), and indeed the games we choose are very similar to those used in similar works, as we note in Lines 470-472.  It is also not clear to us whether your statement that 'the insights are more on the practical side' is a point of praise or a point of criticism that we should be responding to - can you please clarify this?
>
> Weakness 2: You assert that our theoretical contribution is unclear and that where the theoretical contribution lies is not well motivated. This is in stark contrast with the other reviewers, who praise it as: 'novel','very well written' and 'very clear', with theoretical results that are 'elegant, interesting and significant', and also with 'very extensive and good' placement of contribution and discussion of related work ($\textbf{y6Sh}$); 'a very nice contribution' with the theoretical result 'particularly interesting' ($\textbf{yxK2}$); and 'well-written and easy to follow', with theoretical analysis that is 'comprehensive' and intuitive ($\textbf{HFSn}$). We state precisely the nature of our theoretical contribution in Lines 100-104 and summarise it again in Lines 300-305. We find your criticism regarding the theoretical contribution unfairly vague - can you please state more precisely what about the contribution you find unclear and what you would like to be better motivated?
>
> Questions:
>
> Again, your criticism that 'the statements are not clear' is very general and goes against the impression of the other reviewers - can you please let us know which specific statements you find unclear so that we can try to clarify them?
>
> The statement in your question that an MFG has an 'N-player limit' is wrong or misleading. The MFG framework assumes an $\textit{infinite}$ number of players, as we state in both Lines 036-037 and Lines 162-164. Our work does not directly study MARL and we do not present MARL algorithms. Instead, as we describe at the beginning of our introduction, we use the scalability issue facing MARL as a $\textit{motivation}$ for our work on MFGs, as do many other works on MFGs. Since they assume an infinite population, MFGs do not face a scalability issue, and at the same time the solution of the MFG can be used as an approximate solution to the finite-player game in which we may have originally been interested but which is difficult to solve in itself. We already state this in the paper in Lines 042-047 and 182-190. Our algorithms are fundamentally MFG algorithms (they assume an infinite population of symmetric anonymous agents) and relate to other works on MFGs such as Yardim et al. (2023); this is not a concern raised by any other reviewers. To summarise, we learn the solution to the MFG by approximating the assumed infinite population using the finite empirical population of $N$-agents - in turn, we can use the solution to the MFG as an approximate solution to the finite $N$-player game.
>
> We hope that these explanations are helpful, and will encourage you to raise your score significantly.

---

> > ### Author Response · Authors · 2024-11-25
> > **Discussion period closing**
> >
> > Since the discussion period has nearly finished, we kindly ask you to read and respond to our rebuttal. We have provided an extensive response to your review, and we wanted to check that we have been able to address all of your concerns. If there are no further barriers to acceptance, we would appreciate if you could update your score accordingly. Thank you very much for your time.

---

> > ### Comment · Reviewer_jMnr · 2024-11-28
> >
> > Thanks for the authors’ efforts trying to address my comments. I politely disagree with the authors’ criticism of my response that they are contradictory. My points are that the analysis is comprehensive but on only two games. More experiments would help strengthen the value of the proposed method.
> > In practice, communication comes at cost. How to balance the cost and communication efficiency could be a challenge. I think the authors really improved the readability of the paper and I believe this topic is important, especially for engineering and practical applications. Unfortunately, it is unclear whether this paper aims to contribute to mean-field approximation for MARL, or mean-field games, which are quite different but are not distinguished clearly in the manuscript.

---

> > > ### Author Response · Authors · 2024-11-28
> > >
> > > Thank you for responding to our rebuttal.
> > >
> > > In our experiments, the networked agents only have a $\textit{single}$ round of communication per $k$ iteration, i.e. we demonstrate the substantial advantages that come from even this very low-cost communication. Further rounds of communication might give an even greater benefit, but are clearly not required to make the difference between $\textit{no improvement at all}$ in the independent case and equivalent learning speed to the centralised case in Fig. 7.
> > >
> > > Can you please suggest some specific experiments that you think would be useful to explore the cost-benefit tradeoff that you perceive in the communication? Your statement that the tradeoff 'could be a challenge' otherwise feels generic and not sufficient to justify your very low rating of 1, especially since you nevertheless believe the topic of communication 'is important, especially for engineering and practical applications'.
> > >
> > > As we stated in our original rebuttal comment and in the paper itself, we do not aim to contribute to mean-field approximation for MARL, and instead contribute to empirical MFG algorithms which can find approximate solutions to finite-player games with large populations. They can therefore be used to circumvent the scalability problem found in MARL. Since, as you say, MARL and MFGs 'are quite different', and we only use a problem with the former as a motivation for the latter, we do not discuss the distinction at length in the manuscript: this is already well known and related papers - such as those we cite - approach the topic in the same way as us. Nevertheless we are happy to make the distinction clearer in the camera-ready version (unfortunately the deadline to update the manuscript within the discussion period has now passed). In any case, this does not seem to be a reason to "strongly reject" our paper.
> > >
> > > We are glad that you think we have 'really improved the readability of the paper' since this was one of your original main concerns, and also that you think the topic is important. We hope that in light of this you will feel able to increase your score for presentation as well as the overall rating, which does not appear to reflect your qualitative sentiment.

---

### Author Response · Authors · 2024-11-28
**Suggested revisions made to manuscript**

We have revised our manuscript with the updates indicated in our rebuttal comments, to address concerns raised in your reviews. In particular, we:


 1. Added an ablation study of our experience replay buffer (Lines 2050-2088) (xtau, HFSn, y6Sh, yxK2).
2. Reproduced the figures from our main text at a larger size in the appendix (y6Sh, yxK2).
3. Added an intuitive explanation of what $\sigma_{k+1}^i$ is and why the algorithm benefits from it (Lines 264-301) (xtau, yxK2).
4. Updated Thms. 2 and 3 to more accurately and clearly characterise the bounds and rates; we also update their proofs accordingly (xtau, yxK2).
5. Clarify the source of the decrease in divergence and why it is approximate (Lines 1434-1438) (xtau, y6Sh, yxK2).
6. Added experiments involving distribution-dependent transition functions as an element for future work (Lines 2367-2369) (y6Sh).
7. Moved Alg. 1 to be on the same page as its description (yxK2).
8. Included the definition of $\varepsilon$ in Thm 2 (Line 354) (yxK2).
9. Added a definition of the network’s diameter (Line 205) (yxK2).
10. Clarified the tractability issue of non-stationary MFGs (Lines 039-040) (yxK2).
11. Provided theoretical analysis as to why our networked agents can outperform centralised agents in practical settings (Lines 1521-1553) (yxK2).
12. Added a reference to the middle subfigure in Fig. 2 (Lines 505-507) (yxK2).


In accordance with these changes, we made other small edits to keep the main text within the 10 page limit.

Thank you very much for your suggestions, and we hope that these improvements to our paper address your concerns. $\textbf{Please let us know if you have outstanding questions, otherwise we hope you will be able to raise your scores in light of these improvements.}$

---

### Author Response · Authors · 2024-12-04
**Summary**

We thank all the reviewers for their time and input on our paper.

Reviewer y6Sh said our paper is 'novel','very well written' and 'very clear', with theoretical results that are 'elegant, interesting and significant', and 'very extensive and good' placement of contribution and discussion of related work. After our rebuttal, they added: 'I thank the authors for their very detailed response and nice work. My questions have been answered.'

Reviewer HFSn said our paper is 'well-written and easy to follow' and 'very interesting', with theoretical analysis that is 'comprehensive' and intuitive, and experimental results that are 'consistent' with this. They also said 'The authors provide all the necessary theories and experimental settings in the main text, with more detailed proofs and specifics included in the appendix.' We provided an extensive answer to their questions but received no additional follow-up from them, implying all of their questions were addressed.

Reviewer jMnr said our paper has 'extensive results' and 'the analysis is comprehensive', and that after our manuscript update 'the authors really improved the readability of the paper and I believe this topic is important, especially for engineering and practical applications'.

Reviewer xtau said our paper was 'interesting', and we significantly updated the paper to address their concerns - they said they agreed with our updates.

Reviewer yxK2 said our paper was 'a very nice contribution' with the theoretical result 'particularly interesting'. In response to our 'detailed responses' in rebuttal, they said 'I very much want to like this paper. The authors have done a nice job of clarifying several of the questions I had'. They also said they had an 'improved understanding of the paper ... Quite honestly, I think this paper should (ultimately) be accepted and published in a top venue.' They remained concerned that some of our technical details are in the appendix rather than the main text, but did not suggest what could be removed from the main text to accommodate the details from the appendix. Putting technical details in the appendix is extremely common for conference papers with page limits on the main text. No other reviewers said this impeded their understanding, and actively said how clear and easy to understand the paper is, with Reviewer HFSn specifically noting 'The authors provide all the necessary theories and experimental settings in the main text, with more detailed proofs and specifics included in the appendix.'

We hope that this shows our paper is ready for acceptance and will encourage reviewers to raise their scores further. We thank you again for your time.

---

### Meta-Review · Area_Chair_qCap · 2024-12-20

**Metareview:**

The paper explores a networked learning framework for Mean Field Games (MFGs), positioned between centralized and independent learning. It presents theoretical insights that suggest that communication in this networked setup can improve sample complexity and convergence speed. The theoretical results are supported by experiments on two grid-world scenarios.

Overall, the Reviewers appreciated the theoretical contributions. For instance, Reviewer y6Sh highlighted the "elegant and significant" results on the utility of communication in MFGs. However, concerns emerged regarding correctness, scope of the experiments, and clarity of the presentation:
- Reviewer xtau noted restrictive assumptions in Theorem 3 and was not sold on the empirical benefits of the proposed method over independent learning. Eventually this led to an in-depth analysis of the technical content. There was a long discussion where progress was made, but it looks like some points remained unresolved, with the last message of the discussion chain, belonging to the author, listing several questions. The last message from the reviewer reads "Therefore, I think the current version is not ready [to be] published and I will maintain my score."
- Reviewer yxK2 found the paper difficult to follow because the key definitions and details were relegated to the appendix. They seemed to be the only reviewer with this issue.
- Reviewer jMnr suggested that additional experiments on communication tradeoffs would improve the work. It remains somewhat unclear, however, what experiments the reviewer had in mind.

While I find the second and third concerns minor, the first one regarding technical details is substantial and needs to be addressed.

**Additional Comments On Reviewer Discussion:**

The discussion primarily focused on improving clarity and addressing concerns about empirical scope. The Reviewers appreciated the authors for their detailed response. Nevertheless, important concerns remained unresolved regarding clarity and technical details.

---

### Decision · Program_Chairs · 2025-01-22

Reject